# MASKED COMPLETION VIA STRUCTURED DIFFUSION WITH WHITE-BOX TRANSFORMERS

**Druv Pai**[*]   **Ziyang Wu**   **Sam Buchanan**   **Yaodong Yu**   **Yi Ma**
UC Berkeley     UC Berkeley     TTIC            UC Berkeley     UC Berkeley & HKU

## ABSTRACT

Modern learning frameworks often train deep neural networks with massive amounts of unlabeled data to learn representations by solving simple pretext tasks, then use the representations as foundations for downstream tasks. These networks are empirically designed; as such, they are usually not interpretable, their representations are not structured, and their designs are potentially redundant. White-box deep networks, in which each layer explicitly identifies and transforms structures in the data, present a promising alternative. However, existing white-box architectures have only been shown to work at scale in supervised settings with labeled data, such as classification. In this work, we provide the first instantiation of the white-box design paradigm that can be applied to large-scale unsupervised representation learning. We do this by exploiting a fundamental connection between diffusion, compression, and (masked) completion, deriving a deep transformer-like masked autoencoder architecture, called CRATE-MAE, in which the role of each layer is mathematically fully interpretable: they transform the data distribution to and from a structured representation. Extensive empirical evaluations confirm our analytical insights. CRATE-MAE demonstrates highly promising performance on large-scale imagery datasets while using only ∼30% of the parameters compared to the standard masked autoencoder with the same model configuration. The representations learned by CRATE-MAE have explicit structure and also contain semantic meaning. Code is available on [GitHub](GitHub).

## 1 INTRODUCTION

In recent years, deep learning has been called upon to process continually larger quantities of high-dimensional, noisy, and unlabeled data. A key property which makes these ever-larger tasks tractable is that the high-dimensional data tends to have *low-dimensional geometric and statistical structure*. Modern deep networks tend to learn (implicit or explicit) representations of this structure, which are then used to efficiently perform downstream tasks. Learning these representations is thus of central importance in machine learning, and there are so far several common methodologies for this task. We focus our attention below on approaches that incrementally transform the data towards the end representation with *simple, mathematically-interpretable primitives*. Discussion of popular alternatives is postponed to Appendix A.

**Denoising-diffusion models for high-dimensional data.** A popular method for learning implicit representations of high-dimensional data is *learning to denoise*: a model that can *denoise*, i.e., remove noise from a corrupted observation from the data distribution (to the extent information-theoretically possible), can be chained across noise levels to transform the data distribution to and from certain highly structured distributions, such as an isotropic Gaussian, enabling efficient sampling (Ho et al., 2020; Hyvärinen, 2005; Kadkhodaie & Simoncelli, 2021; Sohl-Dickstein et al., 2015; Song et al., 2021; 2023; Vincent, 2011). Crucially, in the case of data with low-dimensional structure—including the highly nonlinear structure characteristic of natural images—these models can be learned efficiently (Chen et al., 2023; Moitra & Risteski, 2020; Oko et al., 2023), and as a result this framework has significant practical impact (Rombach et al., 2022). Despite this progress, these techniques have been largely limited to use in generative modeling; a key reason is the unstructured nature of the final 'noisy' state of the diffusion process, which makes it challenging to control and interpret the model's learned implicit representation of the data.

---

[*]Correspondence to: Druv Pai, `druvpai@berkeley.edu`.

**White-box models and structured representation learning.** In contrast, *white-box* models are designed to produce explicit and structured representations of the data distribution according to a desired parsimonious configuration, such as sparsity (Gregor & LeCun, 2010; Zhai et al., 2020) or (piecewise) linearity (Chan et al., 2022). Recent work (Chan et al., 2022; Yu et al., 2023a) has built white-box deep networks via unrolled optimization: namely, to obtain representations with a desired set of properties, one constructs an objective function which encourages these desiderata, then constructs a deep network where each layer is designed to iteratively optimize the objective. This builds deep networks as a chain of operators, representing well-understood optimization primitives, which iteratively transform the representations to the desired structure. For example, Yu et al. (2023a) uses an information-theoretic objective promoting lossy compression of the data towards a fixed statistical structure to build a transformer-like architecture named CRATE in the above manner. However, such-obtained deep networks have yet to be constructed for most unsupervised contexts. The fundamental difficulty here is that decoder networks must map from representations to data, and hence *invert* (in a distributional sense) the transformations made to the data distribution by the encoder. This renders the unrolled optimization approach used to construct white-box encoders such as CRATE infeasible for constructing decoders, and instead demands a fine-grained understanding of the operators that implement the encoder, and their (distributional) inverses.

**Our contributions.** To overcome this difficulty and extend the applicability of white-box models to unsupervised settings, we demonstrate in this work that these two paradigms have more in common than previously appreciated. First, we show quantitatively that under certain natural regimes, *denoising* and *compression* are highly similar primitive data processing operations: when the target distribution has low-dimensional structure, both operations implement a projection operation onto this structure. Second, using this insight, we demonstrate a quantitative connection between unrolled discretized diffusion models and unrolled optimization-constructed deep networks. This leads to a significant expansion of the existing conceptual toolkit for developing white-box neural network architectures, which we use to derive white-box transformer-like encoder and decoder architectures that together form an autoencoding model that we call CRATE-MAE, illustrated in Fig.1. We evaluate CRATE-MAE on the challenging masked autoencoding task (He et al., 2022) and demonstrate promising performance with large parameter savings over traditional masked autoencoders, along with many side benefits such as emergence of semantic meaning in the representations.

## 2 APPROACH

### 2.1 SETUP AND NOTATION

We use the same notation and basic problem setup as in Yu et al. (2023a). Namely, we have some matrix-valued random variable $\boldsymbol{X} = [\boldsymbol{x}_1, \ldots, \boldsymbol{x}_N] \in \mathbb{R}^{D \times N}$ representing the data, where the $\boldsymbol{x}_i \in \mathbb{R}^D$ are called "tokens" and may be arbitrarily correlated. To obtain representations of the input, we learn an *encoder* $f \colon \mathbb{R}^{D \times N} \to \mathbb{R}^{d \times N}$; our representations are denoted by the random variable $\boldsymbol{Z} = f(\boldsymbol{X}) = [\boldsymbol{z}_1, \ldots, \boldsymbol{z}_N] \in \mathbb{R}^{d \times N}$, where the token representations are $\boldsymbol{z}_i \in \mathbb{R}^d$. In the autoencoding setup, we also learn a *decoder* $g \colon \mathbb{R}^{d \times N} \to \mathbb{R}^{D \times N}$, such that $\boldsymbol{X} \approx \widehat{\boldsymbol{X}} = [\widehat{\boldsymbol{x}}_1, \ldots, \widehat{\boldsymbol{x}}_N] \doteq g(\boldsymbol{Z})$.

Our encoder and decoder will be deep neural networks, and as such they will be composed of several, say $L$, *layers* each. Write $f = f^L \circ \cdots \circ f^1 \circ f^{\mathrm{pre}}$ and $g = g^{\mathrm{post}} \circ g^{L-1} \circ \cdots \circ g^0$, where $f^\ell \colon \mathbb{R}^{d \times N} \to \mathbb{R}^{d \times N}$ and $g^\ell \colon \mathbb{R}^{d \times N} \to \mathbb{R}^{d \times N}$ are the $\ell^{\mathrm{th}}$ layer of the encoder and decoder respectively, and $f^{\mathrm{pre}} \colon \mathbb{R}^{D \times N} \to \mathbb{R}^{d \times N}$ and $g^{\mathrm{post}} \colon \mathbb{R}^{d \times N} \to \mathbb{R}^{D \times N}$ are the pre- and post-processing layers respectively. The *input* to the $\ell^{\mathrm{th}}$ layer of the encoder is denoted $\boldsymbol{Z}^\ell \doteq [\boldsymbol{z}_1^\ell, \ldots, \boldsymbol{z}_N^\ell] \in \mathbb{R}^{d \times N}$, and the *input* to the $\ell^{\mathrm{th}}$ layer of the decoder is denoted $\boldsymbol{Y}^\ell \doteq [\boldsymbol{y}_1^\ell, \ldots, \boldsymbol{y}_N^\ell] \in \mathbb{R}^{d \times N}$.

### 2.2 DESIDERATA, OBJECTIVE, AND OPTIMIZATION

Our goal is to use the encoder $f$ and decoder $g$ to learn *representations* $\boldsymbol{Z}$ which are *parsimonious* (Ma et al., 2022) and *invertible*; namely, they have *low-dimensional, sparse, (piecewise) linear* geometric and statistical structure, and are (approximately) bijective with the original data $\boldsymbol{X}$. Yu et al. (2023a) proposes to implement these desiderata by positing a *signal model* for the representations:

**Low-Dimensional Gaussian Mixture Codebook.** *Let $\boldsymbol{Z} = [\boldsymbol{z}_1, \ldots, \boldsymbol{z}_N] \in \mathbb{R}^{d \times N}$ be a random matrix. We impose the following statistical model on $\boldsymbol{Z}$, parameterized by orthonormal bases*

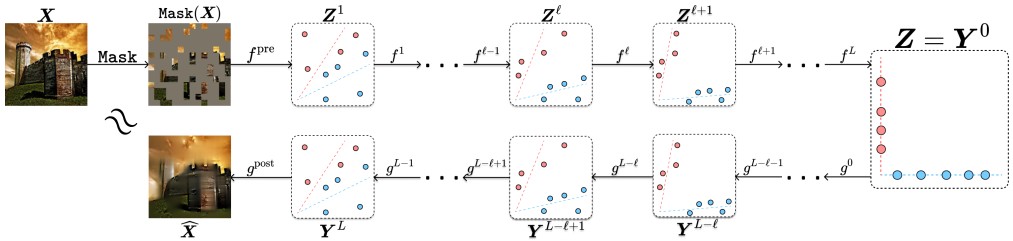

Figure 1: **Diagram of the overall white-box CRATE-MAE pipeline, illustrating the end-to-end (masked) autoencoding process.** The token representations are transformed iteratively towards a parsimonious (e.g., compressed and sparse) representation by each encoder layer $f^\ell$. Furthermore, such representations are transformed back to the original image by the decoder layers $g^\ell$. Each encoder layer $f^\ell$ is meant to be (partially) inverted by a corresponding decoder layer $g^{L-\ell}$.

$\boldsymbol{U}_{[K]} = (\boldsymbol{U}_k)_{k \in [K]} \in (\mathbb{R}^{d \times p})^K$: *each token $\boldsymbol{z}_i$ has marginal distribution given by*

$$\boldsymbol{z}_i \stackrel{d}{=} \boldsymbol{U}_{s_i} \boldsymbol{\alpha}_i, \quad \forall i \in [N] \tag{2.1}$$

*where $(s_i)_{i \in [N]} \in [K]^N$ are random variables corresponding to the subspace indices, and $(\boldsymbol{\alpha}_i)_{i \in [N]} \in (\mathbb{R}^p)^N$ are zero-mean Gaussian variables. If we optionally specify a noise parameter $\sigma \geq 0$, we mean that we "diffuse" the tokens with Gaussian noise: by an abuse of notation, each token $\boldsymbol{z}_i$ has marginal distribution given by*

$$\boldsymbol{z}_i \stackrel{d}{=} \boldsymbol{U}_{s_i} \boldsymbol{\alpha}_i + \sigma \boldsymbol{w}_i, \quad \forall i \in [N] \tag{2.2}$$

*where $(\boldsymbol{w}_i)_{i \in [N]} \in (\mathbb{R}^d)^N$ are i.i.d. standard Gaussian variables, independent of $s_i$ and $\boldsymbol{\alpha}_i$.*

If the $\boldsymbol{U}_k$ are sufficiently incoherent and axis-aligned, we expect such representations to maximize the *sparse rate reduction* objective function (Yu et al., 2023a):

$$\mathbb{E}_{\boldsymbol{Z}}[\Delta R(\boldsymbol{Z} \mid \boldsymbol{U}_{[K]}) - \lambda \|\boldsymbol{Z}\|_0] = \mathbb{E}_{\boldsymbol{Z}}[R(\boldsymbol{Z}) - R^c(\boldsymbol{Z} \mid \boldsymbol{U}_{[K]}) - \lambda \|\boldsymbol{Z}\|_0], \tag{2.3}$$

where $R$ and $R^c$ are *lossy coding rates*, or *rate distortions* (Cover, 1999), which are estimates for the number of bits required to encode the sample up to precision $\epsilon > 0$ using a Gaussian codebook, both unconditionally (for $R$), and conditioned on the samples being drawn from $\boldsymbol{U}_k$ summed over all $k$ (for $R^c$). Closed-form estimates (Ma et al., 2007; Yu et al., 2023a) for such rate distortions are:

$$R(\boldsymbol{Z}) = \frac{1}{2} \log \det(\boldsymbol{I}_N + \alpha \boldsymbol{Z}^\top \boldsymbol{Z}), \qquad \alpha \doteq \frac{d}{N\epsilon^2} \tag{2.4}$$

$$R^c(\boldsymbol{Z} \mid \boldsymbol{U}_{[K]}) = \frac{1}{2} \sum_{k=1}^{K} \log \det(\boldsymbol{I}_N + \beta(\boldsymbol{U}_k^\top \boldsymbol{Z})^\top(\boldsymbol{U}_k^\top \boldsymbol{Z})), \qquad \beta \doteq \frac{p}{N\epsilon^2}. \tag{2.5}$$

Notably, $R^c$ is a measure of *compression against our statistical structure* — it measures how closely the overall distribution of tokens in $\boldsymbol{Z}$ fit a Gaussian mixture on $\boldsymbol{U}_{[K]}$. Meanwhile, the other two terms $R$ and $\| \cdot \|_0$ ensure non-collapse and sparsity of the representations, respectively.

Following Yu et al. (2023a), one then constructs a deep network that *incrementally optimizes the sparse rate reduction* in order to transform the data distribution towards the desired parsimonious configuration (2.1). Specifically, Yu et al. (2023a) proposed to construct the deep neural network $f$ as a two-step alternating optimization procedure which compresses the input against the (learned) local signal model $\boldsymbol{U}_{[K]}^\ell$ at layer $\ell$, by taking a step of gradient descent on $R^c(\boldsymbol{Z} \mid \boldsymbol{U}_{[K]}^\ell)$, and subsequently taking a step of proximal gradient descent on a LASSO objective (Tibshirani, 1996; Wright & Ma, 2022) to sparsify the data in a (learned) dictionary $\boldsymbol{D}^\ell \in \mathbb{R}^{d \times d}$:

$$\boldsymbol{Z}^{\ell+1/2} = \boldsymbol{Z}^\ell + \texttt{MSSA}(\boldsymbol{Z}^\ell \mid \boldsymbol{U}_{[K]}^\ell) \approx \boldsymbol{Z}^\ell - \kappa \nabla_{\boldsymbol{Z}} R^c(\boldsymbol{Z}^\ell \mid \boldsymbol{U}_{[K]}^\ell) \tag{2.6}$$

$$\boldsymbol{Z}^{\ell+1} = \texttt{ISTA}(\boldsymbol{Z}^{\ell+1/2} \mid \boldsymbol{D}^\ell) \approx \underset{\boldsymbol{Z} \geq \boldsymbol{0}}{\arg\min} \left[ \frac{1}{2} \|\boldsymbol{Z}^{\ell+1/2} - \boldsymbol{D}^\ell \boldsymbol{Z}\|_2^2 + \lambda \|\boldsymbol{Z}\|_1 \right], \tag{2.7}$$

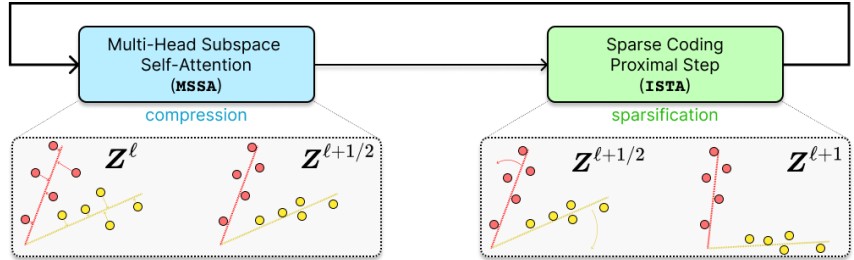

Figure 2: **The compression-sparsification iteration implemented by each layer of CRATE, and each encoder layer of CRATE-MAE.** The compression step, implemented by the MSSA operator, projects the tokens $\boldsymbol{Z}^\ell$ towards the subspace model $\boldsymbol{U}^\ell_{[K]}$ to form $\boldsymbol{Z}^{\ell+1/2}$. The sparsification step, implemented by the ISTA operator, rotates the tokens in $\boldsymbol{Z}^{\ell+1/2}$ towards the coordinate axes, using the sparsifying dictionary $\boldsymbol{D}^\ell$, to get $\boldsymbol{Z}^{\ell+1}$. The steps are performed in sequence and comprise a single of the CRATE-MAE encoder.

where $\mathrm{MSSA}(\cdot)$, the **M**ulti-head **S**ubspace **S**elf-**A**ttention block (Yu et al., 2023a), is defined as

$$\mathrm{MSSA}(\boldsymbol{Z} \mid \boldsymbol{U}_{[K]}) \doteq \frac{p}{N\epsilon^2} \begin{bmatrix} \boldsymbol{U}_1 & \cdots & \boldsymbol{U}_K \end{bmatrix} \begin{bmatrix} (\boldsymbol{U}_1^\top \boldsymbol{Z})\,\mathrm{softmax}((\boldsymbol{U}_1^\top \boldsymbol{Z})^\top (\boldsymbol{U}_1^\top \boldsymbol{Z})) \\ \vdots \\ (\boldsymbol{U}_K^\top \boldsymbol{Z})\,\mathrm{softmax}((\boldsymbol{U}_K^\top \boldsymbol{Z})^\top (\boldsymbol{U}_K^\top \boldsymbol{Z})) \end{bmatrix}, \quad (2.8)$$

and $\mathrm{ISTA}(\cdot)$, the **I**terative **S**hrinkage-**T**hresholding **A**lgorithm block (Yu et al., 2023a), is defined as

$$\mathrm{ISTA}(\boldsymbol{Z} \mid \boldsymbol{D}) \doteq \mathrm{ReLU}(\boldsymbol{Z} - \eta\boldsymbol{D}^\top(\boldsymbol{D}\boldsymbol{Z} - \boldsymbol{Z}) - \eta\lambda\mathbf{1}). \quad (2.9)$$

The MSSA block is exactly the same as a multi-head self-attention block in a transformer, with the changes that the $\boldsymbol{Q}_k/\boldsymbol{K}_k/\boldsymbol{V}_k$ blocks are replaced by a single matrix $\boldsymbol{U}_k$ in each head $k$. The resulting layer $f^\ell$ thus bears significant resemblance to a transformer-like block, and so the CRATE model is a white-box transformer-like architecture constructed via unrolled optimization. Such CRATE models obtain competitive performance on standard tasks while enjoying many side benefits (Yu et al., 2023a;b), yet they have so far only been trained for supervised learning. In the sequel, we introduce a paradigm to obtain fully white-box networks for unsupervised learning, such as autoencoding, through a novel understanding of the CRATE model's distributional layerwise inverse.

## 2.3 UNIFYING COMPRESSION AND DENOISING

To transform our representations to the idealized signal model given by (2.1), we seek to iteratively remove the disturbances or deviations of each sample from this signal model. One way to perform this task is to perform *lossy data compression* (Ma et al., 2007; Psenka et al., 2023; Yu et al., 2020; 2023a): compressed versions of the data, without ancillary disturbances, form the representations. This approach has been favored in the construction of previous white-box deep networks, such as CRATE described above, due to the existence of explicit information-theoretic criteria for compression. In this case, the term $R^c(\boldsymbol{Z} \mid \boldsymbol{U}_{[K]})$, defined in (2.5), measures the lossy compression of the representations $\boldsymbol{Z}$ against the class of statistical models given by (2.1). Thus, an operation to minimize $R^c$, such as (2.6), implements a step of compression to learn better representations.

Another way to remove disturbances from the signal model (2.1), especially if the perturbed model has the noisy structure given in (2.2), is to *denoise*. When the data is highly structured or low-dimensional, one-step denoising becomes statistically and computationally difficult (Pedregosa, 2023). Hence the modern solution to this problem is via *denoising diffusion models*, which take many small denoising steps towards the data distribution at progressively decreasing noise levels (Ho et al., 2020; Karras et al., 2022; Song et al., 2021). Such models use estimates of the so-called *score function* $\nabla \log p_\sigma$ (Hyvärinen, 2005), where $p_\sigma$ is the probability density function of the noised input when the noise has standard deviation $\sigma > 0$. At all sufficiently small values of $\sigma$, the score function $\nabla \log p_\sigma(\widetilde{\boldsymbol{Z}})$ for a particular noised input $\widetilde{\boldsymbol{Z}}$ points towards the closest point to $\widetilde{\boldsymbol{Z}}$ on the data distribution support (Chen et al., 2023; Lu et al., 2023; Yu et al., 2023a), or more generally the modes of the true data distribution, which guides the denoising diffusion model to project $\boldsymbol{Z}$ onto the support of the data distribution and diffuse it within this support.[1]

---

[1]A more mathematical exposition of diffusion models may be found in Appendix A.1.

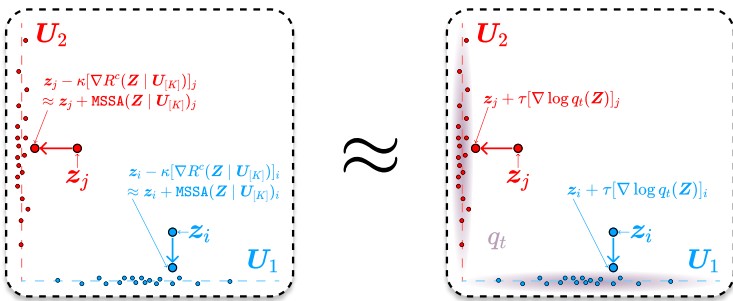

Figure 3: **Compression and denoising against the low-dimensional Gaussian mixture token model (2.1) are equivalent.** *Left:* the effect of compression against the low-dimensional Gaussian mixture model for tokens (2.1), i.e., taking gradient steps on the coding rate $R^c(\cdot \mid \boldsymbol{U}_{[K]})$ — or equivalently, using the MSSA$(\cdot \mid \boldsymbol{U}_{[K]})$ operator — which is shown in Theorem 1 to be equivalent to projecting onto the $\boldsymbol{U}_{[K]}$. *Right:* the effect of denoising against (2.1), i.e., taking gradient steps on the score function of the noisy model (2.2) at small noise levels $\sigma$, or equivalently small times $t$. Up to scaling factors (not pictured), these two operations are equivalent, and have similar geometric and statistical interpretations as a projection onto the support of the data distribution. This connection motivates our structured denoising-diffusion framework, as elaborated in Section 2.3.

In the context of (2.1) and (2.2), both denoising and compression operations conceptually remove additive disturbances from the data, as visualized in Figure 3. In the following result, we make this qualitative observation mathematically precise: we show that under a simplified version of the signal model (2.1), taking a gradient step on $R^c$, a compression primitive, acts as a projection onto the local signal model $\boldsymbol{U}_{[K]}$, just as with the denoising primitive of taking a gradient step on $\log p_\sigma$.

**Theorem 1** (Informal version of Theorem 3 in Appendix A.2). *Suppose $\boldsymbol{Z}$ follows the noisy Gaussian codebook model (2.2), with infinitesimal noise level $\sigma^\ell > 0$ and subspace memberships $s_i$ distributed as i.i.d. categorical random variables on the set of subspace indices $\{1, \ldots, K\}$, independently of all other sources of randomness. Suppose in addition that the number of tokens $N$, the representation dimension $d$, the number of subspaces $K$, and the subspace dimensions $p$ have relative sizes matching those of practical transformer architectures including the CRATE-MAE encoder (specified in detail in Assumption 2). Then the negative compression gradient $-\nabla_{\boldsymbol{z}_i} R^c(\boldsymbol{Z}^\ell \mid \boldsymbol{U}^\ell_{[K]})$ points from $\boldsymbol{z}^\ell_i$ to the nearest $\boldsymbol{U}^\ell_k$.*

Theorem 1 establishes in a representative special case of the Gaussian codebook model (2.1) that at low noise levels, *compression against the signal model* (2.1) *is equivalent to denoising against* (2.1). In the sequel, we use this connection to understand the MSSA operators of the CRATE-MAE encoder, derived in Section 2.2 from a different perspective, as realizing an incremental transformation of the data distribution towards the signal model (2.1) via approximate denoising. This important property guarantees that a corresponding deterministic diffusion process—namely, the time reversal of the denoising process—implies an inverse operator for the compression operation implemented by MSSA. Because these approximate denoising processes transform the data towards a parametric structure, we call them *structured denoising-diffusion processes*.

## 2.4 CONSTRUCTING A DISTRIBUTIONALLY-INVERTIBLE TRANSFORMER LAYER

In Section 2.1, we described a method to construct a white-box transformer-like encoder network via unrolled optimization meant to compress the data against learned geometric and statistical structures, say against a distribution of tokens where each token is marginally distributed as a Gaussian mixture supported on $\boldsymbol{U}_{[K]}$. In Section 2.3, we described in general terms an approach that relates denoising and compression to yield a conceptually similar network using the formalism of diffusion models, this time trainable via autoencoding. In this section, we carry out this procedure concretely to obtain an encoder and decoder layer with similarly interpretable operational characteristics.

To measure compression, we use the $R^c$ function defined in (2.5). By using a standard (reverse-time) diffusion process with a scaling of $R^c$ as a drop-in replacement for the score (see Appendix A.3 for details), we obtain that such a denoising diffusion process may be described by the following

stochastic differential equation (SDE) (Song et al., 2021).

$$\mathrm{d}\boldsymbol{Z}(t) = -\frac{1}{T-t}\nabla R^c(\boldsymbol{Z}(t) \mid \boldsymbol{U}_{[K]})\,\mathrm{d}t + \sqrt{2}\,\mathrm{d}\boldsymbol{B}(t), \qquad \forall t \in [0, T], \tag{2.10}$$

where $(\boldsymbol{B}(t))_{t\in[0,T]}$ is a Brownian motion. *As a design choice*, we wish to assert that our encoder and decoder ought to be deterministic, in particular preferring that our encoder-decoder architecture achieves *sample-wise autoencoding* as opposed to *distribution-wise autoencoding* or *generation*. Thus we need to construct some ordinary differential equation (ODE) which transports the input probability distribution in the same way as (2.10). Such an equation is readily obtained as the *probability flow ODE* (Song et al., 2021), which itself is commonly used for denoising and sampling (Lu et al., 2022; Song et al., 2021; 2023) and has the form

$$\mathrm{d}\boldsymbol{Z}(t) = -\frac{1}{2(T-t)}\nabla R^c(\boldsymbol{Z}(t) \mid \boldsymbol{U}_{[K]})\,\mathrm{d}t, \qquad \forall t \in [0, T]. \tag{2.11}$$

In particular, the $\boldsymbol{Z}(t)$ generated by (2.10) and (2.11) have the same law. A first-order discretization (see Appendix A.3) of (2.11) with step size $\kappa$ obtains the iteration:

$$\boldsymbol{Z}^{\ell+1/2} = \boldsymbol{Z}^\ell + \mathtt{MSSA}(\boldsymbol{Z}^\ell \mid \boldsymbol{U}_{[K]}^\ell) \approx \boldsymbol{Z}^\ell - \kappa\nabla R^c(\boldsymbol{Z}^\ell \mid \boldsymbol{U}_{[K]}^\ell), \tag{2.12}$$

where $\mathtt{MSSA}(\cdot)$ was defined in (2.8). Similar to Yu et al. (2023a), in order to optimize the sparse rate reduction of the features, and in particular to sparsify them, we instantiate a learnable dictionary $\boldsymbol{D}^\ell \in \mathbb{R}^{d\times d}$ and sparsify against it, obtaining

$$\boldsymbol{Z}^{\ell+1} = \mathtt{ISTA}(\boldsymbol{Z}^{\ell+1/2} \mid \boldsymbol{D}^\ell), \tag{2.13}$$

where $\mathtt{ISTA}(\cdot)$ was defined in (2.9). Thus, we obtain a two step iteration for the $\ell^{\text{th}}$ encoder layer $f^\ell$, where $\boldsymbol{Z}^{\ell+1} = f^\ell(\boldsymbol{Z}^\ell)$:

$$\boldsymbol{Z}^{\ell+1/2} = \boldsymbol{Z}^\ell + \mathtt{MSSA}(\boldsymbol{Z}^\ell \mid \boldsymbol{U}_{[K]}^\ell), \qquad \boldsymbol{Z}^{\ell+1} = \mathtt{ISTA}(\boldsymbol{Z}^{\ell+1/2} \mid \boldsymbol{D}^\ell). \tag{2.14}$$

This is the same layer as in CRATE, whose conceptual behavior is illustrated in Figure 2. This equivalence stems from the fact that the diffusion probability flow (2.11) is conceptually and mechanically similar to gradient flow on the compression objective in certain regimes, and so it demonstrates a useful conceptual connection between discretized diffusion and unrolled optimization as iteratively compressing or denoising the signal against the learned data structures.

Note that we parameterized a *different* local signal model $\boldsymbol{U}_{[K]}^\ell$ and dictionary $\boldsymbol{D}^\ell$ at each layer, despite the continuous-time flows in (2.11) using only one (i.e., the final) local signal model. This is because the sparsification step (2.13) transforms the data distribution, and so we require a different local signal model at each layer to model the new (more sparse) data distribution; see Figure 1 for intuition on the iterative transformations. Also, having a different signal model at each layer may allow for more efficient iterative linearization and compression of highly nonlinear structures.

Now that we have shown how the structured diffusion approach can recover the original CRATE architecture (Yu et al., 2023a) as an encoder in our autoencoding problem, we use our new approach to construct a novel matching decoder. The time reversal of the ODE (2.11) is:

$$\mathrm{d}\boldsymbol{Y}(t) = \frac{1}{2t}\nabla R^c(\boldsymbol{Y}(t) \mid \boldsymbol{U}_{[K]})\,\mathrm{d}t, \qquad \forall t \in [0, T], \tag{2.15}$$

in the sense that the $\boldsymbol{Y}(T-t)$ generated by (2.15) has the same law as the $\boldsymbol{Z}(t)$ generated by (2.11), assuming compatible initial conditions. A first-order discretization of (2.15) obtains the iteration:

$$\boldsymbol{Y}^{\ell+1} = \boldsymbol{Y}^{\ell+1/2} - \mathtt{MSSA}(\boldsymbol{Y}^{\ell+1/2} \mid \boldsymbol{V}_{[K]}^\ell) \approx \boldsymbol{Y}^{\ell+1/2} + \kappa\nabla R^c(\boldsymbol{Y}^{\ell+1/2} \mid \boldsymbol{V}_{[K]}^\ell), \tag{2.16}$$

where $\boldsymbol{V}_{[K]}^\ell = (\boldsymbol{V}_1^\ell, \dots, \boldsymbol{V}_K^\ell)$ and each $\boldsymbol{V}_k^\ell \in \mathbb{R}^{d\times p}$ are the bases of the subspaces to "anti-compress" against. In our work, we treat them as *different* from the corresponding $\boldsymbol{U}_k^{L-\ell}$, because the discretization of (2.11) and (2.15) is imperfect, and thus we should not expect a 1-1 correspondence between local signal models in the encoder and decoder. To invert the effect of a sparsifying ISTA step, which our mental model in Figure 2 portrays as a rotation of the subspace supports to a more incoherent configuration, we multiply by another learnable dictionary $\boldsymbol{E}^\ell \in \mathbb{R}^{d\times d}$, obtaining

$$\boldsymbol{Y}^{\ell+1/2} = \boldsymbol{E}^\ell\boldsymbol{Y}^\ell, \qquad \boldsymbol{Y}^{\ell+1} = \boldsymbol{Y}^{\ell+1/2} - \mathtt{MSSA}(\boldsymbol{Y}^{\ell+1/2} \mid \boldsymbol{V}_{[K]}^\ell). \tag{2.17}$$

This constructs the $(\ell+1)^{\text{st}}$ layer $g^\ell$ of our decoder. In the implementation, we add layer normalizations to ensure that the features are roughly constant-size so that the above approximations hold. Figure 4 has a graphical depiction of the encoder and decoder layers.

Figure 4: **Diagram of each encoder layer (*top*) and decoder layer (*bottom*) in CRATE-MAE.** Notice that the two layers are highly anti-parallel — each is constructed to do the operations of the other in reverse order. That is, in the decoder layer $g^\ell$, the ISTA block of $f^{L-\ell}$ is partially inverted first using a linear layer, then the MSSA block of $f^{L-\ell}$ is reversed; this order unravels the transformation done in $f^{L-\ell}$.

## 2.5 A COMPLETE WHITE-BOX TRANSFORMER-LIKE ARCHITECTURE FOR AUTOENCODING

As previously discussed, the encoder is the concatenation of a preprocessing map $f^{\mathrm{pre}} \colon \mathbb{R}^{D \times N} \to \mathbb{R}^{d \times N}$, which has learnable parameters $\boldsymbol{W}^{\mathrm{pre}} \in \mathbb{R}^{d \times D}$ and $\boldsymbol{E}^{\mathrm{pos}} \in \mathbb{R}^{d \times N}$, and has the form:

$$f^{\mathrm{pre}}(\boldsymbol{X}) \doteq \boldsymbol{W}^{\mathrm{pre}}\boldsymbol{X} + \boldsymbol{E}^{\mathrm{pos}}, \tag{2.18}$$

and $L$ transformer-like layers $f^\ell \colon \mathbb{R}^{d \times N} \to \mathbb{R}^{d \times N}$ given by

$$f^\ell(\boldsymbol{Z}^\ell) \doteq \texttt{ISTA}(\boldsymbol{Z}^\ell + \texttt{MSSA}(\boldsymbol{Z}^\ell \mid \boldsymbol{U}^\ell_{[K]}) \mid \boldsymbol{D}^\ell), \qquad \forall \ell \in [L], \tag{2.19}$$

omitting normalization for simplicity. The decoder is the concatenation of $L$ transformer-like layers $g^\ell \colon \mathbb{R}^{d \times N} \to \mathbb{R}^{d \times N}$ given by

$$g^\ell(\boldsymbol{Y}^\ell) \doteq \boldsymbol{E}^\ell \boldsymbol{Y}^\ell - \texttt{MSSA}(\boldsymbol{E}^\ell \boldsymbol{Y}^\ell \mid \boldsymbol{V}^\ell_{[K]}), \qquad \forall \ell \in [L] - 1, \tag{2.20}$$

with a postprocessing map $g^{\mathrm{post}} \colon \mathbb{R}^{d \times N} \to \mathbb{R}^{D \times N}$ which is a learnable linear map $\boldsymbol{W}^{\mathrm{post}} \in \mathbb{R}^{D \times d}$:

$$g^{\mathrm{post}}(\boldsymbol{Y}^L) \doteq \boldsymbol{W}^{\mathrm{post}}\boldsymbol{Y}^L. \tag{2.21}$$

A full diagram of the autoencoding procedure is given in Figure 1.

Our training procedure seeks to learn and represent the structures in the data distribution. For this, we use a pretext task that measures the degree to which these structures have been learned: *masked autoencoding* (He et al., 2022), which "masks out" a large percentage of randomly selected tokens in the input $\boldsymbol{X}$ and then attempts to reconstruct the whole image, measuring success by the resulting autoencoding performance. Conceptually, masked autoencoding can be seen as a nonlinear generalization of the classical *matrix completion* task, which exploits low-dimensional structure to impute missing entries in incomplete data; classical matrix completion can be solved efficiently if and only if the data have low-dimensional structure (Amelunxen et al., 2014; Candès & Recht, 2009; Wright & Ma, 2022). The masked autoencoding loss writes

$$L_{\mathrm{MAE}}(f, g) \doteq \mathbb{E}\big[\|(g \circ f)(\texttt{Mask}(\boldsymbol{X})) - \boldsymbol{X}\|_2^2\big]. \tag{2.22}$$

Further implementation details of this architecture are discussed in Appendices B.1 and B.2.

## 3 EMPIRICAL EVALUATIONS

In this section, we conduct experiments to evaluate CRATE-MAE on real-world datasets and both supervised and unsupervised tasks. Similarly to Yu et al. (2023a), CRATE-MAE is built using simple design choices that we do not claim are optimal. We also do not claim that our results are optimally engineered; in particular, *we do not use the extreme amount of computational resources required to obtain state-of-the-art performance using vision transformer-backed masked autoencoders (MAEs) (He et al., 2022)*. Our goals in this section are to verify that our white-box masked autoencoding model CRATE-MAE has promising performance and learns semantically meaningful representations, and that each operator in CRATE-MAE aligns with our theoretical design. We provide additional experimental details in Appendices B.1 and B.2.

**Network architecture and training configuration.** We implement the encoder and decoder architectures described in Section 2, with a few changes detailed in Appendix B.1. We consider different model sizes of CRATE-MAE by varying the token dimension $d$, number of heads $K$, and number of

Table 1: **Model configurations for different sizes of CRATE-MAE, parameter counts, and comparisons to ViT-MAE models from Gandelsman et al. (2022); He et al. (2022).** We observe that CRATE-MAE-Base uses around 30% of the parameters of ViT-MAE-Base, and a similar number of parameters as ViT-MAE-Small.

| Model Configuration | $L$ | $d$ | $K$ | $N$ | CRATE-MAE # Parameters | ViT-MAE # Parameters |
|---|---|---|---|---|---|---|
| Small (-S) | 12 | 576 | 12 | 196 | 25.4M | 47.6M |
| Base (-B) | 12 | 768 | 12 | 196 | 44.6M | 143.8M |

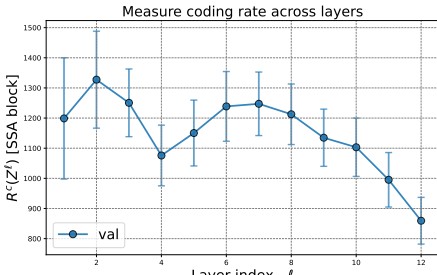
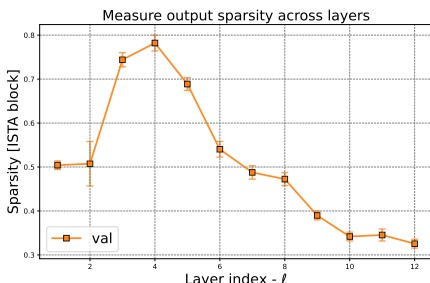

Figure 5: *Left*: **The compression measure** $R^c(\boldsymbol{Z}^{\ell+1/2} \mid \boldsymbol{U}_{[K]}^\ell)$ **at different layers of the encoder.** *Right*: **the sparsity measure** $\|\boldsymbol{Z}^{\ell+1}\|_0/(d \cdot N)$, **at different layers of the encoder.** Measurements were collected from CRATE-MAE-Base averaged over 10000 randomly chosen ImageNet samples. We observe that the compression and sparsity improve consistently over each layer and through the whole network.

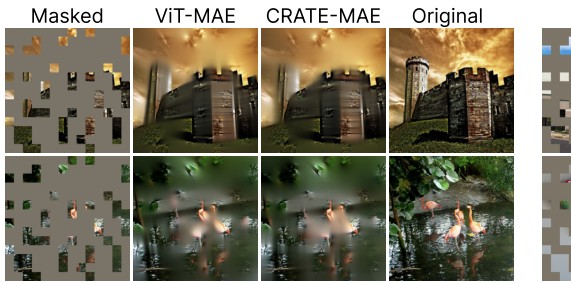
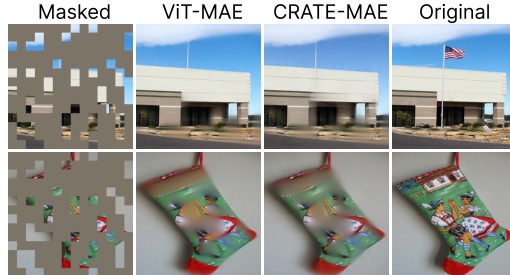

Figure 6: **Autoencoding visualizations of CRATE-MAE-Base and ViT-MAE-Base (He et al., 2022) with 75% patches masked.** We observe that the reconstructions from CRATE-MAE-Base are on par with the reconstructions from ViT-MAE-Base, despite using $< 1/3$ of the parameters.

layers $L$; such parameters will be kept the same for the encoder and decoder, which is contrary to He et al. (2022) but in line with our white-box derivation. Table 1 displays the CRATE-MAE model configurations and number of parameters, and compares with equivalent ViT-MAE model sizes (Gandelsman et al., 2022; He et al., 2022), showing that CRATE-MAE *uses around 30% of the parameters of MAE with the same model configuration*. We consider ImageNet-1K (Deng et al., 2009) as the main experimental setting for our architecture. We apply the AdamW (Loshchilov & Hutter, 2019) optimizer to train CRATE-MAE models for both pre-training and fine-tuning. When fine-tuning, we also use several commonly used downstream datasets: CIFAR10, CIFAR100 (Krizhevsky et al., 2009), Oxford Flowers (Nilsback & Zisserman, 2008), and Oxford-IIT-Pets (Parkhi et al., 2012).

**Layer-wise function analysis.** First, we confirm that our model actually does do layer-wise compression and sparsification, confirming our conceptual understanding as described in Section 2. In Figure 5, we observe that each layer of the encoder tends to compress and sparsify the input features, confirming our theoretical designing of the role of each operator in the network.

**Autoencoding performance.** In Figure 6, we qualitatively compare the masked autoencoding performance of CRATE-MAE-Base to ViT-MAE-Base (He et al., 2022). We observe that both models are able to reconstruct the data well, despite CRATE-MAE using less than a third of the parameters of ViT-MAE. In Table 4 (deferred to Appendix B.4) we display the average reconstruction loss of CRATE-MAE-Base and ViT-MAE-Base, showing a similar quantitative conclusion.

Table 2: **Top-1 classification accuracy of CRATE-MAE models when pre-trained on ImageNet-1K and evaluated via fine-tuning or linear probing for various datasets.** We compare CRATE-MAE to standard ViT-MAE models with many more parameters. Our results show that CRATE-MAE achieves competitive performance on this transfer learning task when either fine-tuning the whole model or just the classification head.

| Classification Accuracy | CRATE-MAE-S | CRATE-MAE-B | ViT-MAE-S | ViT-MAE-B |
|---|---|---|---|---|
| *Fine-Tuning* | | | | |
| CIFAR10 | 96.2 | 96.8 | 97.6 | 98.5 |
| CIFAR100 | 79.0 | 80.3 | 83.0 | 87.0 |
| Oxford Flowers-102 | 71.7 | 78.5 | 84.2 | 92.5 |
| Oxford-IIIT-Pets | 73.7 | 76.7 | 81.7 | 90.3 |
| *Linear Probing* | | | | |
| CIFAR10 | 79.4 | 80.9 | 79.9 | 87.9 |
| CIFAR100 | 56.6 | 60.1 | 62.3 | 68.0 |
| Oxford Flowers-102 | 57.7 | 61.8 | 66.8 | 66.4 |
| Oxford-IIIT-Pets | 40.6 | 46.2 | 51.8 | 80.1 |

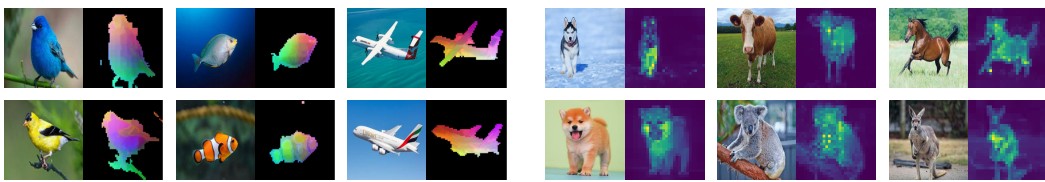

(a) Visualizing PCA of token representations.     (b) Visualizing selected attention head outputs.

Figure 7: *Left:* **Visualizations of the alignment of each image's token representations with the top three principal components (in red, blue, and green respectively) of all token representations of images in the given class.** *Right:* **Visualizations of hand-picked attention map across all attention heads in the last layer of the CRATE-MAE encoder for each image.** We observe in Figure 7a that the top three principal components are aligned with tokens from parts of the image that carry its semantics, and in Figure 7b that the attention maps correctly "attend to" (activate strongly on) exactly the parts of the image which are semantically meaningful.

**Representation learning and emerging semantic properties.** In Table 2, we display the performance of CRATE-MAE models when fine-tuned or linear probed for supervised classification (precise method in Appendix B.1) on a variety of datasets. We observe that the classification accuracies of CRATE-MAE models are competitive with much larger ViT-MAE models. Moreover, the learned representations of CRATE-MAE carry useful semantic content. By taking the alignment of the representations of each token with the top few principal components of the representations of tokens in each class (precise details in Appendix B.3), we observe in Figure 7 (left) that the representations are linearized, and that the top few principal components carry semantic structure. In Figure 7 (right), we observe that the attention heads in the MSSA operator in CRATE-MAE capture the semantics of the input images. These properties have previously been observed in white-box models trained with supervised cross-entropy losses (Yu et al., 2023b); our results demonstrate that they are consequences of the white-box architecture, rather than the loss function.

## 4  CONCLUSION

In this work, we uncover a quantitative connection between denoising and compression, and use it to design a conceptual framework for building white-box (mathematically interpretable) transformer-like deep neural networks which can learn using unsupervised pretext tasks, such as masked autoencoding. We show that such models are more parameter-efficient over their empirically designed cousins, achieve promising performance on large-scale real-world imagery datasets, and learn structured representations that contain semantic meaning. This work demonstrates the potential and practicality of white-box networks derived from first principles for tasks outside supervised classification. We thus believe that this work helps to bridge the theory and practice of deep learning, by unifying on both the conceptual and technical level many previously separated approaches including, but not limited to, diffusion and denoising, compression and rate reduction, transformers, and (masked) autoencoding.

ACKNOWLEDGMENTS

The authors would like to acknowledge help from Tianzhe Chu in preparing the manuscript. Druv Pai acknowledges support from a UC Berkeley College of Engineering fellowship. Yaodong Yu acknowledges support from the joint Simons Foundation-NSF DMS grant #2031899 and AI Community Mini Grant from Future of Life Institute. Yi Ma acknowledges support from the joint Simons Foundation-NSF DMS grant #2031899, the ONR grant N00014-22-1-2102, and the University of Hong Kong.

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

## A    OTHER RELATED WORK

In Section 1, we described the approaches to unsupervised learning of low-dimensional structures in the data that were most relevant to the rest of the work. Here, we discuss some other popular alternatives for completeness.

**Black-box unsupervised representation learning.**    On the other end from white-box models, which learn representations from data that have *a priori* desired structures, are black-box unsupervised learning methods which learn fully data-driven representations. One implementation of this principle includes contrastive learning, which learns representations from computing the statistics of multiple augmentations of the same data point (Bardes et al., 2022; Chen et al., 2020). Another angle is to seek a representation with desirable characteristics for a specific task, such as classification; prior works have considered diffusion models as "representation learners" from this angle (Chen et al., 2024; Xiang et al., 2023). The notion of representation learning we are interested in in this work, namely the transformation of the data distribution towards a structured form that preserves its essential information content, is different from the notion in this latter group of works. Still another implementation is that of *autoencoding* models, the most recently popular of which is the *masked autoencoder* (MAE) (He et al., 2022). Autoencoders attempt to build low-dimensional representations of the data and use them to reconstruct input data (Hinton & Zemel, 1993; Kingma & Welling, 2014; Rezende et al., 2014; Tishby & Zaslavsky, 2015); masked autoencoders specifically mask the input data in training and attempt to impute the missing entries through reconstruction.

The common point in all such unsupervised learning methods so far is that they use *black-box* neural networks, such as ResNets (Chen et al., 2020) or transformers (Caron et al., 2021), as their back-end. Thus, although they sometimes develop semantically meaningful representations of the data (Bardes et al., 2022; Caron et al., 2021; Chen et al., 2020), they are uninterpretable, and their training procedures and internal mechanisms are opaque.

**Deep networks and stochastic dynamics.**    There are many quantitative rapprochements of deep learning and stochastic dynamics. The most well-known of these is diffusion models, which can be modeled as discretizations of Itô diffusion processes (Song et al., 2021). The neural network is usually trained to estimate the so-called *score function*. Diffusion models can be thought of as implementing a particular approximation to optimal transport between a template distribution and the true data distribution (Khrulkov et al., 2023). Different types of stochastic dynamics useful for generative modeling may be derived from optimal transport between the data distribution and a pre-specified template (Albergo et al., 2023; De Bortoli et al., 2021). However, diffusion models are unique among these methods in that they have an *iterative denoising* interpretation (Karras et al., 2022), which this work draws on. Such an interpretation has previously been used to construct deep denoising networks from unrolled diffusion processes (Mei & Wu, 2023), instead of just using the deep networks to do black-box estimation of the score function. Similar studies have interpreted deep networks as discretizations of diffusion processes without this particular denoising interpretation (Li et al., 2022), but the aforementioned unrolled iterative denoising strategy is what we draw upon in this work.

**Other related work.**    Here we also discuss some related work with regards to different modifications of the transformer architecture and training procedures which interface well with our white-box design. For example, Kitaev et al. (2020) suggests that sharing the $Q$ and $K$ matrices in the regular transformer is a mechanism to make the transformer more efficient at no performance cost. This choice is a heuristic, whereas our white-box design suggests that $Q$, $K$, and $V$ should be set equal, and as we see in the paper this comes with some small tradeoffs. Also, since white-box models are derived such that each layer has a defined and understood role, it is natural to ask if such models can be trained layer-wise, i.e., one layer at a time (Bengio et al., 2006). While this is also possible, we leave it to future work; our experiments show that with end-to-end training, a vastly more common method to train deep networks, each layer still follows the role it was designed for.

### A.1    AN OVERVIEW OF DIFFUSION PROCESSES

In this section, we give an overview of the basics of time-reversible Itô diffusion processes, the mathematical foundation for diffusion models. This is to make this paper more self-contained by

providing knowledge about general diffusion processes that we will apply to our special models. The coverage adapts that of Karras et al. (2022); Millet et al. (1989); Song et al. (2021).

Consider a generic Itô diffusion process $(\boldsymbol{z}(t))_{t \in [0,T]}$, where $\boldsymbol{z}(t)$ is an $\mathbb{R}^m$-valued random variable, given by the SDE

$$\mathrm{d}\boldsymbol{z}(t) = b(\boldsymbol{z}(t), t)\, \mathrm{d}t + \Sigma(\boldsymbol{z}(t), t)\, \mathrm{d}\boldsymbol{w}(t), \qquad \boldsymbol{z}(0) \sim P, \qquad \forall t \in [0,T] \qquad \text{(A.1)}$$

where $\boldsymbol{w}$ is a Brownian motion and $P$ is some probability measure on $\mathbb{R}^m$ (in this case representing the data distribution). Here the *drift coefficient* $b \colon \mathbb{R}^m \times \mathbb{R} \to \mathbb{R}^m$ and *diffusion coefficient* $\Sigma \colon \mathbb{R}^m \times \mathbb{R} \to \mathbb{R}^{m \times m}$ are functions. To make sense of (A.1) and also verify the existence of strong (i.e., pathwise well-defined) solutions, we need some regularity on them, and we choose the following assumption:

A1.  $b$ and $\Sigma$ have some spatial smoothness and do not grow too fast, i.e., there is a constant $K \geq 0$ such that for all $\boldsymbol{x}, \widetilde{\boldsymbol{z}} \in \mathbb{R}^m$ we have

$$\sup_{t \in [0,T]} \left[ \|\Sigma(\boldsymbol{x}, t) - \Sigma(\widetilde{\boldsymbol{z}}, t)\|_F + \|b(\boldsymbol{x}, t) - b(\widetilde{\boldsymbol{z}}, t)\|_2 \right] \leq K \|\boldsymbol{x} - \widetilde{\boldsymbol{z}}\|_2 \qquad \text{(A.2)}$$

$$\sup_{t \in [0,T]} \left[ \|\Sigma(\boldsymbol{x}, t)\|_F + \|b(\boldsymbol{x}, t)\|_2 \right] \leq K(1 + \|\boldsymbol{x}\|_2). \qquad \text{(A.3)}$$

In general, $\boldsymbol{z}(t)$ may not have a density w.r.t. the Lebesgue measure on $\mathbb{R}^m$. For example, suppose that $P$ is supported on some low-dimensional linear subspace (or even a Dirac delta measure), and take $\Sigma$ to be the orthoprojector onto this subspace. Then $\boldsymbol{z}(t)$ will be supported on this subspace for all $t$ and thus not have a density w.r.t. the Lebesgue measure. Thus, when further discussing processes of the type (A.1), we make the following assumption

A2.  $\boldsymbol{z}(t)$ has a probability density function $p(\cdot, t)$ for all $t > 0$.

This is guaranteed by either of the following conditions (Millet et al., 1989):

A2.1  $b$ and $\Sigma$ are differentiable in $(\boldsymbol{x}, t)$ and have Hölder-continuous derivatives, and $P$ has a density w.r.t. the Lebesgue measure;

A2.2  The event

$$\{\operatorname{rank}(\Sigma(\boldsymbol{z}(s), s)) = m \text{ for all } s \text{ in some neighborhood of } 0\} \qquad \text{(A.4)}$$

happens $P$-almost surely.

Define $\Psi \colon \mathbb{R}^m \times \mathbb{R} \to \mathbb{R}^{m \times m}$ by

$$\Psi(\boldsymbol{x}, t) \doteq \Sigma(\boldsymbol{x}, t) \Sigma(\boldsymbol{x}, t)^\top. \qquad \text{(A.5)}$$

To discuss time-reversibility, we also need the following local integrability condition, which is another measure of sharp growth of the coefficients (or precisely their derivatives):

A3.  The functions $(\boldsymbol{x}, t) \mapsto \nabla_{\boldsymbol{x}} \cdot (\Psi(\boldsymbol{x}, t) p(\boldsymbol{x}, t))$ are integrable on sets of the form $D \times [t_0, 1]$ for $t_0 > 0$ and $D$ a bounded measurable subset of $\mathbb{R}^m$:

$$\int_{t_0}^1 \int_D \|\nabla_{\boldsymbol{x}} \cdot (\Psi(\boldsymbol{x}, t) p(\boldsymbol{x}, t))\|_2 \, \mathrm{d}\boldsymbol{x} \, \mathrm{d}t < \infty. \qquad \text{(A.6)}$$

To write the notation out more explicitly,

$$\nabla_{\boldsymbol{x}} \cdot (\Psi(\boldsymbol{x}, t) p(\boldsymbol{x}, t)) = \begin{bmatrix} \nabla_{\boldsymbol{x}} \cdot (\Psi^1(\boldsymbol{x}, t) p(\boldsymbol{x}, t)) \\ \vdots \\ \nabla_{\boldsymbol{x}} \cdot (\Psi^m(\boldsymbol{x}, t) p(\boldsymbol{x}, t)) \end{bmatrix} \qquad \text{(A.7)}$$

$$\text{where} \qquad \nabla_{\boldsymbol{x}} \cdot (\Psi^i(\boldsymbol{x}, t) p(\boldsymbol{x}, t)) = \sum_{j=1}^m \frac{\partial}{\partial x_j} [\Psi^{ij}(\boldsymbol{x}, t) p(\boldsymbol{x}, t)] \qquad \text{(A.8)}$$

where $\Psi^i$ is the $i^{\text{th}}$ row of $\Psi$ transposed to a column, and $\Psi^{ij}$ is the $(i, j)^{\text{th}}$ entry of $\Psi$. Note that Millet et al. (1989) phrases this in terms of an local integrability condition on each

$|\nabla_{\boldsymbol{x}} \cdot (\Psi^i(\boldsymbol{x},t)p(\boldsymbol{x},t))|$, which would naturally give a local integrability condition on $\|\nabla_{\boldsymbol{x}} \cdot (\Psi(\boldsymbol{x},t)p(\boldsymbol{x},t))\|_\infty$. However, all norms on $\mathbb{R}^m$ are equivalent, and so this leads to a local integrability condition for $\|\nabla_x \cdot (\Psi(\boldsymbol{x},t)p(\boldsymbol{x},t))\|_2$ as produced. Note that the assumptions do not guarantee that the involved derivatives exist, in which case they are taken in the distributional (e.g., weak) sense, whence they should exist (Millet et al., 1989).

Under assumptions A1—A3, Millet et al. (1989) guarantees the existence of another process $(\widetilde{\boldsymbol{z}}(t))_{t \in [0,T]}$ such that the laws of $\boldsymbol{z}(t)$ and $\widetilde{\boldsymbol{z}}(T-t)$ are the same for all $t \in [0,T]$. This process $(\widetilde{\boldsymbol{z}}(t))_{t \in [0,T]}$ is called the *time reversal* of $(\boldsymbol{z}(t))_{t \in [0,T]}$, and is shown to have law given by

$$d\widetilde{\boldsymbol{z}}(t) = b^\leftarrow(\widetilde{\boldsymbol{z}}(t),t)\, dt + \Sigma^\leftarrow(\widetilde{\boldsymbol{z}}(t),t)\, d\boldsymbol{w}^\leftarrow(t), \qquad \widetilde{\boldsymbol{z}}(0) \sim p(\cdot,T), \qquad \forall t \in [0,T] \quad \text{(A.9)}$$

where $\boldsymbol{w}^\leftarrow(t)$ is an independent Brownian motion and

$$b^\leftarrow(\boldsymbol{x},t) = -b(\boldsymbol{x},T-t) + \frac{\nabla_{\boldsymbol{x}} \cdot [\Psi(\boldsymbol{x},T-t)p(\boldsymbol{x},T-t)]}{p(\boldsymbol{x},T-t)} \quad \text{(A.10)}$$

$$= -b(\boldsymbol{x},T-t) + \nabla_{\boldsymbol{x}} \cdot \Psi(\boldsymbol{x},T-t) + \Psi(\boldsymbol{x},T-t)[\nabla_{\boldsymbol{x}} \log p(\boldsymbol{x},T-t)], \quad \text{(A.11)}$$

$$\Sigma^\leftarrow(\boldsymbol{x},t) = \Sigma(\boldsymbol{x},T-t). \quad \text{(A.12)}$$

We would next like to develop an ODE which transports the probability mass $P$ in the same way as (A.1) — namely, find another process $(\bar{\boldsymbol{z}}(t))_{t \in [0,T]}$ which has deterministic dynamics, yet has the same law as $(\boldsymbol{z}(t))_{t \in [0,T]}$. Song et al. (2021) looks at the Fokker-Planck equations (which can be defined, at least in a weak sense, under assumptions A1–A2) and manipulates them to get the following dynamics for $\bar{\boldsymbol{z}}(t)$:

$$d\bar{\boldsymbol{z}}(t) = \bar{b}(\bar{\boldsymbol{z}}(t),t)\, dt, \qquad \bar{\boldsymbol{z}}(0) \sim P, \qquad \forall t \in [0,T], \quad \text{(A.13)}$$

$$\text{where} \qquad \bar{b}(\boldsymbol{x},t) = b(\boldsymbol{x},t) - \frac{1}{2} \cdot \frac{\nabla_{\boldsymbol{x}} \cdot [\Psi(\boldsymbol{x},t)p(\boldsymbol{x},t)]}{p(\boldsymbol{x},t)} \quad \text{(A.14)}$$

$$= b(\boldsymbol{x},t) - \frac{1}{2}\nabla_{\boldsymbol{x}} \cdot \Psi(\boldsymbol{x},t) - \frac{1}{2}\Psi(\boldsymbol{x},t)[\nabla_x \log p(\boldsymbol{x},t)]. \quad \text{(A.15)}$$

Now to get a similar process for $\widetilde{\boldsymbol{z}}(t)$, namely a process $(\bar{\widetilde{\boldsymbol{z}}}(t))_{t \in [0,T]}$ which evolves deterministically yet has the same law as $(\widetilde{\boldsymbol{z}}(t))_{t \in [0,T]}$, we may either take the time reversal of (A.13) or apply the Fokker-Planck method to (A.9), in both cases obtaining the same dynamics:

$$d\bar{\widetilde{\boldsymbol{z}}}(t) = \bar{b}^\leftarrow(\bar{\widetilde{\boldsymbol{z}}}(t),t)\, dt, \qquad \bar{\widetilde{\boldsymbol{z}}}(0) \sim p(\cdot,T), \qquad \forall t \in [0,T], \quad \text{(A.16)}$$

where

$$\bar{b}^\leftarrow(\boldsymbol{x},t) = -\bar{b}(\boldsymbol{x},T-t) \quad \text{(A.17)}$$

$$= -b(\boldsymbol{x},T-t) + \frac{1}{2} \cdot \frac{\nabla_{\boldsymbol{x}} \cdot [\Psi(\boldsymbol{x},T-t)p(\boldsymbol{x},T-t)]}{p(\boldsymbol{x},T-t)} \quad \text{(A.18)}$$

$$= -b(\boldsymbol{x},t) + \frac{1}{2}\nabla_{\boldsymbol{x}} \cdot \Psi(\boldsymbol{x},T-t) + \frac{1}{2}\Psi(\boldsymbol{x},T-t)[\nabla_{\boldsymbol{x}} \log p(\boldsymbol{x},T-t)]. \quad \text{(A.19)}$$

The quantity $\nabla_{\boldsymbol{x}} \log p(\boldsymbol{x},t)$ is of central importance; it is denoted the *score at time $t$*, and we use the notation $s(\boldsymbol{x},t) \doteq \nabla_x \log p(\boldsymbol{x},t)$ for it. With this substitution, we have the following dynamics for our four processes:

$$d\boldsymbol{z}(t) = b(\boldsymbol{z}(t),t)\, dt + \Sigma(\boldsymbol{z}(t),t)\, d\boldsymbol{w}(t), \quad \boldsymbol{z}(0) \sim P \quad \text{(A.20)}$$

$$d\widetilde{\boldsymbol{z}}(t) = [-b(\widetilde{\boldsymbol{z}}(t),T-t) + \nabla_{\boldsymbol{x}} \cdot \Psi(\widetilde{\boldsymbol{z}}(t),T-t) + \Psi(\widetilde{\boldsymbol{z}}(t),T-t)s(\widetilde{\boldsymbol{z}}(t),T-t)]\, dt \quad \text{(A.21)}$$

$$\qquad + \Sigma(\widetilde{\boldsymbol{z}}(t),T-t)\, d\boldsymbol{w}^\leftarrow(t), \quad \widetilde{\boldsymbol{z}}(0) \sim p(\cdot,T) \quad \text{(A.22)}$$

$$d\bar{\boldsymbol{z}}(t) = \left[ b(\bar{\boldsymbol{z}}(t),t) - \frac{1}{2}\nabla_{\boldsymbol{x}} \cdot \Psi(\bar{\boldsymbol{z}}(t),t) - \frac{1}{2}\Psi(\bar{\boldsymbol{z}}(t),t)s(\bar{\boldsymbol{z}}(t),t) \right] dt, \quad \bar{\boldsymbol{z}}(0) \sim P \quad \text{(A.23)}$$

$$d\bar{\widetilde{\boldsymbol{z}}}(t) = \left[ -b(\bar{\widetilde{\boldsymbol{z}}}(t),T-t) + \frac{1}{2}\nabla_{\boldsymbol{x}} \cdot \Psi(\bar{\widetilde{\boldsymbol{z}}}(t),T-t) \right. \quad \text{(A.24)}$$

$$\left. + \frac{1}{2}\Psi(\bar{\widetilde{\boldsymbol{z}}}(t),T-t)s(\bar{\widetilde{\boldsymbol{z}}}(t),T-t) \right] dt, \quad \bar{\widetilde{\boldsymbol{z}}}(0) \sim p(\cdot,T). \quad \text{(A.25)}$$

In practice, one fits an estimator for $s(\cdot, \cdot)$ and estimates $p(\cdot, T)$ and runs a discretization of either (A.9) or (A.16) to sample approximately from $P$. One common instantiation used in diffusion models (Karras et al., 2022) is the so-called *variance-exploding* diffusion process, which has the coefficient settings

$$b(\boldsymbol{x}, t) = 0, \qquad \Sigma(\boldsymbol{x}, t) = \sqrt{2}\boldsymbol{I} \tag{A.26}$$

which implies that

$$\Psi(\boldsymbol{x}, t) = 2\boldsymbol{I}. \tag{A.27}$$

This means that the four specified processes are of the form

$$\mathrm{d}\boldsymbol{z}(t) = \sqrt{2}\,\mathrm{d}\boldsymbol{w}(t), \quad \boldsymbol{z}(0) \sim P \tag{A.28}$$

$$\mathrm{d}\widetilde{\boldsymbol{z}}(t) = s(\widetilde{\boldsymbol{z}}(t), T - t)\,\mathrm{d}t + \sqrt{2}\,\mathrm{d}\boldsymbol{w}^{\leftarrow}(t), \quad \widetilde{\boldsymbol{z}}(0) \sim p(\cdot, T) \tag{A.29}$$

$$\mathrm{d}\bar{\boldsymbol{z}}(t) = s(\bar{\boldsymbol{z}}(t), t)\,\mathrm{d}t, \quad \bar{\boldsymbol{z}}(0) \sim P \tag{A.30}$$

$$\mathrm{d}\bar{\bar{\boldsymbol{z}}}(t) = -s(\bar{\bar{\boldsymbol{z}}}(t), T - t), \quad \bar{\bar{\boldsymbol{z}}}(0) \sim p(\cdot, T). \tag{A.31}$$

Notice that the determinstic flows are actually gradient flows on the score, which concretely reveals a connection between sampling and optimization, and thus between diffusion models (precisely those which use the probability flow ODE to sample) and unrolled optimization networks.

In this context, we can also establish the connection between diffusion networks and iterative denoising. In the variance-exploding setting, we have

$$\boldsymbol{z}(t) \sim \mathcal{N}(\boldsymbol{z}(0), 2t\boldsymbol{I}), \tag{A.32}$$

which can be easily computed using results from, e.g., Särkkä & Solin (2019). Thus $\boldsymbol{z}(t)$ is a noisy version of $\boldsymbol{z}(0)$, with noise level increasing monotonically with $t$, and sampling $\boldsymbol{z}(0)$ from $\boldsymbol{z}(t)$ conceptually removes this noise. Concretely, *Tweedie's formula* (Efron, 2011) says that the optimal denoising function $\mathbb{E}[\boldsymbol{z}(0) \mid \boldsymbol{z}(t)]$ has a simple form in terms of the score function:

$$\mathbb{E}[\boldsymbol{z}(0) \mid \boldsymbol{z}(t)] = \boldsymbol{z}(t) + 2t \cdot s(\boldsymbol{z}(t), t). \tag{A.33}$$

In other words, the score function $s$ points from the current iterate $\boldsymbol{z}(t)$ to the value of the optimal denoising function, so it is a negative multiple of the conditionally-expected noise. Following the score by (stochastic) gradient flow or its discretization is thus equivalent to iterative denoising.

## A.2 COMPANION TO SECTION 2.3

In this section, we prove a formal version of the result Theorem 1 stated in Section 2.3. That is, we examine a basic yet representative instantiation of the signal model (2.2), and show that under this model, in a natural regime of parameter scales motivated by the architecture of CRATE-MAE applied to standard image classification benchmarks, the operation implemented by taking a gradient step on the compression term of the sparse rate reduction objective (2.3) corresponds to a projection operation at quantization scales $\varepsilon^2$ proportional to the size of the deviation. This leads us in particular to a formal version of the result Theorem 1.

**Signal model.** We consider an instantiation of the model (2.2), elaborated here. That is, we fix a distribution over tokens $\boldsymbol{Z} \in \mathbb{R}^{d \times N}$ induced by the following natural signal model: each token $\boldsymbol{z}_i$ is drawn independently from the normalized isotropic Gaussian measure on one of $K$ $p$-dimensional subspaces with orthonormal bases $\boldsymbol{U}_1, \ldots, \boldsymbol{U}_K \in \mathbb{R}^{d \times p}$,[2] which comprise the low-dimensional structure in the observed tokens, then corrupted with i.i.d. Gaussian noise $\mathcal{N}(\boldsymbol{0}, \frac{\sigma^2}{d}\boldsymbol{I})$; the subspace each token is drawn from is selected uniformly at random, independently of all other randomness in the problem. This signal model therefore corresponds to the setting of uncorrelated tokens, with maximum entropy coordinate distributions within subspaces. It is natural to first develop our theoretical understanding of the connection between compression and the score function in the uncorrelated setting, although in general, the ability of CRATE-MAE to capture correlations in the data through the MSSA block is essential. In connection with the latter issue, we note that our proofs will generalize straightforwardly to the setting of "well-dispersed" correlated tokens: see the discussion in Remark 5.

We make the further following assumptions within this model:

---

[2]More precisely, $\boldsymbol{z}_i$ is distributed according to the pushforward of the normalized isotropic Gaussian measure $\mathcal{N}(\boldsymbol{0}, \frac{1}{p}\boldsymbol{I})$ on $\mathbb{R}^p$ by the bases $\boldsymbol{U}_k$.

1. Inspired by an ablation in Yu et al. (2023a), which suggests that the learned CRATE-MAE model on supervised classification on ImageNet has signal models $U_k$ which are near-incoherent, we will assume that the subspaces $U_k$ have pairwise orthogonal column spaces. Our proofs will generalize straightforwardly to the setting where the subspaces are merely incoherent: see the discussion in Remark 5.

2. We assume that the relative scales of these parameters conform to the CRATE-MAE-Base settings, trained on ImageNet: from Table 1, these parameters are

   (a) $d = 768$;
   (b) $N = 196$;
   (c) $K = 12$;
   (d) $p = d/K = 64$.

   In particular, $d \gg N \gg p$ and $Kp = d$.

These precise parameter values will not play a role in our analysis. We merely require the following quantitative relationships between the parameter values, which are more general than the above precise settings.

**Assumption 2.** *We have $\varepsilon \leq 1$, $U_k^\top U_{k'} = \mathbb{1}_{k=k'} I$ for all $k \neq k'$, and the following parameter settings and scales:*

- $d \geq N \geq p \geq K \geq 2$;

- $Kp = d$;

- $C_1\sqrt{N \log N} \leq \frac{1}{2}N/K$, *where $C_1$ is the same as the universal constant $C_1$ in the statement of Proposition 12;*

- $6C_2^2 N \leq d$, *where $C_2$ is the same as the universal constant $C_3$ in the statement of Proposition 15;*

- $2C_4^2 N \leq d$, *where $C_4$ is the same as the universal constant $C_1$ in Proposition 11;*

**Note:** there is no self-reference, as the third inequality is not used to prove Proposition 12, the fourth is not used to prove Proposition 15, and the fifth is not used to prove Proposition 11.

The first and second inequalities together imply in particular that $p \geq N/K$. The third inequality implies that $C_1\sqrt{N \log N} < N/K$. The first, second, and and third inequalities together imply that $p > C_1\sqrt{N \log N}$, and that $0 < N/K - C_1\sqrt{N \log N} < N/K < N/K + C_1\sqrt{N \log N} < N$.

These inequalities are verifiable in practice if one wishes to explicitly compute the absolute constants $C_1, C_2, C_3, C_4$, and indeed they hold for our CRATE-MAE-Base model.

Formally, let $\mu(K, p, \sigma^2)$ denote the probability measure on $\mathbb{R}^{d \times N}$ corresponding to the noisy Gaussian mixture distribution specified above. We let $Z_\natural \sim \mu$ denote an observation distributed according to this signal model: formally, there exists a (random) map $i \mapsto s_i$, for $i \in [N]$ and $s_i \in [K]$, such that

$$z_{\natural i} = U_{s_i}\alpha_i + \delta_i, \quad i = 1, \ldots, n, \tag{A.34}$$

where $\Delta = [\delta_1 \quad \ldots \quad \delta_N] \sim_{\text{i.i.d.}} \mathcal{N}(0, \frac{\sigma^2}{d}I)$, and (independently) $\alpha_i \sim_{\text{i.i.d.}} \mathcal{N}(0, \frac{1}{p}I)$. It is convenient to write this observation model in block form. To this end, let $K_k = \sum_{i=1}^N \mathbb{1}_{s_i = k}$ for $k \in [K]$ denote the number of times the $k$-th subspace is represented amongst the columns of $Z_\natural$ (a random variable). Then by rotational invariance of the Gaussian distribution, we have

$$Z_\natural \overset{d}{=} [U_1 A_1 \quad \ldots \quad U_K A_K] \Pi + \Delta, \tag{A.35}$$

where $\overset{d}{=}$ denotes equality in distribution, $\Pi \in \mathbb{R}^{N \times N}$ is a uniformly random permutation matrix, and each $A_k \in \mathbb{R}^{p \times K_k}$. We also define $X_\natural$ to be the noise-free version of $Z_\natural$.

Because of this equality in distribution, we will commit the mild abuse of notation of identifying the block representation (A.35) with the observation model (A.34) that follows the distribution $\mu$.

**Denoising in the uncorrelated tokens model.** In the uncorrelated tokens model (A.35), the marginal distribution of each column of $\boldsymbol{Z}_\natural$ is identical, and equal to an equiproportional mixture of (normalized) isotropic Gaussians on the subspaces $\boldsymbol{U}_1, \ldots \boldsymbol{U}_k$, convolved with the noise distribution $\mathcal{N}(\boldsymbol{0}, \frac{\sigma^2}{d}\boldsymbol{I})$. This marginal distribution was studied in Yu et al. (2023a), where it was shown that when the perturbation level $\sigma^2 \to 0$, the score function for this marginal distribution approximately implements a projection operation onto the nearest subspace $\boldsymbol{U}_k$.

Hence, we can connect compression, as implemented in the MSSA block of the CRATE-MAE architecture, to denoising in the uncorrelated tokens model by showing that at similar local scales, and for suitable settings of the model parameters, the compression operation implements a projection onto the low-dimensional structure of the distribution, as well.

**Compression operation.** The MSSA block of the CRATE-MAE architecture arises from taking an (approximate) gradient step on the $R^c$ term of the sparse rate reduction objective (2.3). This term writes

$$R^c(\boldsymbol{Z} \mid \boldsymbol{U}_{[K]}) = \frac{1}{2}\sum_{k=1}^{K} \log \det \left( \boldsymbol{I} + \beta(\boldsymbol{U}_k^\top \boldsymbol{Z})^\top \boldsymbol{U}_k^\top \boldsymbol{Z} \right), \tag{A.36}$$

where

$$\beta = \frac{p}{N\varepsilon^2}, \tag{A.37}$$

and $\varepsilon > 0$ is the quantization error. Calculating the gradient, we have

$$\nabla_{\boldsymbol{Z}} R^c(\boldsymbol{Z} \mid \boldsymbol{U}_{[K]}) = \sum_{k=1}^{K} \boldsymbol{U}_k \boldsymbol{U}_k^\top \boldsymbol{Z} \left( \beta^{-1}\boldsymbol{I} + (\boldsymbol{U}_k^\top \boldsymbol{Z})^\top \boldsymbol{U}_k^\top \boldsymbol{Z} \right)^{-1}. \tag{A.38}$$

Minimizing the sparse rate reduction objective corresponds to taking a gradient descent step on $R^c(\cdot \mid \boldsymbol{U}_{[K]})$. Performing this operation at the observation from the uncorrelated tokens model $\boldsymbol{Z}_\natural$, the output can be written as

$$\boldsymbol{Z}^+ = \boldsymbol{Z}_\natural - \eta \nabla R^c(\boldsymbol{Z}_\natural \mid \boldsymbol{U}_{[K]}), \tag{A.39}$$

where $\eta > 0$ is the step size.

**Main result on projection.** We will see shortly that the behavior of the compression output (A.39) depends on the relative scales of the perturbation about the low-dimensional structure $\sigma^2$ and the target quantization error $\varepsilon^2$.

**Theorem 3.** *There are universal constants $C_1, C_2, C_3, C_4 > 0$ such that the following holds. Suppose Assumption 2 holds, and moreover suppose that $\sigma \le 1$ and $C_1\beta\sigma \le \frac{1}{2}$. Then with probability at least $1 - KC_2\left(e^{-C_3 d} + e^{-C_4 N/K} + N^{-2}\right)$, it holds*

$$\left\| \boldsymbol{Z}^+ - \left[ (\boldsymbol{\Delta} - \eta\mathcal{P}_{\boldsymbol{U}_{[K]}}(\beta\boldsymbol{\Delta}\boldsymbol{\Pi}^\top)\boldsymbol{\Pi}) + \frac{1 + \beta^{-1} - \eta}{1 + \beta^{-1}}\boldsymbol{X}_\natural \right] \right\| \tag{A.40}$$

$$\le C_5 K\eta \left( \sigma^2\beta^2 + \sigma(1 + \sqrt{N/d}) + \sqrt{K}\beta\sigma^2(1 + \sqrt{N/d}) + \sqrt{N/d} \right). \tag{A.41}$$

*Here, $\mathcal{P}_{\boldsymbol{U}_{[K]}}$ implements a projection onto the relevant subspaces for each token in the limiting case as $\varepsilon \to 0$, and is precisely defined in (A.116) and (A.117).*

We give the proof of Theorem 3 below. First, we make three remarks on interpreting the result, our technical assumptions, and our analysis.

*Remark* 4. Theorem 3 admits the following interesting interpretation in an asymptotic setting, where we can identify the leading-order behavior of the gradient and confirm our hypothesis about the connection between compression and score-following. Choose $\eta = \beta^{-1}$, so that the guarantee in Theorem 3 incurs some cancellation, and moreover delineate more precise dependencies on the RHS

of the guarantee:

$$\left\| \boldsymbol{Z}^+ - \left[ (\boldsymbol{\Delta} - \mathcal{P}_{\boldsymbol{U}_{[K]}}(\boldsymbol{\Delta}\boldsymbol{\Pi}^\top)\boldsymbol{\Pi}) + \frac{1}{1+\beta^{-1}}\boldsymbol{X}_\natural \right] \right\| \tag{A.42}$$

$$\lesssim \frac{NK^2\varepsilon^2}{d} \left( \frac{\sigma^2 d^2}{N^2 K^2 \varepsilon^4} + \sigma(1+\sqrt{N/d}) + \frac{d\sigma^2}{N\sqrt{K}\varepsilon^2}(1+\sqrt{N/d}) + \sqrt{N/d} \right) \tag{A.43}$$

$$\lesssim K^{3/2}\sigma^2 + \frac{\sigma^2 d}{N\varepsilon^2} + \frac{NK^2}{d}\left( \sigma + \sqrt{\frac{N}{d}} \right)\varepsilon^2, \tag{A.44}$$

where we used Assumption 2, which implies $p = d/K$ and $N/d \le 1$. We will check in due course whether we have satisfied the hypotheses of Theorem 3, so that this guarantee indeed applies. To this end, we optimize this bound as a function of $\varepsilon > 0$, since this is a parameter of the compression model. The optimal $\varepsilon$ is straightforward to compute using calculus: it satisfies

$$\varepsilon^2 = \sqrt{ \frac{\sigma^2 d}{N} \bigg/ \frac{K^2 N}{d}\left( \sigma + \sqrt{\frac{N}{d}} \right) } \tag{A.45}$$

$$= \frac{\sigma d}{NK\sqrt{\sigma + \sqrt{\frac{N}{d}}}}, \tag{A.46}$$

and the value of the residual arising from Theorem 3 with this choice of $\varepsilon$ is no larger than an absolute constant multiple of

$$K^{3/2}\sigma^2 + \sqrt{ \frac{K^2\sigma^2 d}{N}\left( \frac{N\sigma}{d} + \left( \frac{N}{d} \right)^{3/2} \right) } = K\sigma\left( \sqrt{K}\sigma + \sqrt{\sigma + \sqrt{\frac{N}{d}}} \right). \tag{A.47}$$

Moreover, with this choice of $\varepsilon$, $\beta$ satisfies

$$\beta^{-1} = \frac{\varepsilon^2 NK}{d} = \sqrt{ \frac{\sigma}{1 + \sqrt{\frac{N}{d\sigma^2}}} }. \tag{A.48}$$

In particular, the condition $\beta\sigma \lesssim 1$ in Theorem 3 demands

$$\sqrt{\sigma + \sqrt{\frac{N}{d}}} \lesssim 1, \tag{A.49}$$

which holds for sufficiently small $\sigma$ and sufficiently large $d \ge N$, showing that Theorem 3 can be nontrivially applied in this setting. If we consider a simplifying limiting regime where $N, d \to +\infty$ such that $N/d \to 0$ and $N/K \to +\infty$, we observe the following asymptotic behavior of the guarantee of Theorem 3:

$$\left\| \boldsymbol{Z}^+ - \left[ (\boldsymbol{\Delta} - \mathcal{P}_{\boldsymbol{U}_{[K]}}(\boldsymbol{\Delta}\boldsymbol{\Pi}^\top)\boldsymbol{\Pi}) + \frac{1}{1+\sqrt{\sigma}}\boldsymbol{X}_\natural \right] \right\| \lesssim K\sigma^{3/2}\left( 1 + \sqrt{K}\sigma \right). \tag{A.50}$$

This demonstrates that a gradient step on $R^c$ performs denoising: there is a noise-level-dependent shrinkage effect applied to the signal $\boldsymbol{X}_\natural$, which vanishes as $\sigma \to 0$, and meanwhile the noise term $\boldsymbol{\Delta}$ is reduced.

Moreover, as $\sigma \to 0$, we can express the limiting form of $\mathcal{P}_{\boldsymbol{U}_{[K]}}$ exactly as an orthogonal projection, since this drives $\beta^{-1} \to 0$: following (A.116) and (A.117), we have here

$$\mathcal{P}_{\boldsymbol{U}_{[K]}} = [\mathcal{P}_1 \quad \dots \quad \mathcal{P}_K], \tag{A.51}$$

where

$$\mathcal{P}_k \to \sum_{k' \ne k} \boldsymbol{U}_{k'} \operatorname{proj}_{\operatorname{im}(\boldsymbol{A}_{k'})^\perp} \boldsymbol{U}_{k'}^\top. \tag{A.52}$$

This shows that, in an asymptotic sense, a gradient step on $R^c$ serves to *suppress the effect of the perturbation applied to the observations $\boldsymbol{Z}_\natural$ about the local signal model $\boldsymbol{X}_\natural$.* This verifies our claim

previously that in this setting, there is a correspondence between a score-following algorithm and a compression-based approach: locally, both project the observations onto the structures of the signal model.

It can be shown moreover that the shrinkage effect on $X_\natural$ demonstrated here appears as a consequence of using the $R^c$ "compression" term for the gradient step in CRATE-MAE; when the gradient step is taken instead on the full $\Delta R$ rate reduction objective (which is computationally prohibitive, of course), there is zero shrinkage, and perfect denoising is performed for a wider variety of step sizes $\eta$ than the choice made here. We see the introduction of this shrinkage effect this as the price of constructing an efficient and interpretable network architecture. In practice, the ISTA block of CRATE-MAE counteracts this shrinkage effect, which is anyways minor at reasonable parameter scales.

*Remark* 5. We have made two assumptions which may not hold exactly in practice: namely, we have assumed that the $U_k$'s have orthogonal columns, namely $U_k^\top U_{k'} = \mathbb{1}_{k=k'} I$, and we have assumed that the linear combination coefficients $A_k$ that form the matrix $X_\natural$ are i.i.d. samples from Gaussian distributions. Both these assumptions can be made more realistic, at the cost of additional (non-instructive) complexity in the analysis; we briefly go over how.

Relaxing the orthogonality condition $U_k^\top U_{k'} = \mathbb{1}_{k=k'} I$ to near-orthogonality, namely $\|U_k^\top U_{k'} - \mathbb{1}_{k=k'} I\| \leq \nu$ for a small $\nu$, as observed in practice (Yu et al., 2023a) would introduce additional small error terms in the proof, say polynomial in $\nu$. The magnitudes of these errors could in principle be precisely tracked, whence one could obtain a similar result to Theorem 3.

Secondly, we have assumed that the $A_k$'s have independent columns which are sampled from (the same) Gaussian distribution. However, in the conceptual framework for CRATE-MAE, we exploit the joint distribution (and in particular the correlations) between the tokens in order to obtain good performance for our model. Our analysis is not completely agnostic to this fact; as we will see, the proof of Theorem 3 only leverages the independence of the columns of each $A_k$'s in order to obtain high-probability upper bounds on the smallest and largest singular value of the token matrices. If these bounds were ensured by some other method, such as appropriate normalization and incoherence, a similar conclusion to Theorem 3 could hold in the more realistic correlated tokens model. Going beyond well-conditioned token matrices for each subspace would require additional modeling assumptions, and additional investigative experimental work to determine a realistic basis for such assumptions.

*Remark* 6. We have not attempted to optimize constants or rates of concentration in the proof of Theorem 3, preferring instead to pursue a straightforward analysis that leads to a qualitative interpretation of the behavior of the rate reduction gradient in our model problem. Minor improvements to the concentration analysis would enable the parameter scaling requirements in Assumption 2 to be relaxed slightly, and the probability bound in Theorem 3 that scales as $K/N^2$ can easily be improved to any positive power of $1/N$.

*Proof of Theorem 3.* We start by noticing that, by orthonormality of the subspaces $U_k$, we have by (A.35)

$$U_k^\top Z_\natural = \begin{bmatrix} \mathbf{0} & \dots & A_k & \dots & \mathbf{0} \end{bmatrix} \Pi + U_k^\top \Delta, \tag{A.53}$$

so that

$$\left( \beta^{-1} I + (U_k^\top Z_\natural)^\top U_k^\top Z_\natural \right)^{-1} = \Pi^\top \left( \underbrace{\begin{bmatrix} \beta^{-1} I & & & & \\ & \ddots & & & \\ & & \beta^{-1} I + A_k^\top A_k & & \\ & & & \ddots & \\ & & & & \beta^{-1} I \end{bmatrix}}_{D_k} + \Xi_k \right)^{-1} \Pi, \tag{A.54}$$

because permutation matrices are orthogonal matrices, and where the perturbation $\boldsymbol{\Xi}_k$ is defined by

$$\boldsymbol{\Xi}_k = \boldsymbol{\Pi}\boldsymbol{\Delta}^\top \boldsymbol{U}_k \boldsymbol{U}_k^\top \boldsymbol{\Delta}\boldsymbol{\Pi}^\top + \begin{bmatrix} \mathbf{0} & \dots & \boldsymbol{\Delta}_1^\top \boldsymbol{U}_k \boldsymbol{A}_k & \dots & \mathbf{0} \\ \vdots & & \vdots & & \vdots \\ \boldsymbol{A}_k^\top \boldsymbol{U}_k^\top \boldsymbol{\Delta}_1 & \dots & \boldsymbol{\Delta}_k^\top \boldsymbol{U}_k \boldsymbol{A}_k + \boldsymbol{A}_k^\top \boldsymbol{U}_k^\top \boldsymbol{\Delta}_k & \dots & \boldsymbol{A}_k^\top \boldsymbol{U}_k^\top \boldsymbol{\Delta}_K \\ \vdots & & \vdots & & \vdots \\ \mathbf{0} & \dots & \boldsymbol{\Delta}_K^\top \boldsymbol{U}_k \boldsymbol{A}_k & \dots & \mathbf{0} \end{bmatrix}, \tag{A.55}$$

and where we have defined (implicitly) in addition

$$[\boldsymbol{\Delta}_1 \quad \dots \quad \boldsymbol{\Delta}_K] = \boldsymbol{\Delta}\boldsymbol{\Pi}^\top. \tag{A.56}$$

The matrix $\boldsymbol{D}_k \succ \mathbf{0}$, so we can write

$$\left(\beta^{-1}\boldsymbol{I} + (\boldsymbol{U}_k^\top \boldsymbol{Z}_\natural)^\top \boldsymbol{U}_k^\top \boldsymbol{Z}_\natural\right)^{-1} = \boldsymbol{\Pi}^\top \boldsymbol{D}_k^{-1}\left(\boldsymbol{I} + \boldsymbol{\Xi}_k \boldsymbol{D}_k^{-1}\right)^{-1}\boldsymbol{\Pi}, \tag{A.57}$$

from which it follows

$$\boldsymbol{U}_k^\top \boldsymbol{Z}_\natural \left(\beta^{-1}\boldsymbol{I} + (\boldsymbol{U}_k^\top \boldsymbol{Z}_\natural)^\top \boldsymbol{U}_k^\top \boldsymbol{Z}_\natural\right)^{-1} \tag{A.58}$$

$$= \left([\mathbf{0} \quad \dots \quad \boldsymbol{A}_k(\beta^{-1}\boldsymbol{I} + \boldsymbol{A}_k^\top \boldsymbol{A}_k)^{-1} \quad \dots \quad \mathbf{0}] + \boldsymbol{U}_k^\top \boldsymbol{\Delta}\boldsymbol{\Pi}^\top \boldsymbol{D}_k^{-1}\right)\left(\boldsymbol{I} + \boldsymbol{\Xi}_k \boldsymbol{D}_k^{-1}\right)^{-1}\boldsymbol{\Pi}. \tag{A.59}$$

The task before us is therefore to control $\|\boldsymbol{\Xi}_k \boldsymbol{D}_k^{-1}\| < 1$, in order to apply the Neumann series to further simplify this expression. We will do this in stages: first, we invoke several auxiliary lemmas to construct a high-probability event on which the random quantities in the preceding expression are controlled about their nominal values; next, we show that the Neumann series can be applied on this event and a main term extracted; finally, we simplify this main term further in order to establish the claimed expression.

**High-probability event construction.**    In order to achieve the appropriate control on all random quantities, we would like to construct a high-probability event on which the random quantities are not too large. By Propositions 9, 10 and 11 and union bound, there exist universal constants $C_i > 0$ for which

$$\mathbb{P}\begin{bmatrix} & \|\boldsymbol{\Delta}\| \le \sigma(C_1 + \sqrt{N/d}) \\ \forall k \in [K]: & \|\boldsymbol{A}_k\| \le 1 + C_2\sqrt{N/d} \\ \forall k \in [K]: & \|\boldsymbol{A}_k^\top \boldsymbol{A}_k - \boldsymbol{I}\| \le C_3\sqrt{N/d} \end{bmatrix} \ge 1 - C_4 K(e^{-C_5 d} + e^{-C_6 N/K} + N^{-2}). \tag{A.60}$$

The event we compute the probability of, which we denote $E^\star$, is precisely the good event that we want. Formally,

$$E^\star \doteq \left\{ \begin{array}{ll} & \|\boldsymbol{\Delta}\| \le \sigma(C_1 + \sqrt{N/d}) \\ \forall k \in [K]: & \|\boldsymbol{A}_k\| \le 1 + C_2\sqrt{N/d} \\ \forall k \in [K]: & \|\boldsymbol{A}_k^\top \boldsymbol{A}_k - \boldsymbol{I}\| \le C_3\sqrt{N/d} \end{array} \right\}. \tag{A.61}$$

We know that $E^\star$ occurs with high probability, and are able to strongly control the random quantities to the degree desired.

**Main term extraction.**    By Lemma 8 and our hypotheses on the problem parameters, we have on $E^\star$ that

$$\|\boldsymbol{\Xi}_k \boldsymbol{D}_k^{-1}\| \le C\beta\sigma < 1. \tag{A.62}$$

We can therefore apply the Neumann series to obtain

$$\boldsymbol{U}_k^\top \boldsymbol{Z}_\natural \left(\beta^{-1}\boldsymbol{I} + (\boldsymbol{U}_k^\top \boldsymbol{Z}_\natural)^\top \boldsymbol{U}_k^\top \boldsymbol{Z}_\natural\right)^{-1} \tag{A.63}$$

$$= \left([\mathbf{0} \quad \dots \quad \boldsymbol{A}_k(\beta^{-1}\boldsymbol{I} + \boldsymbol{A}_k^\top \boldsymbol{A}_k)^{-1} \quad \dots \quad \mathbf{0}] + \boldsymbol{U}_k^\top \boldsymbol{\Delta}\boldsymbol{\Pi}^\top \boldsymbol{D}_k^{-1}\right)\left(\boldsymbol{I} - \boldsymbol{\Xi}_k \boldsymbol{D}_k^{-1} + \sum_{j=2}^\infty (-1)^j \left(\boldsymbol{\Xi}_k \boldsymbol{D}_k^{-1}\right)^j\right)\boldsymbol{\Pi}. \tag{A.64}$$

Again on $E^\star$, we have

$$\left\|\sum_{j=2}^\infty (-1)^j \left(\boldsymbol{\Xi}_k \boldsymbol{D}_k^{-1}\right)^j\right\| \le \sum_{j=2}^\infty \|\boldsymbol{\Xi}_k \boldsymbol{D}_k^{-1}\|^j \le C(\beta\sigma)^2 \frac{1}{1 - C\beta\sigma} \le C'(\beta\sigma)^2. \tag{A.65}$$

Moreover, as in the proof of Lemma 8, we have on the previous event that

$$\left\| U_k^\top \Delta \Pi^\top D_k^{-1} \right\| \le C\beta\sigma. \tag{A.66}$$

Thus, if we define a "main term"

$$M_k = \left[ \begin{bmatrix} \mathbf{0} & \dots & A_k(\beta^{-1}I + A_k^\top A_k)^{-1} & \dots & \mathbf{0} \end{bmatrix} \left( I - \Xi_k D_k^{-1} \right) + U_k^\top \Delta \Pi^\top D_k^{-1} \right] \Pi, \tag{A.67}$$

we have on the same event as previously

$$\left\| U_k^\top Z_\natural \left( \beta^{-1}I + (U_k^\top Z_\natural)^\top U_k^\top Z_\natural \right)^{-1} - M_k \right\| \le C(\beta\sigma)^2. \tag{A.68}$$

To conclude, we need only study this main term, since $U_k$ has operator norm 1.

**Simplifying the main term.** Our approach will be to control the main term $M_k$ around a simpler expression, using basic perturbation theory; by the triangle inequality for the operator norm, this will give control of the desired gradient term. After distributing, $M_k$ is a sum of three terms; we will start with the simplest term. We first compute

$$U_k^\top \Delta \Pi^\top D_k^{-1} = U_k^\top \begin{bmatrix} \beta\Delta_1 & \dots \Delta_k \left( \beta^{-1}I + A_k^\top A_k \right)^{-1} & \dots & \beta\Delta_K \end{bmatrix}. \tag{A.69}$$

We are going to argue that the residual

$$\left\| U_k^\top \Delta \Pi^\top D_k^{-1} - U_k^\top \begin{bmatrix} \beta\Delta_1 & \dots & \mathbf{0} & \dots & \beta\Delta_K \end{bmatrix} \right\| \tag{A.70}$$

is small. To this end, note that by the fact that $U_k$ has unit operator norm,

$$\left\| U_k^\top \Delta \Pi^\top D_k^{-1} - U_k^\top \begin{bmatrix} \beta\Delta_1 & \dots & \mathbf{0} & \dots & \beta\Delta_K \end{bmatrix} \right\| \tag{A.71}$$

$$\le \left\| \begin{bmatrix} \mathbf{0} & \dots & \Delta_k \left( \beta^{-1}I + A_k^\top A_k \right)^{-1} & \dots & \mathbf{0} \end{bmatrix} \right\| \tag{A.72}$$

$$= \left\| \Delta_k \left( \beta^{-1}I + A_k^\top A_k \right)^{-1} \right\| \tag{A.73}$$

$$\le \left\| \Delta_k \right\| \left\| \left( \beta^{-1}I + A_k^\top A_k \right)^{-1} \right\|. \tag{A.74}$$

By (A.61) and (A.120) from Lemma 7, the second term here is controlled on $E^\star$. For the first term, we note that by definition and the fact that the unit sphere is invariant to rotations (and permutations are rotations),

$$\left\| \Delta \right\| = \sup_{\|u\|_2 \le 1} \|\Delta u\|_2 = \sup_{\|u\|_2 \le 1} \left\| \begin{bmatrix} \Delta_1 & \dots & \Delta_K \end{bmatrix} u \right\|_2 \tag{A.75}$$

$$= \sup_{\|u\|_2 \le 1} \left\| \sum_{i=1}^K \Delta_i u_i \right\|_2, \tag{A.76}$$

where $u_i$ are coordinate-subset-induced partitions of the vector $u$ induced by those of $\Delta\Pi^\top$. This yields immediately

$$\left\| \Delta \right\| \le \sup_{\|u\|_2 \le 1} \sum_{i=1}^K \|\Delta_i u_i\|_2 \le \left( \max_{k \in [K]} \|\Delta_k\| \right) \sup_{\|u\|_2 \le 1} \sum_{i=1}^K \|u_i\|_2 \le \sqrt{K} \left( \max_{k \in [K]} \|\Delta_k\| \right), \tag{A.77}$$

by the triangle inequality and inequalities for $\ell^p$ norms. Similarly, choosing a specific $u$ in the operator norm expression, namely one that is supported entirely on one of the coordinate partitions $u_i$, shows that

$$\left\| \Delta \right\| \ge \|\Delta_i u_i\|_2 \tag{A.78}$$

for any $i$, whence

$$\max_{k \in [K]} \|\Delta_k\| \le \|\Delta\|. \tag{A.79}$$

It follows that we control the first term above on $E^\star$. Combining this reasoning, we conclude from the above

$$\left\| U_k^\top \Delta \Pi^\top D_k^{-1} - U_k^\top \begin{bmatrix} \beta\Delta_1 & \dots & \mathbf{0} & \dots & \beta\Delta_K \end{bmatrix} \right\| \tag{A.80}$$

$$\le \sigma(C + \sqrt{N/d}) \left( \frac{1}{1 + \beta^{-1}} + \frac{C'\sqrt{N/d}}{1 + \beta^{-1}} \right) \tag{A.81}$$

$$\lesssim \sigma(1 + C\sqrt{N/d}), \tag{A.82}$$

where the second line uses Assumption 2 to remove the higher-order residual.

Next, we recall that $\Xi_k$ is a sum of two terms; we will do one term at a time for concision. We have first

$$\begin{bmatrix} \mathbf{0} & \dots & \boldsymbol{A}_k(\beta^{-1}\boldsymbol{I} + \boldsymbol{A}_k^\top \boldsymbol{A}_k)^{-1} & \dots & \mathbf{0} \end{bmatrix} \boldsymbol{\Pi}\boldsymbol{\Delta}^\top \boldsymbol{U}_k \boldsymbol{U}_k^\top \boldsymbol{\Delta}\boldsymbol{\Pi}^\top \tag{A.83}$$

$$= \begin{bmatrix} \mathbf{0} & \dots & \boldsymbol{A}_k(\beta^{-1}\boldsymbol{I} + \boldsymbol{A}_k^\top \boldsymbol{A}_k)^{-1} & \dots & \mathbf{0} \end{bmatrix} \begin{bmatrix} \boldsymbol{\Delta}_1^\top \\ \vdots \\ \boldsymbol{\Delta}_K^\top \end{bmatrix} \boldsymbol{U}_k \boldsymbol{U}_k^\top \begin{bmatrix} \boldsymbol{\Delta}_1^\top \\ \vdots \\ \boldsymbol{\Delta}_K^\top \end{bmatrix}^\top \tag{A.84}$$

$$= \boldsymbol{A}_k(\beta^{-1}\boldsymbol{I} + \boldsymbol{A}_k^\top \boldsymbol{A}_k)^{-1}\boldsymbol{\Delta}_k^\top \boldsymbol{U}_k \boldsymbol{U}_k^\top \begin{bmatrix} \boldsymbol{\Delta}_1 & \dots & \boldsymbol{\Delta}_K \end{bmatrix}. \tag{A.85}$$

We then multiply this term by $\boldsymbol{D}_k^{-1}$ on the right to get the term that appears in $\boldsymbol{M}_k$ (ignoring the multiplication on the right by $\boldsymbol{\Pi}$, because it does not change operator norms). In particular, we will control

$$\left\| \boldsymbol{A}_k(\beta^{-1}\boldsymbol{I} + \boldsymbol{A}_k^\top \boldsymbol{A}_k)^{-1}\boldsymbol{\Delta}_k^\top \boldsymbol{U}_k \boldsymbol{U}_k^\top \begin{bmatrix} \boldsymbol{\Delta}_1 & \dots & \boldsymbol{\Delta}_K \end{bmatrix} \boldsymbol{D}_k^{-1} \right\|, \tag{A.86}$$

showing that this term is small. We will accomplish this with the block diagonal structure of $\boldsymbol{D}_k$: indeed, this gives that $\boldsymbol{D}_k^{-1}$ is obtained by blockwise inversion, and hence

$$\left\| \boldsymbol{A}_k(\beta^{-1}\boldsymbol{I} + \boldsymbol{A}_k^\top \boldsymbol{A}_k)^{-1}\boldsymbol{\Delta}_k^\top \boldsymbol{U}_k \boldsymbol{U}_k^\top \begin{bmatrix} \boldsymbol{\Delta}_1 & \dots & \boldsymbol{\Delta}_K \end{bmatrix} \boldsymbol{D}_k^{-1} \right\| \tag{A.87}$$

$$= \left\| \boldsymbol{A}_k(\beta^{-1}\boldsymbol{I} + \boldsymbol{A}_k^\top \boldsymbol{A}_k)^{-1}\boldsymbol{\Delta}_k^\top \boldsymbol{U}_k \boldsymbol{U}_k^\top \begin{bmatrix} \beta\boldsymbol{\Delta}_1 & \dots & \boldsymbol{\Delta}_k\left(\beta^{-1}\boldsymbol{I} + \boldsymbol{A}_k^\top \boldsymbol{A}_k\right)^{-1} & \dots & \beta\boldsymbol{\Delta}_K \end{bmatrix} \right\| \tag{A.88}$$

$$\leq \sqrt{K} \max\left\{ \left\| \boldsymbol{A}_k(\beta^{-1}\boldsymbol{I} + \boldsymbol{A}_k^\top \boldsymbol{A}_k)^{-1}\boldsymbol{\Delta}_k^\top \boldsymbol{U}_k \boldsymbol{U}_k^\top \boldsymbol{\Delta}_k(\beta^{-1}\boldsymbol{I} + \boldsymbol{A}_k^\top \boldsymbol{A}_k)^{-1} \right\|, \right. \tag{A.89}$$

$$\left. \max_{k' \neq k} \beta\left\| \boldsymbol{A}_k(\beta^{-1}\boldsymbol{I} + \boldsymbol{A}_k^\top \boldsymbol{A}_k)^{-1}\boldsymbol{\Delta}_k^\top \boldsymbol{U}_k \boldsymbol{U}_k^\top \boldsymbol{\Delta}_{k'} \right\| \right\}, \tag{A.90}$$

where the second line uses (A.77). We will give a coarse control of this term—the error could be improved further by exploiting more thoroughly independence of the blocks $\boldsymbol{\Delta}_k$ to show that the maximum over $k' \neq k$ in the last line of the preceding expression is smaller. We have by submultiplicativity of the operator norm and the triangle inequality

$$\left\| \boldsymbol{A}_k(\beta^{-1}\boldsymbol{I} + \boldsymbol{A}_k^\top \boldsymbol{A}_k)^{-1}\boldsymbol{\Delta}_k^\top \boldsymbol{U}_k \boldsymbol{U}_k^\top \boldsymbol{\Delta}_k(\beta^{-1}\boldsymbol{I} + \boldsymbol{A}_k^\top \boldsymbol{A}_k)^{-1} \right\| \tag{A.91}$$

$$\leq \left( \frac{1}{1 + \beta^{-1}} + \frac{C\sqrt{N/d}}{1 + \beta^{-1}} \right)^2 \left\| \boldsymbol{\Delta}_k^\top \boldsymbol{U}_k \boldsymbol{U}_k^\top \boldsymbol{\Delta}_k \right\| \tag{A.92}$$

$$\leq \left( 1 + C\sqrt{N/d} \right) \left\| \boldsymbol{\Delta}_k^\top \boldsymbol{U}_k \boldsymbol{U}_k^\top \boldsymbol{\Delta}_k \right\|, \tag{A.93}$$

where the first line uses Lemma 7, and the second line uses Assumption 2 to simplify the residual as above. We have meanwhile from the definition of $E^\star$

$$\left\| \boldsymbol{\Delta}_k^\top \boldsymbol{U}_k \boldsymbol{U}_k^\top \boldsymbol{\Delta}_k \right\| \leq \left\| \boldsymbol{\Delta}_k \right\|^2 \lesssim \sigma^2 \left( 1 + \sqrt{N/d} \right), \tag{A.94}$$

because $\boldsymbol{U}_k \boldsymbol{U}_k^\top$ is an orthogonal projection, and again using Assumption 2 to simplify the residual. We can argue analogously to simplify the other term in the maximum appearing above, and this yields

$$\left\| \boldsymbol{A}_k(\beta^{-1}\boldsymbol{I} + \boldsymbol{A}_k^\top \boldsymbol{A}_k)^{-1}\boldsymbol{\Delta}_k^\top \boldsymbol{U}_k \boldsymbol{U}_k^\top \begin{bmatrix} \boldsymbol{\Delta}_1 & \dots & \boldsymbol{\Delta}_K \end{bmatrix} \boldsymbol{D}_k^{-1} \right\| \tag{A.95}$$

$$\lesssim \sqrt{K}\beta\sigma^2 \left( 1 + C\sqrt{N/d} \right) \left( 1 + \sqrt{N/d} \right), \tag{A.96}$$

where we used the fact that $\varepsilon \leq 1$ and the rest of Assumption 2 implies that $\beta \geq 1$. This residual simplifies using Assumption 2 to

$$\left\| \boldsymbol{A}_k(\beta^{-1}\boldsymbol{I} + \boldsymbol{A}_k^\top \boldsymbol{A}_k)^{-1}\boldsymbol{\Delta}_k^\top \boldsymbol{U}_k \boldsymbol{U}_k^\top \begin{bmatrix} \boldsymbol{\Delta}_1 & \dots & \boldsymbol{\Delta}_K \end{bmatrix} \boldsymbol{D}_k^{-1} \right\| \lesssim \sqrt{K}\beta\sigma^2 \left( 1 + C\sqrt{N/d} \right). \tag{A.97}$$

Next, we examine the last term, which is the other summand arising in the definition of $\Xi_k$. We have

$$
\begin{bmatrix} \mathbf{0} & \dots & \boldsymbol{A}_k(\beta^{-1}\boldsymbol{I}+\boldsymbol{A}_k^\top\boldsymbol{A}_k)^{-1} & \dots & \mathbf{0} \end{bmatrix}
\begin{bmatrix}
\mathbf{0} & \dots & \boldsymbol{\Delta}_1^\top\boldsymbol{U}_k\boldsymbol{A}_k & \dots & \mathbf{0} \\
\vdots & & \vdots & & \vdots \\
\boldsymbol{A}_k^\top\boldsymbol{U}_k^\top\boldsymbol{\Delta}_1 & \dots & \boldsymbol{\Delta}_k^\top\boldsymbol{U}_k\boldsymbol{A}_k+\boldsymbol{A}_k^\top\boldsymbol{U}_k^\top\boldsymbol{\Delta}_k & \dots & \boldsymbol{A}_k^\top\boldsymbol{U}_k^\top\boldsymbol{\Delta}_K \\
\vdots & & \vdots & & \vdots \\
\mathbf{0} & \dots & \boldsymbol{\Delta}_K^\top\boldsymbol{U}_k\boldsymbol{A}_k & \dots & \mathbf{0}
\end{bmatrix}
\tag{A.98}
$$

$$
= \boldsymbol{A}_k(\beta^{-1}\boldsymbol{I}+\boldsymbol{A}_k^\top\boldsymbol{A}_k)^{-1}\begin{bmatrix} \boldsymbol{A}_k^\top\boldsymbol{U}_k^\top\boldsymbol{\Delta}_1 & \dots & (\boldsymbol{\Delta}_k^\top\boldsymbol{U}_k\boldsymbol{A}_k+\boldsymbol{A}_k^\top\boldsymbol{U}_k^\top\boldsymbol{\Delta}_k) & \dots & \boldsymbol{A}_k^\top\boldsymbol{U}_k^\top\boldsymbol{\Delta}_K \end{bmatrix}. \tag{A.99}
$$

Now multiplying on the right by $\boldsymbol{D}_k^{-1}$ gives the term (again ignoring the right-multiplication by $\boldsymbol{\Pi}$, which does not affect the operator norm)

$$
\boldsymbol{A}_k(\beta^{-1}\boldsymbol{I}+\boldsymbol{A}_k^\top\boldsymbol{A}_k)^{-1}\begin{bmatrix} \beta\boldsymbol{A}_k^\top\boldsymbol{U}_k^\top\boldsymbol{\Delta}_1 & \dots & (\boldsymbol{\Delta}_k^\top\boldsymbol{U}_k\boldsymbol{A}_k+\boldsymbol{A}_k^\top\boldsymbol{U}_k^\top\boldsymbol{\Delta}_k)(\beta^{-1}\boldsymbol{I}+\boldsymbol{A}_k^\top\boldsymbol{A}_k)^{-1} & \dots & \beta\boldsymbol{A}_k^\top\boldsymbol{U}_k^\top\boldsymbol{\Delta}_K \end{bmatrix}. \tag{A.100}
$$

We will argue that this term is close to the term

$$
\boldsymbol{A}_k(\beta^{-1}\boldsymbol{I}+\boldsymbol{A}_k^\top\boldsymbol{A}_k)^{-1}\begin{bmatrix} \beta\boldsymbol{A}_k^\top\boldsymbol{U}_k^\top\boldsymbol{\Delta}_1 & \dots & \mathbf{0} & \dots & \beta\boldsymbol{A}_k^\top\boldsymbol{U}_k^\top\boldsymbol{\Delta}_K \end{bmatrix} \tag{A.101}
$$

in operator norm. The argument is similar to the preceding arguments: for this, it suffices to bound

$$
\left\| \begin{bmatrix} \mathbf{0} & \dots & \boldsymbol{A}_k(\beta^{-1}\boldsymbol{I}+\boldsymbol{A}_k^\top\boldsymbol{A}_k)^{-1}(\boldsymbol{\Delta}_k^\top\boldsymbol{U}_k\boldsymbol{A}_k+\boldsymbol{A}_k^\top\boldsymbol{U}_k^\top\boldsymbol{\Delta}_k)(\beta^{-1}\boldsymbol{I}+\boldsymbol{A}_k^\top\boldsymbol{A}_k)^{-1} & \dots & \mathbf{0} \end{bmatrix} \right\|, \tag{A.102}
$$

which is the same as controlling the operator norm of the nonzero block. Using submultiplicativity and Lemma 7 along with the simplifications we have done above leveraging Assumption 2, we obtain

$$
\left\| \boldsymbol{A}_k(\beta^{-1}\boldsymbol{I}+\boldsymbol{A}_k^\top\boldsymbol{A}_k)^{-1}(\boldsymbol{\Delta}_k^\top\boldsymbol{U}_k\boldsymbol{A}_k+\boldsymbol{A}_k^\top\boldsymbol{U}_k^\top\boldsymbol{\Delta}_k)(\beta^{-1}\boldsymbol{I}+\boldsymbol{A}_k^\top\boldsymbol{A}_k)^{-1} \right\| \tag{A.103}
$$

$$
\leq \left(1+C\sqrt{N/d}\right)\left\| \boldsymbol{\Delta}_k^\top\boldsymbol{U}_k\boldsymbol{A}_k+\boldsymbol{A}_k^\top\boldsymbol{U}_k^\top\boldsymbol{\Delta}_k \right\|. \tag{A.104}
$$

Meanwhile, on $E^\star$ we have the operator norm of $\boldsymbol{\Delta}_k$ and $\boldsymbol{A}_k$ controlled, using again (A.79). Applying then the triangle inequality and submultiplicativity, we obtain

$$
\left\| \boldsymbol{\Delta}_k^\top\boldsymbol{U}_k\boldsymbol{A}_k+\boldsymbol{A}_k^\top\boldsymbol{U}_k^\top\boldsymbol{\Delta}_k \right\| \lesssim \sigma\left(1+\sqrt{N/d}\right), \tag{A.105}
$$

again simplifying with Assumption 2. This shows that (A.100) is close to (A.101) with deviations of the order $\lesssim \sigma(1+\sqrt{N/d})$.

**Aggregating the previous results.** Combining our perturbation analysis above, we have established control

$$
\left\| \boldsymbol{M}_k - \left[ \left(\boldsymbol{I}-\boldsymbol{A}_k(\beta^{-1}\boldsymbol{I}+\boldsymbol{A}_k^\top\boldsymbol{A}_k)^{-1}\boldsymbol{A}_k^\top\right)\boldsymbol{U}_k^\top\begin{bmatrix} \beta\boldsymbol{\Delta}_1 & \dots & \mathbf{0} & \dots & \beta\boldsymbol{\Delta}_K \end{bmatrix} \right.\right. \tag{A.106}
$$

$$
\left.\left. + \begin{bmatrix} \mathbf{0} & \dots & \boldsymbol{A}_k(\beta^{-1}\boldsymbol{I}+\boldsymbol{A}_k^\top\boldsymbol{A}_k)^{-1} & \dots & \mathbf{0} \end{bmatrix}\boldsymbol{\Pi} \right] \right\| \tag{A.107}
$$

$$
\lesssim \sigma(1+\sqrt{N/d}) + \sqrt{K}\beta\sigma^2(1+\sqrt{N/d}). \tag{A.108}
$$

It is convenient to include one additional stage of simplification here: namely, we use Lemma 7 once more to simplify the second term in the nominal value of $\boldsymbol{M}_k$ appearing here. Namely, we have (arguing as we have above, once again)

$$
\left\| \begin{bmatrix} \mathbf{0} & \dots & \boldsymbol{A}_k(\beta^{-1}\boldsymbol{I}+\boldsymbol{A}_k^\top\boldsymbol{A}_k)^{-1} & \dots & \mathbf{0} \end{bmatrix} - \begin{bmatrix} \mathbf{0} & \dots & \frac{1}{1+\beta^{-1}}\boldsymbol{A}_k & \dots & \mathbf{0} \end{bmatrix} \right\| \tag{A.109}
$$

$$
= \left\| \frac{1}{1+\beta^{-1}}\boldsymbol{A}_k - \boldsymbol{A}_k\left(\beta^{-1}\boldsymbol{I}+\boldsymbol{A}_k^\top\boldsymbol{A}_k\right)^{-1} \right\| \tag{A.110}
$$

$$
\lesssim \sqrt{N/d}, \tag{A.111}
$$

from which it follows

$$\left\| M_k - \left[ \left( I - A_k (\beta^{-1} I + A_k^\top A_k)^{-1} A_k^\top \right) U_k^\top \left[ \beta \Delta_1 \quad \ldots \quad 0 \quad \ldots \quad \beta \Delta_K \right] \right. \right. \tag{A.112}$$

$$\left. \left. + \left[ 0 \quad \ldots \quad \tfrac{1}{1+\beta^{-1}} A_k \quad \ldots \quad 0 \right] \right] \Pi \right\| \tag{A.113}$$

$$\lesssim \sigma(1 + \sqrt{N/d}) + \sqrt{K} \beta \sigma^2 (1 + \sqrt{N/d}) + \sqrt{N/d}. \tag{A.114}$$

Meanwhile, recall the residual of scale $\lesssim (\sigma\beta)^2$ arising when we controlled the gradient term around $M_k$:

$$\left\| U_k^\top Z_\natural \left( \beta^{-1} I + (U_k^\top Z_\natural)^\top U_k^\top Z_\natural \right)^{-1} - M_k \right\| \le C(\beta\sigma)^2. \tag{A.115}$$

Combining these two bounds with the triangle inequality controls the gradient term around its nominal value. Now, we sum these errors over $k$ (again with the triangle inequality) to obtain control of the aggregate gradient around its nominal value. We introduce notation to concisely capture the accumulations of the (approximate) orthogonal projections arising in the nominal value of the main term: for each $k \in [K]$, define

$$\mathcal{P}_k = \sum_{k' \ne k} U_{k'} \left( I - A_{k'} (\beta^{-1} I + A_{k'}^\top A_{k'})^{-1} A_{k'}^\top \right) U_{k'}^\top, \tag{A.116}$$

and define an overall (approximate) projection operator (which acts on block matrices partitioned compatibly with the class sizes $N_k$, as in (A.35)) by

$$\mathcal{P}_{U_{[K]}} = \left[ \mathcal{P}_1 \quad \ldots \quad \mathcal{P}_K \right]. \tag{A.117}$$

Then the above argument implies

$$\left\| \nabla_Z R^c(Z_\natural \mid U_{[K]}) - \mathcal{P}_{U_{[K]}}(\beta \Delta \Pi^\top) \Pi - \frac{1}{1+\beta^{-1}} X_\natural \right\| \tag{A.118}$$

$$\lesssim K \left( \sigma^2 \beta^2 + \sigma(1 + \sqrt{N/d}) + \sqrt{K} \beta \sigma^2 (1 + \sqrt{N/d}) + \sqrt{N/d} \right), \tag{A.119}$$

which is enough to conclude.

$\square$

### A.2.1 KEY AUXILIARY LEMMAS

In this section we state and prove two key concentration inequalities that are used in the proof of the main theorem. They rely on simpler results which will be conveyed in subsequent subsections.

**Lemma 7.** *There exist universal constants $C, C' > 0$ such that the following holds. Let $d, p, N, K \in \mathbb{N}$ be such that Assumption 2 holds. Let $A_k$, $k \in [K]$, be defined as above. Let $E^\star$ be the good event defined in (A.61). If $E^\star$ occurs, then for $k \in [K]$ we have*

$$\left\| (\beta^{-1} I + A_k^\top A_k)^{-1} - \frac{1}{1+\beta^{-1}} I \right\| \le \frac{C\sqrt{N/d}}{(1+\beta^{-1})}. \tag{A.120}$$

*and in addition*

$$\left\| A_k (\beta^{-1} I + A_k^\top A_k)^{-1} - \frac{1}{1+\beta^{-1}} A_k \right\| \le \frac{C'\sqrt{N/d}}{(1+\beta^{-1})}. \tag{A.121}$$

*Proof.* Since $E^\star$ holds, for all $k \in [K]$ we have

$$\|A_k\| \le 1 + C_1 \sqrt{N/d}, \qquad \|A_k^\top A_k - I\| \le C_2 \sqrt{N/d}. \tag{A.122}$$

By Assumption 2, we have $\|A_k^\top A_k - I\| < 1$, so $A_k^\top A_k$ is well-conditioned. Write

$$\Xi \doteq A_k^\top A_k - I, \tag{A.123}$$

so that

$$(\beta^{-1}\boldsymbol{I} + \boldsymbol{A}_k^\top \boldsymbol{A}_k)^{-1} = ((1+\beta^{-1})\boldsymbol{I} + \boldsymbol{\Xi})^{-1} \tag{A.124}$$

$$= \frac{1}{1+\beta^{-1}}\left(\boldsymbol{I} + \frac{1}{1+\beta^{-1}}\boldsymbol{\Xi}\right)^{-1} \tag{A.125}$$

$$= \frac{1}{1+\beta^{-1}}\sum_{j=0}^{\infty}\left(-\frac{1}{1+\beta^{-1}}\right)^j \boldsymbol{\Xi}^j \tag{A.126}$$

$$= \frac{1}{1+\beta^{-1}}\boldsymbol{I} + \frac{1}{1+\beta^{-1}}\sum_{j=1}^{\infty}\left(-\frac{1}{1+\beta^{-1}}\right)^j \boldsymbol{\Xi}^j \tag{A.127}$$

by the Neumann series. This gives us

$$\left\|(\beta^{-1}\boldsymbol{I} + \boldsymbol{A}_k^\top \boldsymbol{A}_k)^{-1} - \frac{1}{1+\beta^{-1}}\boldsymbol{I}\right\| = \left\|\frac{1}{1+\beta^{-1}}\sum_{j=1}^{\infty}\left(-\frac{1}{1+\beta^{-1}}\right)^j \boldsymbol{\Xi}^j\right\| \tag{A.128}$$

$$\leq \frac{1}{1+\beta^{-1}}\sum_{j=1}^{\infty}\left(\frac{1}{1+\beta^{-1}}\right)^j \|\boldsymbol{\Xi}\|^j \tag{A.129}$$

$$\leq \frac{1}{1+\beta^{-1}}\sum_{j=1}^{\infty}\left(\frac{C_2\sqrt{N/d}}{1+\beta^{-1}}\right)^j \tag{A.130}$$

$$= \frac{C_2\sqrt{N/d}}{(1+\beta^{-1})(1+\beta^{-1} - C_2\sqrt{N/d})}. \tag{A.131}$$

Meanwhile, by Assumption 2, it holds

$$C_2\sqrt{N/d} \leq \sqrt{1/6}, \tag{A.132}$$

so it follows

$$\frac{C_2\sqrt{N/d}}{1+\beta^{-1} - C_2\sqrt{N/d}} \leq 2C_2\sqrt{N/d}. \tag{A.133}$$

By the submultiplicativity of the operator norm, we thus have

$$\left\|\boldsymbol{A}_k(\beta^{-1}\boldsymbol{I} + \boldsymbol{A}_k^\top \boldsymbol{A}_k)^{-1} - \frac{1}{1+\beta^{-1}}\boldsymbol{A}_k\right\| \leq \|\boldsymbol{A}_k\|\left\|(\beta^{-1}\boldsymbol{I} + \boldsymbol{A}_k^\top \boldsymbol{A}_k)^{-1} - \frac{1}{1+\beta^{-1}}\boldsymbol{I}\right\| \tag{A.134}$$

$$\leq \frac{[1 + C_1\sqrt{N/d}]C_2\sqrt{N/d}}{(1+\beta^{-1})(1+\beta^{-1} - C_2\sqrt{N/d})} \tag{A.135}$$

$$\leq 2\frac{[1 + C_1\sqrt{N/d}]C_2\sqrt{N/d}}{1+\beta^{-1}} \tag{A.136}$$

$$= 2\frac{C_2\sqrt{N/d} + C_1 C_2 N/d}{1+\beta^{-1}}. \tag{A.137}$$

By Assumption 2, we have that there exists some absolute constant $C_3 > 0$ with $C_3 \cdot N/d \leq \sqrt{N/d}$, which gives

$$\left\|\boldsymbol{A}_k(\beta^{-1}\boldsymbol{I} + \boldsymbol{A}_k^\top \boldsymbol{A}_k)^{-1} - \frac{1}{1+\beta^{-1}}\boldsymbol{A}_k\right\| \leq 2\frac{(C_2 + C_1 C_2 C_3^{-1})\sqrt{N/d}}{1+\beta^{-1}}, \tag{A.138}$$

as desired. $\qquad\square$

**Lemma 8.** *There exist universal constants $C_1, C_2 > 0$ such that the following holds. Let $d, p, N, K \in \mathbb{N}$ be such that Assumption 2 holds. Let $\boldsymbol{A}_k$, $k \in [K]$, be defined as above. Let $\boldsymbol{D}_k$ be defined as in (A.54). Let $\boldsymbol{\Xi}_k$ be defined as in (A.55). Let $E^\star$ be the good event defined in (A.61). If $E^\star$ occurs, then for $k \in [K]$ we have*

$$\|\boldsymbol{\Xi}_k \boldsymbol{D}_k\|^{-1} \leq C_1\beta[\sigma^2 + \sigma(C_2 + \sqrt{N/d})]. \tag{A.139}$$

*Proof.* Since we have

$$
\boldsymbol{D}_k = \begin{bmatrix} \beta^{-1}\boldsymbol{I} & & & & \\ & \ddots & & & \\ & & \beta^{-1}\boldsymbol{I} + \boldsymbol{A}_k^\top \boldsymbol{A}_k & & \\ & & & \ddots & \\ & & & & \beta^{-1}\boldsymbol{I} \end{bmatrix} \tag{A.140}
$$

it holds that

$$
\boldsymbol{D}_k^{-1} = \begin{bmatrix} \beta\boldsymbol{I} & & & & \\ & \ddots & & & \\ & & (\beta^{-1}\boldsymbol{I} + \boldsymbol{A}_k^\top \boldsymbol{A}_k)^{-1} & & \\ & & & \ddots & \\ & & & & \beta\boldsymbol{I} \end{bmatrix}. \tag{A.141}
$$

We will use the straightforward estimate $\|\boldsymbol{\Xi}_k \boldsymbol{D}_k^{-1}\| \le \|\boldsymbol{\Xi}_k\|\|\boldsymbol{D}_k^{-1}\|$ and bound the two matrices' operator norms individually. By the previous expression,

$$
\|\boldsymbol{D}_k^{-1}\| = \max\{\beta, \|(\beta^{-1}\boldsymbol{I} + \boldsymbol{A}_k^\top \boldsymbol{A}_k)^{-1}\|\} \le \beta, \tag{A.142}
$$

because $\boldsymbol{A}_k^\top \boldsymbol{A}_k \succeq \boldsymbol{0}$, so we need only control the operator norm of $\boldsymbol{\Xi}_k$. To this end, note the convenient expression

$$
\boldsymbol{\Xi}_k = \boldsymbol{\Pi}\boldsymbol{\Delta}^\top U_k U_k^\top \boldsymbol{\Delta}\boldsymbol{\Pi}^\top + 2\,\mathrm{sym}\left((\boldsymbol{\Delta}\boldsymbol{\Pi}^\top)^\top [\boldsymbol{0} \quad \ldots \quad U_k \boldsymbol{A}_k \quad \ldots \quad \boldsymbol{0}]\right), \tag{A.143}
$$

where $\mathrm{sym}(\,\cdot\,)$ denotes the symmetric part operator. By the triangle inequality, the operator norm of $\boldsymbol{\Xi}_k$ is no larger than the sum of the operator norms of each term in the previous expression. The operator norm of the first term is no larger than $\|\boldsymbol{\Delta}\|^2$, because $\boldsymbol{\Pi}$ is a permutation matrix and $\boldsymbol{U}U_k^\top$ is an orthogonal projection. Meanwhile, using that the symmetric part operator is the orthogonal projection onto the space of symmetric matrices, it follows

$$
\left\|2\,\mathrm{sym}\left((\boldsymbol{\Delta}\boldsymbol{\Pi}^\top)^\top [\boldsymbol{0} \quad \ldots \quad U_k \boldsymbol{A}_k \quad \ldots \quad \boldsymbol{0}]\right)\right\| \le 2\left\|(\boldsymbol{\Delta}\boldsymbol{\Pi}^\top)^\top [\boldsymbol{0} \quad \ldots \quad U_k \boldsymbol{A}_k \quad \ldots \quad \boldsymbol{0}]\right\|, \tag{A.144}
$$

and then we find as above that the RHS is no larger than $2\|\boldsymbol{\Delta}\|\|\boldsymbol{A}_k\|$. Since the good event $E^\star$ defined in (A.61) holds by assumption, we have that there are constants $C_1, C_2 > 0$ such that

$$
\|\boldsymbol{\Delta}\| \le \sigma\left(C_1 + \sqrt{\frac{N}{d}}\right) \tag{A.145}
$$

$$
\|\boldsymbol{A}_k\| \le 1 + C_2\sqrt{\frac{N}{d}}. \tag{A.146}
$$

By Assumption 2 we have $d \ge N$, so that $\sqrt{N/d} \le 1$. Therefore we have

$$
\|\boldsymbol{\Delta}\| \le \sigma\left(C_1 + 1\right) = C_3\sigma \tag{A.147}
$$

for $C_3 \doteq C_1 + 1$ another universal constant. Thus on this good event we have

$$
2\|\boldsymbol{\Delta}\|\|\boldsymbol{A}_k\| \le C_3\sigma\left(1 + C_2\sqrt{N/d}\right). \tag{A.148}
$$

Therefore, we have

$$
\|\boldsymbol{\Xi}_k\| \le \|\boldsymbol{\Delta}\|^2 + 2\|\boldsymbol{\Delta}\|\|\boldsymbol{A}_k\| \tag{A.149}
$$

$$
\le C_3^2\sigma^2 + C_3\sigma(1 + C_2\sqrt{N/d}) \tag{A.150}
$$

$$
\le C_4[\sigma^2 + \sigma(1 + C_2\sqrt{N/d})] \tag{A.151}
$$

where $C_4 = \max\{C_3, C_3^2\}$ is another universal constant. Thus on $E^\star$ we have

$$
\|\boldsymbol{\Xi}_k \boldsymbol{D}_k^{-1}\| \le C_4\beta[\sigma^2 + \sigma(1 + C_2\sqrt{N/d})] \le C_5\beta[\sigma^2 + \sigma(1 + \sqrt{N/d})] \tag{A.152}
$$

for $C_5 > 0$ another obvious universal constant. $\qquad\square$

### A.2.2 CONCENTRATION INEQUALITIES FOR OUR SETTING

In this section we prove some simple concentration inequalities that are adapted to the problem setting. These results are used to prove the key lemmata above, and indeed are also invoked in the main theorem. They follow from even simpler concentration inequalities that are abstracted away from the problem setting, which we discuss in the following subsections.

**Proposition 9.** *There are universal constants* $C_1, C_2, C_3 > 0$ *such that the following holds. Let* $d, N \in \mathbb{N}$ *be such that Assumption 2 holds. Let* $\mathbf{\Delta} \in \mathbb{R}^{d \times N}$ *be defined as above. Then*

$$\mathbb{P}\left[\|\mathbf{\Delta}\| > \sigma\left(C_1 + \sqrt{\frac{N}{d}}\right)\right] \leq C_2 e^{-C_3 d}. \tag{A.153}$$

*Proof.* We use Proposition 14 with the parameters $q = d$, $n = N$, and $x = \sigma/\sqrt{d}$, which obtains

$$\mathbb{P}[\|\mathbf{\Delta}\| > s] \leq C_1 \exp\left(-d\left\{\frac{s\sqrt{d}/\sigma - \sqrt{N}}{C_2\sqrt{d}} - 1\right\}^2\right), \qquad \forall s > \frac{\sigma}{\sqrt{d}}(\sqrt{N} + C_2\sqrt{d}) \tag{A.154}$$

Notice that we have

$$\frac{s\sqrt{d}/\sigma - \sqrt{N}}{C_2\sqrt{d}} - 1 = \frac{1}{C_2}\left(\frac{s}{\sigma} - \sqrt{\frac{N}{d}}\right) - 1, \qquad \frac{\sigma}{\sqrt{d}}(\sqrt{N} + C_2\sqrt{d}) = \sigma\left(\sqrt{\frac{N}{d}} + C_2\right). \tag{A.155}$$

To make the squared term equal to 1, we pick

$$s = \sigma\left(\sqrt{\frac{N}{d}} + 2C_2\right), \tag{A.156}$$

which gives

$$\mathbb{P}\left[\|\mathbf{\Delta}\| > \sigma\left(2C_2 + \sqrt{\frac{N}{d}}\right)\right] \leq C_2 e^{-d}. \tag{A.157}$$

$\square$

**Proposition 10.** *There are universal constants* $C_1, C_2, C_3, C_4 > 0$ *such that the following holds. Let* $p, N, K \in \mathbb{N}$ *be such that Assumption 2 holds. Let* $\mathbf{A}_k$, $k \in [K]$, *be defined as above. Then*

$$\mathbb{P}\left[\|\mathbf{A}_k\| > 1 + C_1\sqrt{\frac{N}{d}}\right] \leq C_2 \exp\left(-C_3\frac{N}{K}\right) + \frac{C_4}{N^2} \tag{A.158}$$

*Proof.* By Propositions 12 and 14 with parameters $n = n$, $k = K$, $q = p$, and $x = 1/\sqrt{p}$, if we define

$$N_{\min} \doteq \left\lfloor \frac{N}{K} - C_1\sqrt{N \log N} \right\rfloor, \qquad N_{\max} \doteq \left\lceil \frac{N}{K} + C_1\sqrt{N \log N} \right\rceil \tag{A.159}$$

then we have

$$\mathbb{P}[\|\mathbf{A}_k\| > s] \leq \sum_{n=N_{\min}}^{N_{\max}} \mathbb{P}[\|\mathbf{A}_k\| > s \mid K_k = n]\mathbb{P}[K_k = n] + \frac{C_2}{N^2} \tag{A.160}$$

$$\leq \sum_{n=N_{\min}}^{N_{\max}} C_3 \exp\left(-n\left\{\frac{s\sqrt{p} - \sqrt{p}}{C_4\sqrt{4}} - 1\right\}^2\right)\mathbb{P}[K_k = n] + \frac{C_2}{N^2}, \tag{A.161}$$

for all $s$ obeying

$$s \geq \frac{1}{\sqrt{p}}\left(\sqrt{p} + C_4\sqrt{N_{\max}}\right) \tag{A.162}$$

$$= 1 + C_4\sqrt{\frac{N_{\max}}{p}}. \tag{A.163}$$

Thus we have that the concentration holds for all $s$ obeying

$$s \geq 1 + C_4 \sqrt{\frac{N_{\max}}{p}}. \tag{A.164}$$

In order to cancel out the most interior terms, we choose

$$s = 1 + 2C_4 \sqrt{\frac{N_{\max}}{p}}. \tag{A.165}$$

This choice obtains

$$\mathbb{P}[\|\boldsymbol{A}_k\| > s] \leq \sum_{n=N_{\min}}^{N_{\max}} C_3 \exp\left(-n \left\{ \frac{s\sqrt{p} - \sqrt{p}}{C_4\sqrt{n}} - 1 \right\}^2\right) \mathbb{P}[K_k = n] + \frac{C_2}{N^2} \tag{A.166}$$

$$= \sum_{n=N_{\min}}^{N_{\max}} C_3 \exp\left(-n \left\{ 2 \underbrace{\sqrt{\frac{N_{\max}}{n}}}_{\geq 1} - 1 \right\}^2\right) \mathbb{P}[K_k = n] + \frac{C_2}{N^2} \tag{A.167}$$

$$\leq \sum_{n=N_{\min}}^{N_{\max}} C_3 \exp(-n) \mathbb{P}[K_k = n] + \frac{C_2}{N^2} \tag{A.168}$$

$$\leq \sum_{n=N_{\min}}^{N_{\max}} C_3 \exp(-N_{\min}) \mathbb{P}[K_k = n] + \frac{C_2}{N^2} \tag{A.169}$$

$$\leq C_3 \exp(-N_{\min}) + \frac{C_2}{N^2} \tag{A.170}$$

$$= C_3 \exp\left(-\frac{N}{K} + C_1\sqrt{N \log N}\right) + \frac{C_2}{N^2} \tag{A.171}$$

$$\leq C_3 \exp\left(-\frac{N}{K} + \frac{1}{2}\sqrt{N \log N}\right) + \frac{C_2}{N^2} \tag{A.172}$$

$$\leq C_3 \exp\left(-\frac{1}{2} \cdot \frac{N}{K}\right) + \frac{C_2}{N^2}. \tag{A.173}$$

To obtain the conclusion of the theorem, note that any $s$ such that

$$s \geq 1 + 2C_4 \sqrt{\frac{N_{\max}}{p}} = 1 + 2C_4 \sqrt{\frac{N/K}{p} + C_1 \frac{\sqrt{N \log N}}{p}} = 1 + 2C_4 \sqrt{\frac{N}{d} + C_1 \frac{\sqrt{N \log N}}{p}} \tag{A.174}$$

enjoys the same high-probability bound. By Assumption 2, we have

$$1 + 2C_4 \sqrt{\frac{N}{d} + C_1 \frac{\sqrt{N \log N}}{p}} \leq 1 + 2C_4 \sqrt{\frac{N}{d} + \frac{1}{2} \cdot \frac{N/K}{p}} \tag{A.175}$$

$$= 1 + 2C_4 \sqrt{\frac{N}{d} + \frac{1}{2} \cdot \frac{N}{d}} = 1 + 2C_4 \sqrt{\frac{3}{2}} \cdot \sqrt{\frac{N}{d}} \tag{A.176}$$

whence the ultimate conclusion is obtained by combining constants. $\square$

**Proposition 11.** *There are universal constants $C_1, C_2, C_3, C_4 > 0$ such that the following holds. Let $p, N, K \in \mathbb{N}$ be such that Assumption 2 holds. Let $\boldsymbol{A}_k$, $k \in [K]$, be defined as above. Then*

$$\mathbb{P}\left[\|\boldsymbol{A}_k^\top \boldsymbol{A}_k - \boldsymbol{I}\| > C_1 \sqrt{\frac{N}{d}}\right] \leq C_2 \exp\left(-C_3 \frac{N}{K}\right) + \frac{C_4}{N^2}. \tag{A.177}$$

*Proof.* By Propositions 12 and 15 with parameters $n = n$, $k = K$, $q = p$, and $x = 1/\sqrt{p}$, if we define

$$N_{\min} \doteq \left\lfloor \frac{N}{K} - C_1 \sqrt{N \log N} \right\rfloor, \qquad N_{\max} \doteq \left\lceil \frac{N}{K} + C_1 \sqrt{N \log N} \right\rceil \tag{A.178}$$

then we have

$$\mathbb{P}[\|\boldsymbol{A}_k^\top \boldsymbol{A}_k - \boldsymbol{I}\| > s] \tag{A.179}$$

$$\leq \sum_{n=N_{\min}}^{N_{\max}} \mathbb{P}[\|\boldsymbol{A}_k^\top \boldsymbol{A}_k - \boldsymbol{I}\| > s \mid K_k = n]\mathbb{P}[K_k = n] + \frac{C_2}{N^2} \tag{A.180}$$

$$\leq \frac{C_2}{N^2} + \sum_{n=N_{\min}}^{N_{\max}} \mathbb{P}[K_k = n] \cdot \begin{cases} C_3 \exp\left(-n \left\{ \frac{1}{C_4^2 C_5 \sqrt{n/p}} s - 1 \right\}^2 \right), & \text{if } C_4^2 C_5 \sqrt{n/p} \leq s \leq C_4^2 \\ C_3 \exp\left(-n \left\{ \frac{1}{C_4 C_5 \sqrt{n/p}} \sqrt{s} - 1 \right\}^2 \right), & \text{if } s \geq C_4^2. \end{cases} \tag{A.181}$$

In order to cancel the most terms, we choose

$$s = 2C_4^2 C_5 \sqrt{\frac{N_{\max}}{p}}. \tag{A.182}$$

In order to assure ourselves that this choice still has $s \leq C_4^2$ (so that we can use the first case for all $n$), we have

$$s = 2C_4^2 C_5 \sqrt{\frac{N_{\max}}{p}} \tag{A.183}$$

$$= 2C_4^2 C_5 \sqrt{\frac{N/K + C_1 \sqrt{N \log N}}{p}} \tag{A.184}$$

$$= 2C_4^2 C_5 \sqrt{\frac{N/K + \frac{1}{2} N/K}{p}} \tag{A.185}$$

$$= 2\sqrt{\frac{3}{2}} C_4^2 C_5 \cdot \sqrt{\frac{N/K}{p}} \tag{A.186}$$

$$= \sqrt{6} C_4^2 C_5 \cdot \sqrt{\frac{N}{d}} \tag{A.187}$$

$$\leq C_4^2 \quad \text{when} \quad \sqrt{6} C_5 \sqrt{\frac{N}{d}} \leq 1. \tag{A.188}$$

Of course, this condition is assured by Assumption 2. Now that we have this, we know $s$ falls in the first, and so we have

$$\mathbb{P}\left[\|\boldsymbol{A}_k^\top \boldsymbol{A}_k - \boldsymbol{I}\| > C_1 \sqrt{\frac{N}{d}}\right] \tag{A.189}$$

$$\leq \frac{C_2}{N^2} + \sum_{n=N_{\min}}^{N_{\max}} \mathbb{P}[K_k = n] \cdot \begin{cases} C_3 \exp\left(-n \left\{ \frac{1}{C_4^2 C_5 \sqrt{n/p}} s - 1 \right\}^2 \right), & \text{if } C_4^2 C_5 \sqrt{n/p} \leq s \leq C_4^2 \\ C_3 \exp\left(-n \left\{ \frac{1}{C_4 C_5 \sqrt{n/p}} \sqrt{s} - 1 \right\}^2 \right), & \text{if } s \geq C_4^2 \end{cases} \tag{A.190}$$

$$\leq \frac{C_2}{N^2} + \sum_{n=N_{\min}}^{N_{\max}} C_3 \exp\left(-n \left\{ \frac{2C_4^2 C_5 \sqrt{N_{\max}/p}}{C_4^2 C_5 \sqrt{n/p}} - 1 \right\}^2 \right) \mathbb{P}[K_k = n] \tag{A.191}$$

$$= \frac{C_2}{N^2} + \sum_{n=N_{\min}}^{N_{\max}} C_3 \exp\left(-n \left\{ 2\sqrt{\frac{N_{\max}}{n}} - 1 \right\}^2 \right) \mathbb{P}[K_k = n] \tag{A.192}$$

$$\leq C_3 \exp\left(-\frac{1}{2} \cdot \frac{N}{K}\right) + \frac{C_2}{N^2} \tag{A.193}$$

where the last inequality follows from the exact same argument as in Proposition 10. $\square$

### A.2.3 GENERIC CONCENTRATION INEQUALITIES

In this subsection we prove the base-level concentration inequalities used throughout the proofs in this paper.

**Binomial concentration.**

**Proposition 12.** *There exist universal constants $C_1, C_2 > 0$ such that the following holds. Let $n, k \in \mathbb{Z}$. For each $i \in [k]$, let $B_i \sim \mathrm{Bin}(n, 1/k)$, such that the $B_i$ are identically (marginally) distributed but not necessarily independent binomial random variables. Let $E$ be an event. Then for any $i \in [k]$, we have*

$$\mathbb{P}[E] \leq \sum_{b=\lfloor n/k - C_1\sqrt{n\log n}\rfloor}^{\lceil n/k + C_1\sqrt{n\log n}\rceil} \mathbb{P}[E \mid B_i = b]\mathbb{P}[B_i = b] + \frac{C_2}{n^2}. \tag{A.194}$$

*Proof.* We have

$$\mathbb{P}[E] = \mathbb{E}[\mathbb{E}[E \mid B_i]] = \sum_{b=0}^{n} \mathbb{P}[E \mid B_i = b]\mathbb{P}[B_i = b]. \tag{A.195}$$

Each $B_i$ is unconditionally distributed as $\mathrm{Bin}(n, 1/k)$. By union bound and Hoeffding's inequality (Vershynin, 2018, Theorem 2.2.6), we have

$$\mathbb{P}[|B_i - n/k| \geq t] \leq 2\exp\left(-\frac{2t^2}{n}\right). \tag{A.196}$$

Inverting this inequality obtains that there exists some (simple) universal constants $C_1, C_2 > 0$ such that

$$\mathbb{P}\left[|B_i - n/k| \geq C_1\sqrt{n\log n}\right] \leq \frac{C_2}{n^3}. \tag{A.197}$$

Thus, if we define

$$b_{\min} \doteq \left\lfloor \frac{n}{k} - C_1\sqrt{n\log n}\right\rfloor, \qquad b_{\max} \doteq \left\lceil \frac{n}{k} + C_1\sqrt{n\log n}\right\rceil, \tag{A.198}$$

then we have

$$\mathbb{P}[E] = \sum_{b=0}^{n} \mathbb{P}[E \mid B_i = b]\mathbb{P}[B_i = b] \tag{A.199}$$

$$= \sum_{b=0}^{b_{\min}-1} \underbrace{\mathbb{P}[E \mid B_i = b]}_{\leq 1}\underbrace{\mathbb{P}[B_i = b]}_{\leq C_2/n^3} + \sum_{b=b_{\max}+1}^{n} \underbrace{\mathbb{P}[E \mid B_i = b_1]}_{\leq 1}\underbrace{\mathbb{P}[B_i = b]}_{\leq C_2/n^3} \tag{A.200}$$

$$+ \sum_{b=b_{\min}}^{b_{\max}} \mathbb{P}[E \mid B_i = b]\mathbb{P}[B_i = b] \tag{A.201}$$

$$\leq \sum_{b=0}^{b_{\min}-1} \frac{C_2}{n^3} + \sum_{b=b_{\max}+1}^{n} \frac{C_2}{n^3} + \sum_{b=b_{\min}}^{b_{\max}} \mathbb{P}[E \mid B_i = b]\mathbb{P}[B_i = b] \tag{A.202}$$

$$\leq \sum_{b=0}^{n} \frac{C_2}{n^3} + \sum_{b=b_{\min}}^{b_{\max}} \mathbb{P}[E \mid B_i = b]\mathbb{P}[B_i = b] \tag{A.203}$$

$$= \frac{C_2}{n^2} + \frac{C_2}{n^3} + \sum_{b=b_{\min}}^{b_{\max}} \mathbb{P}[E \mid B_i = b]\mathbb{P}[B_i = b] \tag{A.204}$$

$$\leq \frac{2C_2}{n^2} + \sum_{b=b_{\min}}^{b_{\max}} \mathbb{P}[E \mid B_i = b]\mathbb{P}[B_i = b]. \tag{A.205}$$

$\square$

*Remark* 13. Notice that a simple adaptation of this argument can turn the additive probability $C_3/n^2$ into $C_3'/n^z$ for any positive integer $z \in \mathbb{N}$ (where $C_3'$ depends on $z$). However, trying to replace it with $C_3'e^{-C'n}$ is more difficult.

**Gaussian concentration.**

**Proposition 14.** *There are universal constants $C_1, C_2, C_3 > 0$ such that the following holds. Let $n, q \in \mathbb{N}$, and let $\boldsymbol{M} \in \mathbb{R}^{q \times n}$ be such that $M_{ij} \sim_{\text{i.i.d.}} \mathcal{N}(0, x^2)$. Then*

$$\mathbb{P}[\|\boldsymbol{M}\| > s] \leq C_1 \exp\left(-n\left\{\frac{s/x - \sqrt{q}}{C_2\sqrt{n}} - 1\right\}^2\right), \qquad \forall s > x\left\{\sqrt{q} + C_2\sqrt{n}\right\} \tag{A.206}$$

$$\mathbb{P}[\|\boldsymbol{M}\| > s] \leq C_1 \exp\left(-q\left\{\frac{s/x - \sqrt{n}}{C_3\sqrt{q}} - 1\right\}^2\right), \qquad \forall s > x\left\{\sqrt{n} + C_3\sqrt{q}\right\}. \tag{A.207}$$

*Proof.* Define $\overline{\boldsymbol{M}} \doteq \frac{1}{x}\boldsymbol{M}$, so that $M_{ij} \sim_{\text{i.i.d.}} \mathcal{N}(0,1)$. By Vershynin (2018, Example 2.5.8, Lemma 3.4.2), we see that each row $\overline{\boldsymbol{M}}_i$ has Orlicz norm $\|\overline{\boldsymbol{M}}_i\|_{\psi_2} \leq C_1$ for some universal constant $C_1 > 0$.

By Vershynin (2018, Theorem 4.6.1) we have for some other universal constant $C_2 > 0$ that for all $t > 0$,

$$\sqrt{q} - C_1^2 C_2(\sqrt{n} + t) \leq \sigma_{\min(n,q)}(\overline{\boldsymbol{M}}) \leq \sigma_1(\overline{\boldsymbol{M}}) \leq \sqrt{q} + C_1^2 C_2(\sqrt{n} + t) \tag{A.208}$$

with probability at least $1 - 2e^{-t^2}$. Defining $C_3 \doteq C_1^2 C_2$ and noting that $\|\cdot\| = \sigma_1(\cdot)$, we have with the same probability that

$$\|\overline{\boldsymbol{M}}\| - \sqrt{q} \leq C_3\left(\sqrt{n} + t\right). \tag{A.209}$$

Simplifying, we obtain

$$\|\overline{\boldsymbol{M}}\| - \sqrt{q} \leq C_3\left(\sqrt{n} + t\right) \tag{A.210}$$

$$\frac{1}{x}\|\boldsymbol{M}\| - \sqrt{q} \leq C_3\left(\sqrt{n} + t\right) \tag{A.211}$$

$$\|\boldsymbol{M}\| - x\sqrt{q} \leq C_3 x\left(\sqrt{n} + t\right) \tag{A.212}$$

$$\|\boldsymbol{M}\| \leq x\left\{\sqrt{q} + C_3\left(\sqrt{n} + t\right)\right\}. \tag{A.213}$$

Define $s > 0$ by

$$s \doteq x\left\{\sqrt{q} + C_3\left(\sqrt{n} + t\right)\right\} \iff t = \frac{s/x - \sqrt{q}}{C_3} - \sqrt{n}. \tag{A.214}$$

Note that the range of validity is

$$t > 0 \iff s > x\left\{\sqrt{q} + C_3\sqrt{n}\right\}. \tag{A.215}$$

For $s$ in this range, we have

$$\mathbb{P}[\|\boldsymbol{M}\| > s] \leq 2\exp\left(-\left\{\frac{s/x - \sqrt{q}}{C_3} - \sqrt{n}\right\}^2\right) = 2\exp\left(-n\left\{\frac{s/x - \sqrt{q}}{C_3\sqrt{n}} - 1\right\}^2\right). \tag{A.216}$$

The other inequality follows from applying this inequality to $\boldsymbol{M}^\top$. $\qquad\square$

**Proposition 15.** *There are universal constants $C_1, C_2, C_3 > 0$ such that the following holds. Let $n, q \in \mathbb{N}$, and let $\boldsymbol{M} \in \mathbb{R}^{q \times n}$ be such that $M_{ij} \sim_{\text{i.i.d.}} \mathcal{N}(0, x^2)$. Then*

$$\mathbb{P}\left[\|\boldsymbol{M}^\top\boldsymbol{M} - qx^2\boldsymbol{I}\| > s\right] \tag{A.217}$$

$$\leq \begin{cases} C_1 \exp\left(-n\left\{\frac{1}{C_2^2 C_3\sqrt{nq}x^2}s - 1\right\}^2\right), & \text{if } C_2^2 C_3\sqrt{nq}x^2 \leq s \leq C_2^2 qx^2 \\ C_1 \exp\left(-n\left\{\frac{1}{C_2 C_3\sqrt{n}x}\sqrt{s} - 1\right\}^2\right), & \text{if } s \geq C_2^2 qx^2. \end{cases} \tag{A.218}$$

*Proof.* Define $\overline{\boldsymbol{M}} \doteq \frac{1}{x}\boldsymbol{M}$, so that $\overline{M}_{ij} \sim_{\text{i.i.d.}} \mathcal{N}(0,1)$. By Vershynin (2018, Example 2.5.8, Lemma 3.4.2), we see that each row has Orlicz norm $\|\overline{\boldsymbol{M}}_i\|_{\psi_2} \leq C_1$ for some universal constant $C_1 > 0$.

By Vershynin (2018, Eq. 4.22) we have for some other universal constant $C_2 > 0$ that for all $t > 0$,

$$\left\|\frac{1}{q}\overline{\boldsymbol{M}}^\top \overline{\boldsymbol{M}} - \boldsymbol{I}\right\| \leq C_1^2 \max\{\delta, \delta^2\} \quad \text{where} \quad \delta \doteq C_2 \frac{\sqrt{n}+t}{\sqrt{q}}. \tag{A.219}$$

with probability at least $1 - 2e^{-t^2}$. Simplifying, we obtain

$$\left\|\frac{1}{q}\overline{\boldsymbol{M}}^\top \overline{\boldsymbol{M}} - \boldsymbol{I}\right\| \leq C_1^2 \max\{\delta, \delta^2\} \tag{A.220}$$

$$\left\|\overline{\boldsymbol{M}}^\top \overline{\boldsymbol{M}} - q\boldsymbol{I}\right\| \leq C_1^2 q \max\{\delta, \delta^2\} \tag{A.221}$$

$$\left\|(x^{-1}\boldsymbol{M})^\top (x^{-1}\boldsymbol{M}) - q\boldsymbol{I}\right\| \leq C_1^2 q \max\{\delta, \delta^2\} \tag{A.222}$$

$$\left\|x^{-2}\boldsymbol{M}^\top \boldsymbol{M} - q\boldsymbol{I}\right\| \leq C_1^2 q \max\{\delta, \delta^2\} \tag{A.223}$$

$$\left\|\boldsymbol{M}^\top \boldsymbol{M} - qx^2\boldsymbol{I}\right\| \leq C_1^2 qx^2 \cdot \max\{\delta, \delta^2\}. \tag{A.224}$$

Now from simple algebra and the fact that $n \geq 1$, we have

$$\max\{\delta, \delta^2\} = \delta \iff 0 \leq t \leq C_2^{-1}\sqrt{q} - \sqrt{n} \tag{A.225}$$

$$\max\{\delta, \delta^2\} = \delta^2 \iff t \geq C_2^{-1}\sqrt{q} - \sqrt{n}. \tag{A.226}$$

Now define $s \geq 0$ by

$$s \doteq C_1^2 qx^2 \cdot \max\{\delta, \delta^2\}. \tag{A.227}$$

Thus in the first case we have

$$s \doteq C_1^2 C_2 \sqrt{q}x^2(\sqrt{n}+t) \iff t = \frac{1}{C_1^2 C_2 \sqrt{q}x^2}s - \sqrt{n}, \tag{A.228}$$

and in particular the first case holds when

$$\max\{\delta, \delta^2\} = \delta \iff 0 \leq t \leq C_2^{-1}\sqrt{q} - \sqrt{n} \iff C_1^2 C_2 \sqrt{nq}x^2 \leq s \leq C_1^2 qx^2. \tag{A.229}$$

Meanwhile, in the second case, we have

$$s \doteq C_1^2 C_2^2 (\sqrt{n}+t)^2 x^2 \iff t = \frac{1}{C_1 C_2 x}\sqrt{s} - \sqrt{n} \tag{A.230}$$

where we obtain only one solution to the quadratic equation by requiring $t \geq 0$, and in particular the second case holds when

$$\max\{\delta, \delta^2\} = \delta \iff t \geq C_2^{-1}\sqrt{q} - \sqrt{n} \iff s \geq C_1^2 qx^2. \tag{A.231}$$

Thus we have

$$\mathbb{P}[\|\boldsymbol{M}^\top \boldsymbol{M} - qx^2\boldsymbol{I}\| > s] \tag{A.232}$$

$$\leq \begin{cases} 2\exp\left(-\left\{\frac{1}{C_1^2 C_2 \sqrt{q}x^2}s - \sqrt{n}\right\}^2\right), & \text{if } C_1^2 C_2 \sqrt{nq}x^2 \leq s \leq C_1^2 qx^2 \\ 2\exp\left(-\left\{\frac{1}{C_1 C_2 x}\sqrt{s} - \sqrt{n}\right\}^2\right), & \text{if } s \geq C_1^2 qx^2 \end{cases} \tag{A.233}$$

$$= \begin{cases} 2\exp\left(-n\left\{\frac{1}{C_1^2 C_2 \sqrt{nq}x^2}s - 1\right\}^2\right), & \text{if } C_1^2 C_2 \sqrt{nq}x^2 \leq s \leq C_1^2 qx^2 \\ 2\exp\left(-n\left\{\frac{1}{C_1 C_2 \sqrt{n}x}\sqrt{s} - 1\right\}^2\right), & \text{if } s \geq C_1^2 qx^2. \end{cases} \tag{A.234}$$

$\square$

### A.3 COMPANION TO SECTION 2.4

In this section, we justify the scaling applied to $\nabla R^c$ in Section 2.4 and supply the discretization scheme.

First, suppose that $\boldsymbol{Z}_\natural^\ell$ satisfies (2.1), and $\boldsymbol{Z}_t \doteq \boldsymbol{Z}_\natural^\ell + \sigma_t \boldsymbol{W}$, where $\boldsymbol{W}$ is a standard Gaussian matrix, so that $\boldsymbol{Z}_t$ satisfies (2.2) with noise level $\sigma_t > 0$. Let $q_t$ be the density of $\boldsymbol{Z}_t$. Theoretical analysis from (Lu et al., 2023) and empirical analysis from (Song & Ermon, 2019) demonstrates that under generic conditions, we have that

$$\|\nabla q_t(\boldsymbol{Z}_t)\|_2 \propto \frac{1}{\sigma_t^2}, \tag{A.235}$$

ignoring all terms in the right-hand side except for those involving $\sigma_t$. On the other hand, from the proof of Theorem 3, we obtain that $-\nabla R^c(\boldsymbol{Z}_t)$ has constant (in $\sigma_t$) magnitude with high probability. Thus, in order to have them be the same magnitude, we need to divide $-\nabla R^c(\boldsymbol{Z}_t)$ by $\sigma_t^2$ to have it be a drop-in replacement for the score function, as alluded to in Sections 2.3 and 2.4.

Second, we wish to explicitly state our discretization scheme given in Section 2.4. To wit, we provide a discretization scheme that turns the structured diffusion ODE (2.15) into its discretized analogue; the other discretization of the structured denoising ODE (2.11) occurs similarly. To begin with, define the shorthand notation

$$f(t, \boldsymbol{Y}(t)) \doteq \nabla R^c(\boldsymbol{Y}(t) \mid \boldsymbol{U}_{[K]}(T - t)), \tag{A.236}$$

so that we have

$$\mathrm{d}\boldsymbol{Y}(t) = \frac{1}{2t} f(t, \boldsymbol{Y}(t))\, \mathrm{d}t. \tag{2.11}$$

Fix $L$, and let $0 < t_1 < t_2 < \cdots < t_L = T$, such that $t_1$ is small. (These will be specified shortly in order to supply the discretization scheme.) A suitable first-order discretization is given by

$$\boldsymbol{Y}^{\ell+1} \approx \boldsymbol{Y}^\ell + \frac{t_{\ell+1} - t_\ell}{2t_\ell} f(t_\ell, \boldsymbol{Y}^\ell). \tag{A.237}$$

Thus it remains to set $t_1, \ldots, t_L$ such that

$$\frac{t_{\ell+1} - t_\ell}{2t_\ell} = \kappa \tag{A.238}$$

for some constant $\kappa$, we observe that we must set

$$t_{\ell+1} = (1 + 2\kappa)t_\ell, \tag{A.239}$$

so that the time grows exponentially in the index. The reverse process time decays exponentially in the index, which matches practical discretization schemes for ordinary diffusion models (Song & Ermon, 2019). Finally, we have $T = t_L = (1 + 2\kappa)^L t_1$, so that $t_1 = \frac{T}{(1+2\kappa)^L}$.

# B  ADDITIONAL EXPERIMENT DETAILS

## B.1  EXPERIMENT DETAILS AND CLARIFICATIONS

In all setups, as is standard practice (He et al., 2022), we append a trainable class token $z_{\text{cls}}^1 \in \mathbb{R}^d$ after masking and linear projection, namely,

$$f^{\text{pre}}(\boldsymbol{X}) = [z_{\text{cls}}^1, \boldsymbol{W}^{\text{pre}}\boldsymbol{X} + \boldsymbol{E}^{\text{pos}}]. \tag{B.1}$$

Everything else goes through with $Z$ having $N + 1$ instead of $N$ tokens, indexing the $N$ image tokens' intermediate representations as $z_i^\ell$, etc., and indexing the class token intermediate representation as $z_{\text{cls}}^\ell$. At the end, the post-processing map is

$$g^{\text{post}}(\boldsymbol{Y}^L) = \boldsymbol{W}^{\text{post}}\boldsymbol{Y}_{1:N}^L \tag{B.2}$$

where $\boldsymbol{Y}_{1:N}^L \in \mathbb{R}^{d \times N}$ is the matrix whose columns are the columns of $\boldsymbol{Y}^L$ corresponding to image tokens, i.e., the second through last columns of $\boldsymbol{Y}^L$. Thus, the class token $z_{\text{cls}}^1$ (and its representation $z_{\text{cls}}$, i.e., the first column of $\boldsymbol{Z}$) are neither masked or reconstructed.

For training using masked autoencoding, we follow a modified recipe of (He et al., 2022). We mask a fixed percentage $\mu \in [0, 1]$ of randomly selected tokens in $\boldsymbol{X}$; that is, we randomly set $(1 - \mu)N$ image tokens $\boldsymbol{x}_i$ to $\boldsymbol{0} \in \mathbb{R}^D$, obtaining $\overline{\boldsymbol{X}}$. Then $\overline{\boldsymbol{X}}$ is the input to the encoder. Unlike in (He et al., 2022), the decoder receives no special treatment, and operates on all token representations. The loss is computed only on the masked image patches. Refer to He et al. (2022) for more MAE implementation details.

For fine-tuning using supervised classification, we use the representation $z_{\text{cls}}$ of the so-far-vestigial class token as the feature used for classification. Namely, we obtain the unnormalized log-probabilities for the classes as $\boldsymbol{u} \doteq \boldsymbol{W}^{\text{head}} \text{LN}(z_{\text{cls}})$, where LN is a trainable layer-norm and $\boldsymbol{W}^{\text{head}} \in \mathbb{R}^{C \times d}$ is a trainable weight matrix, where $C$ is the number of classes. The output $\boldsymbol{u} \in \mathbb{R}^C$ is the input to the softmax cross-entropy loss. All model parameters are trainable during fine-tuning, while in linear probing only the weight matrix $\boldsymbol{W}^{\text{head}}$ is trainable (and in fact learned via full-batch logistic regression).

In all training setups, we average the loss over all samples in the batch.

We pre-train CRATE-MAE on ImageNet-1K (Deng et al., 2009). We employ the AdamW optimizer (Loshchilov & Hutter, 2019). We configure the learning rate as $3 \times 10^{-5}$, weight decay as $0.1$, and batch size as $4,096$.

We fine-tune and linear probe our pre-trained CRATE-MAE on the following target datasets: CI-FAR10/CIFAR100 (Krizhevsky et al., 2009), Oxford Flowers-102 (Nilsback & Zisserman, 2008), Oxford-IIIT-Pets (Parkhi et al., 2012). For each fine-tuning task, we employ the AdamW optimizer (Loshchilov & Hutter, 2019). We configure the learning rate as $5 \times 10^{-5}$, weight decay as $0.01$, and batch size as $256$. For each linear probing task, we use the linear probing functionality in Scikit-Learn (Pedregosa et al., 2011). For each evaluation we choose several regularizers $C \in \{10^0, 10^1, 10^2, 10^3, 10^4, 10^5\}$, train a logistic regression model on features from the whole dataset, and choose the logistic regression model with the best performance. All numbers are reported on the test sets.

For experiments, we use the model configurations reported in Table 1. From this table, there are two unspecified hyperparameters, namely $\lambda$ and $\eta$. In all experiments we fix $\eta = 0.1$, sincce it only multiplies with a trainable matrix and $\lambda$. In numerical experiments we use $\lambda = 0.5$, while figures are generated from models with hyperparameter $\lambda = 5.0$, though the difference in numerical performance and figure quality is marginal between the two settings (Table 6).

To allow transfer learning, in all training and evaluations setups we first resize our input data to 224 height and width. For data augmentations during pre-training and fine-tuning, we also adopt several standard techniques: random cropping, random horizontal flipping, and random augmentation (with number of transformations $n = 2$ and magnitude of transformations $m = 14$).

## B.2  PYTORCH-LIKE PSEUDOCODE

Listing 1: PyTorch-Like Code for MSSA and ISTA

```python
class ISTA:
    # initialization
    def __init__(self, dim, hidden_dim, dropout = 0., step_size=0.1,
         lambd=0.1):
        super().__init__()
        self.weight = nn.Parameter(torch.Tensor(dim, dim))
        with torch.no_grad():
            init.kaiming_uniform_(self.weight)
        self.step_size = step_size
        self.lambd = lambd
    # forward pass
    def forward(self, x):
        x1 = F.linear(x, self.weight, bias=None)
        grad_1 = F.linear(x1, self.weight.t(), bias=None)
        grad_2 = F.linear(x, self.weight.t(), bias=None)
        grad_update = self.step_size * (grad_2 - grad_1) - self.step_size
             * self.lambd
        output = F.relu(x + grad_update)
        return output

class MSSA:
    # initialization
    def __init__(self, dim, heads = 8, dim_head = 64, dropout = 0.):
        inner_dim = dim_head * heads
        project_out = not (heads == 1 and dim_head == dim)
        self.heads = heads
        self.scale = dim_head ** -0.5
        self.attend = Softmax(dim = -1)
        self.dropout = Dropout(dropout)
        self.qkv = Linear(dim, inner_dim, bias=False)
        self.to_out = Sequential(Linear(inner_dim, dim), Dropout(dropout))
             if project_out else nn.Identity()
    # forward pass
    def forward(self, x):
        w = rearrange(self.qkv(x), 'b n (h d) -> b h n d', h = self.heads)
        dots = matmul(w, w.transpose(-1, -2)) * self.scale
        attn = self.attend(dots)
        attn = self.dropout(attn)
        out = matmul(attn, w)
        out = rearrange(out, 'b h n d -> b n (h d)')
        return self.to_out(out)
```

Listing 2: PyTorch-Like Code for CRATE-MAE Encoder

```python
class CRATE_Encoder:
    # initialization
    def __init__(self, dim, depth, heads, dim_head, mlp_dim, dropout =
         0.):
        self.layers = []
        self.depth = depth
        for _ in range(depth):
            self.layers.extend([LayerNorm(dim), MSSA(dim, heads, dim_head,
                 dropout)])
            self.layers.extend([LayerNorm(dim), ISTA(dim, mlp_dim,
                 dropout)])
    # forward pass
    def forward(self, x):
        for ln1, attn, ln2, ff in self.layers:
            x_ = attn(ln1(x)) + x
            x = ff(ln2(x_))
        return x
```

Listing 3: PyTorch-Like Code for CRATE-MAE Decoder

```python
class CRATE_Decoder:
    # initialization
    def __init__(self, dim, depth, heads, dim_head, mlp_dim, dropout =
        0.):
        # define layers
        self.layers = []
        self.depth = depth
        for _ in range(depth):
            self.layers.extend([LayerNorm(dim), Linear(in_features=dim,
                out_features=dim, bias=False)])
            self.layers.extend([LayerNorm(dim), MSSA(dim, heads, dim_head,
                dropout)])
    # forward pass
    def forward(self, x):
        for ln1, f_linear, ln2, attn in self.layers:
            x_ = f_linear(ln1(x))
            x = ln2(x_) - attn(ln2(x_))
        return x
```

## B.3 VISUALIZATION METHODOLOGY

In this subsection we formally describe the procedures we used to generate the visualizations used to evaluate the segmentation property of CRATE-MAE in Figure 7. Much of this evaluation is the same as in Yu et al. (2023b), which initially demonstrates the emergent segmentation properties of white-box architectures.

### B.3.1 PCA VISUALIZATIONS

We recapitulate the method to visualize the patch representations in Figure 7a using PCA from Amir et al. (2021); Oquab et al. (2023); Yu et al. (2023b).

We first select $J$ images that belong to the same class, $\{\boldsymbol{X}_j\}_{j=1}^J$, and extract the token representations for each image at layer $\ell$, i.e., $\left[\boldsymbol{z}_{j,1}^\ell, \ldots, \boldsymbol{z}_{j,N}^\ell\right]$ for $j \in [J]$. In particular, $\boldsymbol{z}_{j,i}^\ell$ represents the $i^{\text{th}}$ token representation at the $\ell^{\text{th}}$ layer for the $j^{\text{th}}$ image. We then compute the first principal components of $\widehat{\boldsymbol{Z}}^\ell = \{\widehat{\boldsymbol{z}}_{1,1}^\ell, \ldots, \widehat{\boldsymbol{z}}_{1,N}^\ell, \ldots, \widehat{\boldsymbol{z}}_{J,1}^\ell, \ldots, \widehat{\boldsymbol{z}}_{J,N}^\ell\}$, and use $\widehat{\boldsymbol{z}}_{j,i}^\ell$ to denote the aggregated token representation for the $i$-th token of $\boldsymbol{X}_j$, i.e., $\widehat{\boldsymbol{z}}_{j,i}^\ell = [(\boldsymbol{U}_1^*\widehat{\boldsymbol{z}}_{j,i}^\ell)^\top, \ldots, (\boldsymbol{U}_K^*\widehat{\boldsymbol{z}}_{j,i}^\ell)^\top]^\top \in \mathbb{R}^{(p \cdot K) \times 1}$. We denote the first eigenvector of the matrix $\widehat{\boldsymbol{Z}}^*\widehat{\boldsymbol{Z}}$ by $\boldsymbol{u}_0$ and compute the projection values as $\{\sigma_\lambda(\langle \boldsymbol{u}_0, \boldsymbol{z}_{j,i}^\ell \rangle)\}_{i,j}$, where $\sigma_\lambda(x) = \begin{cases} x, & |x| \geq \lambda \\ 0, & |x| < \lambda \end{cases}$ is the hard-thresholding function. We then select a subset of token representations from $\widehat{\boldsymbol{Z}}$ with $\sigma_\lambda(\langle \boldsymbol{u}_0, \boldsymbol{z}_{j,i}^\ell \rangle) > 0$. which correspond to non-zero projection values after thresholding, and we denote this subset as $\widehat{\boldsymbol{Z}}_s \subseteq \widehat{\boldsymbol{Z}}$. This selection step is used to remove the background (Oquab et al., 2023). We then compute the first three right singular vectors of $\widehat{\boldsymbol{Z}}_s$ with the first three eigenvectors of the matrix $\widehat{\boldsymbol{Z}}_s^*\widehat{\boldsymbol{Z}}_s$ denoted as $\{\boldsymbol{u}_1, \boldsymbol{u}_2, \boldsymbol{u}_3\}$. We define the RGB tuple for each token as:

$$[r_{j,i}, g_{j,i}, b_{j,i}] = [\langle \boldsymbol{u}_1, \boldsymbol{z}_{j,i}^\ell \rangle, \langle \boldsymbol{u}_2, \boldsymbol{z}_{j,i}^\ell \rangle, \langle \boldsymbol{u}_3, \boldsymbol{z}_{j,i}^\ell \rangle], \quad i \in [N], j \in [J], \boldsymbol{z}_{j,i}^\ell \in \widehat{\boldsymbol{Z}}_s. \tag{B.3}$$

Next, for each image $\boldsymbol{X}_j$ we compute $\boldsymbol{R}_j, \boldsymbol{G}_j, \boldsymbol{B}_j$, where $\boldsymbol{R}_j = [r_{j,1}, \ldots, r_{j,N}]^\top \in \mathbb{R}^{d \times 1}$ (similar for $\boldsymbol{G}_j$ and $\boldsymbol{B}_j$). Then we reshape the three matrices into $\sqrt{N} \times \sqrt{N}$ and visualize the "principal components" of image $\boldsymbol{X}_j$ via the RGB image $(\boldsymbol{R}_j, \boldsymbol{G}_j, \boldsymbol{B}_j) \in \mathbb{R}^{3 \times \sqrt{N} \times \sqrt{N}}$.

### B.3.2 VISUALIZING ATTENTION MAPS

We recapitulate the method to visualize attention maps in Abnar & Zuidema (2020); Caron et al. (2021).

Table 3: **Top-1 classification accuracy of CRATE-MAE-Base when pre-trained on ImageNet-1K and fine-tuned on classification for various datasets.** We compare a fine-tuned model which was pre-trained on the MAE task with a model trained from scratch on classification ("*random init*") using exactly the same experimental conditions. Our results show that the representation learning occurring during pre-training substantially improves performance on downstream tasks.

| Classification Performance | CIFAR 10 | CIFAR 100 | Oxford Flowers | Oxford-Pets |
|---|---|---|---|---|
| CRATE-MAE-Base (trained) | 96.8 | 80.3 | 78.5 | 76.7 |
| CRATE-MAE-Base (random init) | 85.1 | 58.8 | 38.0 | 28.8 |

Table 4: **Average reconstruction loss over the training and validation sets of ImageNet-1K for both CRATE-MAE-Base and ViT-MAE-Base**. We see that the performance of CRATE-MAE-Base, while a bit worse than ViT-MAE-Base, obtains promising performance on the challenging masked autoencoding task.

| Reconstruction Loss | Training Loss | Validation Loss |
|---|---|---|
| CRATE-MAE-Base | 0.265 | 0.302 |
| ViT-MAE-Base | 0.240 | 0.267 |

Table 5: **Top-1 classification accuracy of CRATE-MAE-Base when pre-trained on ImageNet-1K and linear probed for classification on CIFAR-10, when pre-trained using different mask percentage (i.e., number of masked tokens in each sample).** This shows that CRATE-MAE models with 75% of the tokens masked during training tend to have the most structured representations that are useful for downstream tasks, an empirical conclusion that echoes He et al. (2022), but a wide range of mask percentages result in good representations. (*Note:* This table uses the hyperparameter setting $\lambda = 5.0$.)

| Classification Accuracy | 25% Masked | 50% Masked | 75% Masked | 90% Masked |
|---|---|---|---|---|
| CRATE-MAE-Base | 69.78 | 75.97 | 75.99 | 73.45 |

For the $k^{\text{th}}$ head at the $\ell^{\text{th}}$ layer of the encoder of CRATE-MAE, we compute the *self-attention map* $\boldsymbol{A}_k^\ell \in \mathbb{R}^N$ defined as follows:

$$\boldsymbol{A}_k^\ell = \begin{bmatrix} A_{k,1}^\ell \\ \vdots \\ A_{k,N}^\ell \end{bmatrix} \in \mathbb{R}^N, \quad \text{where} \quad A_{k,i}^\ell = \frac{\exp(\langle \boldsymbol{U}_k^{\ell*} \boldsymbol{z}_i^\ell, \boldsymbol{U}_k^{\ell*} \boldsymbol{z}_{\text{cls}}^\ell \rangle)}{\sum_{j=1}^N \exp(\langle \boldsymbol{U}_k^{\ell*} \boldsymbol{z}_j^\ell, \boldsymbol{U}_k^{\ell*} \boldsymbol{z}_{\text{cls}}^\ell \rangle)}. \tag{B.4}$$

where $\boldsymbol{z}_{\text{cls}}^\ell$ is the $\ell^{\text{th}}$ layer representation of the class token.

For each image, we reshape the attention matrix $\boldsymbol{A}_k^{L-1}$ for the penultimate layer $L-1$ into a $\sqrt{N} \times \sqrt{N}$ matrix and visualize the heatmaps as shown in Figure 7b. For example, the $i^{\text{th}}$ row and the $j^{\text{th}}$ column element of each heatmap in Figure 7b corresponds to the $m^{\text{th}}$ component of $\boldsymbol{A}_k^\ell$, where $m = (i-1) \cdot \sqrt{N} + j$. In Figure 7b, for each image we select one attention head $k$ of CRATE-MAE and visualize the attention matrix $\boldsymbol{A}_k^{L-1}$.

## B.4 Additional Experiments

In this section we perform more experiments to explore properties of the CRATE-MAE architecture.

First, in Table 3, we compare the fine-tuning performance of an CRATE-MAE-Base model with MAE pretraining against an CRATE-MAE-Base model with no pretraining at all (i.e., randomly initialized). We apply the same fine-tuning process to both models, and we observe a massive disparity in performance, where the pre-trained model succeeds while the randomly initialized model performs poorly. This indicates that the organized representations of a pre-trained CRATE-MAE-Base model are a strong starting point when fine-tuning for downstream tasks.

Next, in Table 4, we evaluate the reconstruction loss (measured in mean-squared error) of CRATE-MAE-Base versus ViT-MAE-Base, evaluated on the training and test sets of ImageNet-1K. We observe that while CRATE-MAE-Base performs slightly worse at masked reconstruction, the performance is still very reasonable.

Table 6: **Top-1 classification accuracy of CRATE-MAE-Base when pre-trained on ImageNet-1K and linear probed for classification on various datasets, when pre-trained using different $\lambda$.** This shows that CRATE-MAE models perform reasonably well at CRATE-MAE-Base scale across different values of $\lambda$.

| Classification Accuracy | $\lambda = 0.1$ | $\lambda = 0.5$ | $\lambda = 5.0$ |
|---|---|---|---|
| CRATE-MAE-Base | 83.33 | 80.87 | 75.99 |

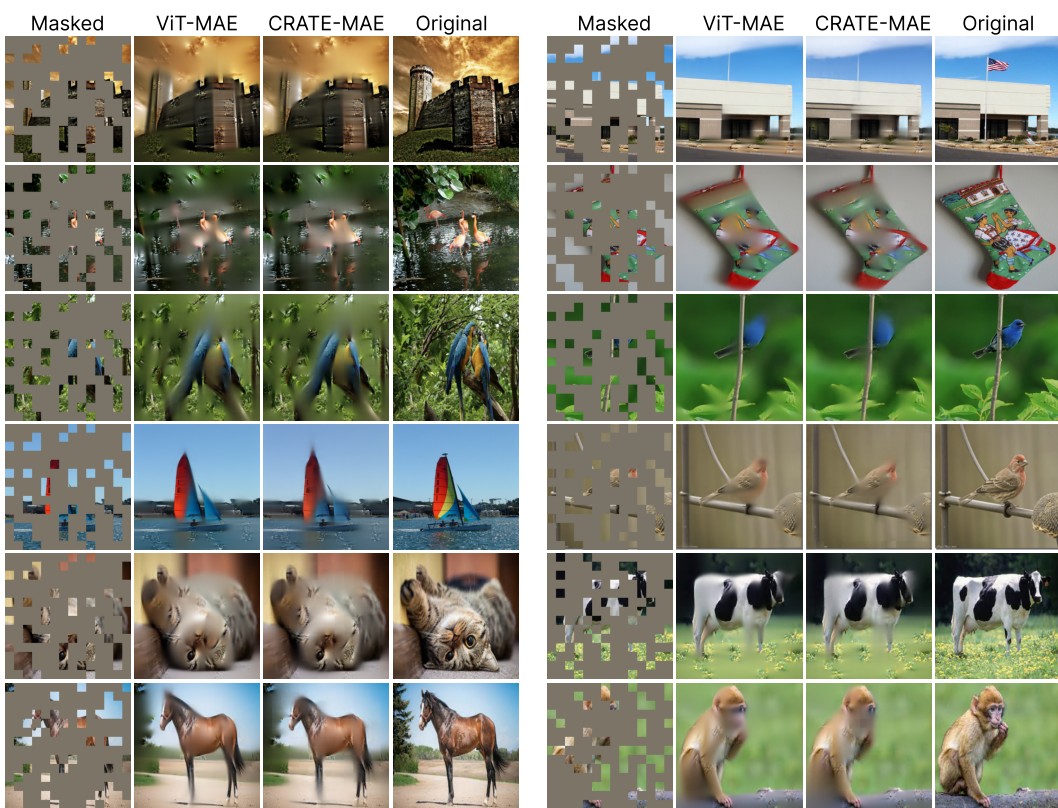

Figure 8: **More instances of parity between CRATE-MAE-Base and ViT-MAE-Base in the masked autoencoding task.** Echoing the message of Figure 6, we find that CRATE-MAE-Base and ViT-MAE-Base have similar performance on the masked autoencoding task, even as the CRATE-MAE-Base model is significantly more parameter-efficient.

In Table 5, we check the downstream performance and feature learning of CRATE-MAE-Base models when varying the mask size. We use test accuracy of linear probing on CIFAR10 as a proxy for the quality of the learned features. Our results show that the performance is maximized when the number of masked tokens is 75% of the number of total tokens, i.e., when 75% of tokens are maxed out. This confirms the experiments in He et al. (2022).

In Table 6, we check the downstream performance and feature learning of CRATE-MAE-Base models when varying the hyperparameter $\lambda$. We again use test accuracy of linear probing on CIFAR10 as a proxy for the quality of the learned features. Our results show that the performance is maximized when $\lambda = 0.1$, but all models perform reasonably well at CRATE-MAE-Base scale.

In Figure 8 we provide more examples of the masked autoencoding efficacy of both CRATE-MAE-Base and ViT-MAE-Base. Our results confirm those of Figure 6, namely CRATE-MAE-Base achieves parity with the much larger ViT-MAE-Base on the challenging masked autoencoding task.

In Figure 9, we provide more examples of the linearity of the representations within CRATE-MAE-Base. We see that over a wide variety of images, the first three principal components of each class correlate strongly with semantically meaningful patches of the input image. These results extend Figure 7a.

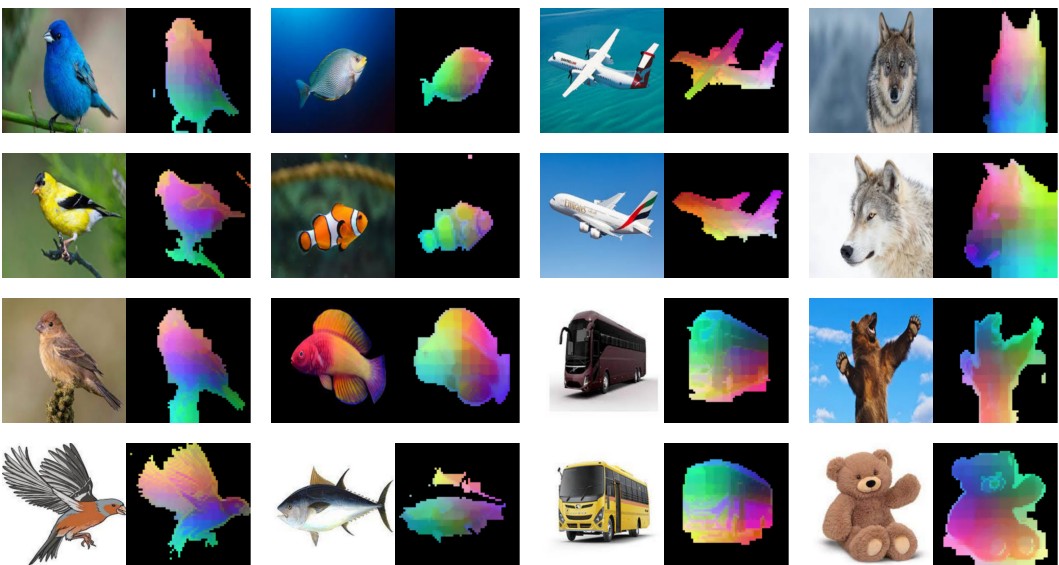

Figure 9: **More examples of linearized representations in CRATE-MAE-Base.** Echoing the message of Figure 7a, we find that CRATE-MAE-Base has a linear feature space. In particular, the first three principal components strongly correlate with the main semantic content of the image.

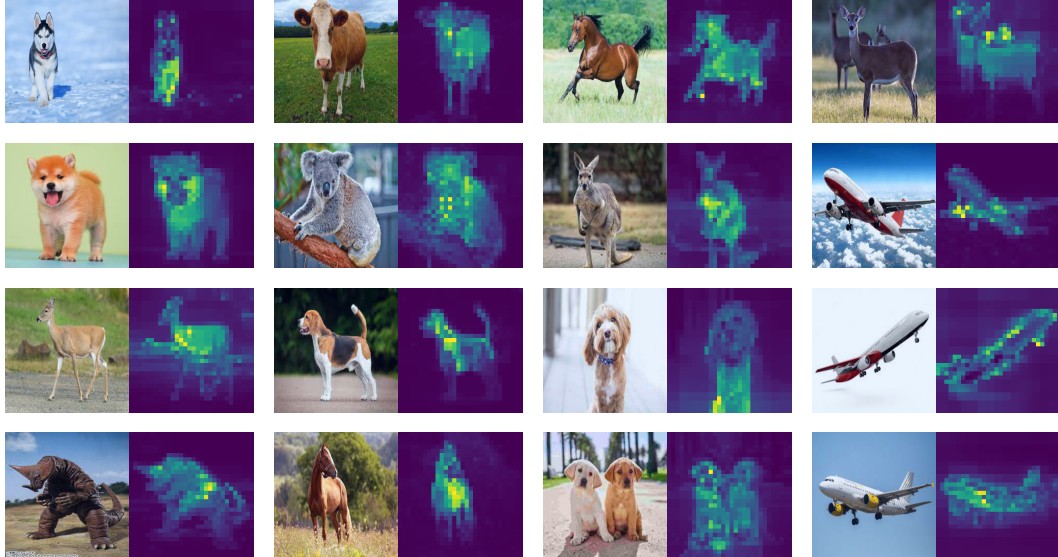

Figure 10: **More examples of interpretable attention maps in CRATE-MAE-Base.** Echoing the message of Figure 7b, we find that CRATE-MAE-Base has human-interpretable attention maps which semantically segment the foreground of input images.

In Figure 10, we provide more examples of the interpretability of selected attention outputs within CRATE-MAE-Base. We see that over a wide variety of images, the attention map succeeds at capturing many semantics of the input image. These results extend Figure 7b.

In Figure 11 we apply (nearly) the same methodology involved in constructing Figure 7, a qualitative demonstration of the feature quality of the CRATE-MAE-Base encoder, to evaluate the feature quality of the ViT-MAE-Base encoder. One necessary difference is that to evaluate the attention maps, the ViT does not have the $U_{[K]}^{\ell}$ matrices that CRATE does, but instead has three sets of matrices $Q_{[K]}^{\ell}$, $K_{[K]}^{\ell}$, and $V_{[K]}^{\ell}$; thus, we construct the attention maps via the following equation:

$$A_{k,i}^{\ell} = \frac{\exp(\langle K_k^{\ell} z_i^{\ell}, Q_k^{\ell} z_{\text{cls}}^{\ell}\rangle)}{\sum_{j=1}^{N} \exp(\langle K_k^{\ell} z_j^{\ell}, Q_k^{\ell} z_{\text{cls}}^{\ell}\rangle)}. \tag{B.5}$$

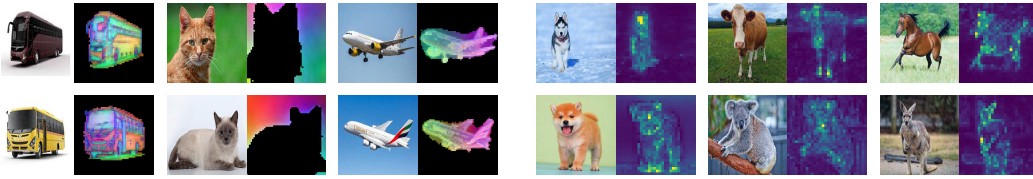

(a) Visualizing PCA of ViT-MAE-Base.  (b) Visualizing attention maps of ViT-MAE-Base.

Figure 11: **A comparison of CRATE to ViT-MAE in the setting of Figure 7.** We use the visualizations of PCA on the token representations and attention maps, introduced in Figure 7, to qualitatively evaluate the representation quality of the ViT-MAE-Base. By comparing this figure (Figure 11) and Figure 7, we observe that CRATE-MAE-Base attention maps contain more clear semantics than those from ViT-MAE-Base, while both CRATE-MAE-Base and ViT-MAE-Base have nearly-linear representation spaces wherein semantic concepts correspond to the first three principal components.

Overall, Figure 7 and Figure 11 demonstrate that, at least empirically, the attention semantics in CRATE-MAE-Base are significantly better and clearer than ViT-MAE-Base. The reason that CRATE-MAE models have semantically meaningful attention maps may be due to our white-box design, namely the fact that in CRATE-MAE we have $\boldsymbol{Q}_{[K]}^{\ell} = \boldsymbol{K}_{[K]}^{\ell} = \boldsymbol{V}_{[K]}^{\ell} = \boldsymbol{U}_{[K]}^{\ell*}$; indeed, the fact that setting $\boldsymbol{Q}_{[K]}^{\ell} = \boldsymbol{K}_{[K]}^{\ell} = \boldsymbol{V}_{[K]}^{\ell} = \boldsymbol{U}_{[K]}^{\ell}$ yields semantically meaningful attention maps has been shown in other work, albeit in a different setting (Yu et al., 2023b). The reason that the ViT-MAE has semantically meaningful attention maps, albeit not as clear and worse than CRATE-MAE, may be due to several different factors such as using the class token as a register (Darcet et al., 2023). Nevertheless, the features in both models have linear structures, or at least each class' representations have three principal components which correlate strongly with the semantics of the class.

