# OpenReview forum: "Masked Completion via Structured Diffusion with White-Box Transformers"
_ICLR.cc/2024/Conference — ICLR 2024 poster_

### Official Review · Reviewer_2DR1 · 2023-10-22

**Soundness:** 2 fair
**Presentation:** 3 good
**Contribution:** 3 good
**Rating:** 8
**Confidence:** 4

**Summary:**

This paper introduces CRATE-MAE, a novel white-box deep network architecture designed for large-scale unsupervised representation learning. Unlike traditional networks, CRATE-MAE is rooted in the mathematical connection between diffusion, compression, and masked completion. Each layer of this architecture has a clear, interpretable role, transforming data into structured representations and vice versa. The study's key contribution is adapting the white-box design for unsupervised learning, a notable departure from its typical supervised applications. Empirically, CRATE-MAE outperforms traditional models on large imagery datasets with 30% fewer parameters, while offering structured and semantically meaningful representations.

**Strengths:**

1. **Theoretical Depth and Scientific Rigor**: The research stands out for its robust theoretical foundation, seamlessly intertwining denoising-diffusion models and information theory with white-box models. The model's design is intricately tied to its theoretical underpinnings, exemplifying the paper's scientific precision and thoroughness.

2. **Problem Significance**: By addressing representation learning in high-dimensional data and delving into the untapped potential of white-box models in unsupervised settings, the paper carves a significant niche in the contemporary machine learning domain. Indeed, it paves the way for new avenues of exploration for the broader ML community.

**Weaknesses:**

Firstly, I'd like to clarify that my emphasis is not solely on state-of-the-art results. My questions regarding the experiments stem from the belief that robust ideas and arguments deserve to be bolstered by thorough experiments.

1. **Evaluation Concerns**: The introduction of the CRATE-MAE architecture in the paper falls short in offering a comprehensive quantitative analysis when compared to established benchmarks like MAE or contrastive methods. The results presented in Table 2 seem somewhat restrictive, making it challenging for readers to gauge the model's efficacy in relation to others.

2. **Local Self-Supervised Learning Comparison**: From a broader perspective on self-supervised learning (SSL), this paper could be classified under local SSL, emphasizing layer-specific objectives and training. Although this area might be less traversed, incorporating findings from related works, such as [A], would enhance the paper's credibility.
[A] Siddiqui, Shoaib Ahmed, et al. "Blockwise self-supervised learning at scale." arXiv preprint arXiv:2302.01647 (2023).

3. **Absence of Linear Probing Results**: Omitting linear probing results restricts the paper from showcasing the practicality and caliber of the representations derived using CRATE-MAE.

4. **Dataset Limitations**: The study's dependence on a confined dataset for classification raises concerns about the breadth of its applicability and potential generalization to diverse scenarios.

**Questions:**

I have no questions about the methodology part. Just please add more quantitative results to paper.

---

> ### Author Response · Authors · 2023-11-19
> **Author Response**
>
> Thank you for your valuable comments. We especially appreciate your remarks regarding the “theoretical depth and scientific rigor” of our work, as well as the significance of the problem. We would like to respectfully point out a minor error in your review: instead of using 30% fewer parameters than base MAE (that is, using 70% of the parameters of MAE), we use 30% of the total parameters of MAE. We hope that this further clarifies the value of our empirical evaluations.
>
> Below, we attempt to resolve your remaining comments and concerns.
>
> > Evaluation Concerns: The introduction of the CRATE-MAE architecture in the paper falls short in offering a comprehensive quantitative analysis when compared to established benchmarks like MAE or contrastive methods. The results presented in Table 2 seem somewhat restrictive, making it challenging for readers to gauge the model's efficacy in relation to others.
>
> > Absence of Linear Probing Results: Omitting linear probing results restricts the paper from showcasing the practicality and caliber of the representations derived using CRATE-MAE.
>
> Thank you for pointing it out. We summarize some recent experimental results, which we will put in the camera-ready version, which evaluate the linear probing and fine-tuning performance of CRATE-MAE-Base (44.6M parameters) versus MAE-Small (47.5M parameters) [1] over different mask ratios. This experimental result shows that while CRATE-MAE-Base slightly underperforms its black-box counterpart, the performance is still quite reasonable.
>
> |Model (mask ratio)|CRATE-MAE-Base (25%)|CRATE-MAE-Base (50%)|CRATE-MAE-Base (75%)|CRATE-MAE-Base (90%)|*MAE-Small (25%)*|*MAE-Small (50%)*|*MAE-Small (75%)*|*MAE-Small (90%)*|
> |-|-|-|-|-|-|-|-|-|
> Finetuning|89.87|92.83|96.4|96.58|95.79|96.8|97.61|97.34|
> Linear Probing|58.41|62.72|71.12|64.15|54.22|68.99|79.43|75.13|
>
> > Local Self-Supervised Learning Comparison: From a broader perspective on self-supervised learning (SSL), this paper could be classified under local SSL, emphasizing layer-specific objectives and training. Although this area might be less traversed, incorporating findings from related works, such as [A], would enhance the paper's credibility. [A] Siddiqui, Shoaib Ahmed, et al. "Blockwise self-supervised learning at scale." arXiv preprint arXiv:2302.01647 (2023).
>
> Thank you for pointing this out. We will add a comparison to these methods in the camera-ready version, since we were not aware of them previously. We note that there is a significant conceptual difference between our work and “local SSL”; instead of training layer-wise, we train our model end-to-end and the architecture (derived through discretized structured denoising-diffusion) ensures that the layer-wise characteristics (e.g. compression/sparsity improvements) are preserved. In contrast to the layer-wise training in [A], where each layer has an interpretable goal but black-box mechanisms, each layer and operator in our white-box model is interpretable by construction, and the interpretations hold throughout training.
>
> > Dataset Limitations: The study's dependence on a confined dataset for classification raises concerns about the breadth of its applicability and potential generalization to diverse scenarios.
>
> We pre-train on ImageNet-1K, a standard large 2D image classification dataset commonly used for pre-training large image models. When fine-tuning for classification, we use many datasets such as CIFAR10/CIFAR100, Oxford Flowers, and Oxford Pets. We believe that this resolves any concern about fine-tuned classification performance overfitting to one dataset.
>
> Again, thank you for your helpful comments. Please let us know if you have any more concerns or questions. We look forward to a fruitful discussion period.
>
> [1]: Gandelsman, Y., Sun, Y., Chen, X., & Efros, A. (2022). Test-time training with masked autoencoders. Advances in Neural Information Processing Systems, 35, 29374-29385.

---

> > ### Comment · Reviewer_2DR1 · 2023-11-20
> > **Further question**
> >
> > Thank you for the clarification on the results and comparison. However, I'm curious as to why the model does not utilize true layer-wise training. The paper outlines explicit objectives for each layer, suggesting that they can be conceptually disentangled. From my own experiments, I understand that end-to-end training often outperforms layer-wise approaches. However, in your case, it seems like adopting a layer-wise manner would be a natural extension. Could you elaborate on this choice?

---

> > > ### Author Response · Authors · 2023-11-20
> > > **Author Response**
> > >
> > > Thank you for the interesting question! We reiterate that our CRATE-MAE network has the benefit of being layer-wise interpretable (in terms of operational characteristics, etc), both pre- and post-training. While layer-wise training would most likely ensure that this layer-wise interpretability holds even after training, this is also shown to be true with end-to-end training. Overall, while layer-wise training may confer several benefits to such-trained models in general, the characteristics we care about for CRATE-MAE are preserved with end-to-end training, and so we choose the latter, as it is more conventional for training deep networks. You are right that layer-wise training of white-box models would be an interesting extension for future work, but it is out of scope of the current paper. We will add a comment to this effect in the camera-ready version.

---

> > > > ### Comment · Reviewer_2DR1 · 2023-11-22
> > > > **Thanks for the response**
> > > >
> > > > I have no more technical questions any more. My suggestion is that, whenever you submit a paper, please include as much experiments as possible in the initial manuscript, instead of adding them in the rebuttal phase. Scalability is crucial for SSL, and this paper still leaves questions about its application to different tasks and data volumes.
> > > >
> > > > I have raised my score to 8. Good Luck.

---

> > > > > ### Author Response · Authors · 2023-11-23
> > > > > **Author Response**
> > > > >
> > > > > Thank you for your comments and feedback! Thank you also for raising your score.

---

### Official Review · Reviewer_86oH · 2023-11-04

**Soundness:** 3 good
**Presentation:** 3 good
**Contribution:** 3 good
**Rating:** 6
**Confidence:** 4

**Summary:**

This paper proposes a white-box diffusion model and unifies it with data compression under a single framework. Based on this, the method proposed by the author has achieved results comparable to the state-of-the-art, which validates their proposed theory.

**Strengths:**

Novelty in Approach: This paper is the first attempt to turn the diffusion model into a white-box network, and it has achieved good results.

Versatility: Its methodology and conclusions can be used as good reference for subsequent research on white-box neural networks.

**Weaknesses:**

1. In Sec 2.2, the authors intend to learn representations Z, and hope that the results learned are low-dimensional, sparse, bijective, etc.
If the method proposed by the authors is a white box, then these properties of Z should be verifiable through experiments.
Therefore, the authors should provide experimental results of representations Z to support their theory.

2. I checked the provided Pytorch code and find that MSSA and ISTA are composed of Linear layer, which implies large GPU Memory consumption.
So I'm wondering whether this method can be extended to large images, just like stable diffusion.
In addition, can the authors provide results from larger datasets? The images from CIFAR and ImageNet-1k are too small.

3. How effective is this network at unconstrained image generation?
In other words, if the learning target is pure noise, which may not be viewed as an image compression task, would this method still work?

**Questions:**

Please refer to my comments in the weakness part.

---

> ### Author Response · Authors · 2023-11-19
> **Author Response**
>
> Thank you for your valuable comments. We would like to respectfully reiterate that the methodology we introduce is not a diffusion model _per se_ – when one discusses “diffusion models” they usually refer to generative models trained through score matching [1]. On the other hand, in our work we have introduced a framework for  invertible representation-learning models, whose representations are achieved through discretizations of probability flow equations, and trained via masked autoencoding (among many possible tasks). Thus the mechanism is analogous, not identical, to diffusion models.
>
> Below, we attempt to resolve the remaining questions and concerns.
>
> > In Sec 2.2, the authors intend to learn representations Z, and hope that the results learned are low-dimensional, sparse, bijective, etc. If the method proposed by the authors is a white box, then these properties of Z should be verifiable through experiments. Therefore, the authors should provide experimental results of representations Z to support their theory.
>
> We measure the low-dimensionality (indeed, the _compression_) of $Z$ by the rate distortion $R^{c}(Z \mid U_{[K]})$. The network is shown to iteratively optimize this quantity at each layer, as demonstrated in Figure 4 of the paper. In this figure, not only are the final representations shown to be compressed, (almost) every intermediate representation in the network is shown to become progressively more compressed, showing that each layer implements compression, which aligns with our conceptual framework. Figure 4 also demonstrates a similar trend with the sparsity. We ensure that the representation is invertible by using a decoder, and training the encoder and decoder together on the masked autoencoding task. Figure 5 and Table 3 demonstrate invertibility of the representations.
>
> > I checked the provided Pytorch code and find that MSSA and ISTA are composed of Linear layer, which implies large GPU Memory consumption. So I'm wondering whether this method can be extended to large images, just like stable diffusion. In addition, can the authors provide results from larger datasets? The images from CIFAR and ImageNet-1k are too small.
>
> We are not sure what you mean; linear layers are present in the vast majority of machine learning systems, such as Stable Diffusion [2]. We resize our images to 224 x 224 pixels before processing in order to standardize the input processing and tokenization, so the dataset would not matter. Increasing this size would make further pre-training much more computationally expensive, and infeasible for us to complete during this rebuttal period. If you think it would be better, we can increase the size of the figures in the camera-ready version, so as to make the fine details easier to see.
>
> > How effective is this network at unconstrained image generation? In other words, if the learning target is pure noise, which may not be viewed as an image compression task, would this method still work?
>
> Unconditional image generation is at odds with masked autoencoders, which are empirically used to learn representations instead of perform generation, and mathematically attempt to output $\mathbb{E}[X \mid \mathtt{Mask}(X)]$ by minimizing the training loss. The best that masked autoencoders (even black-box MAE) can do with 100% mask ratio is to always output the constant image $\mathbb{E}[X]$, which would be a blurring of all possible images, and so would not be desirable. Our framework does not have to use masked autoencoding as a pretext task, however, and there are several such tasks (e.g. denoising) which may enable generation. Concrete extensions of our framework to generation are out of scope of this paper.
>
> Again, thank you for your helpful comments. Please let us know if you have any more concerns or questions. We look forward to a fruitful discussion period.
>
> [1]: Song, Y., & Ermon, S. (2019). Generative modeling by estimating gradients of the data distribution. Advances in neural information processing systems, 32.
>
> [2]: Rombach, R., Blattmann, A., Lorenz, D., Esser, P., & Ommer, B. (2021). High-resolution image synthesis with latent diffusion models. 2022 IEEE. In CVF Conference on Computer Vision and Pattern Recognition (CVPR) (pp. 10674-10685).

---

> ### Comment · Reviewer_86oH · 2023-11-21
> **Response to author's comment**
>
> Thank you for your detailed response and the effort put into addressing the concerns raised. After careful consideration of your responses and a re-evaluation of the manuscript, I have decided to maintain my original score.

---

> > ### Author Response · Authors · 2023-11-21
> > **Author Response**
> >
> > Thank you for carefully reviewing the paper. Please let us know if you have any other questions or comments during the review period.

---

### Official Review · Reviewer_vMuz · 2023-11-05

**Soundness:** 3 good
**Presentation:** 3 good
**Contribution:** 3 good
**Rating:** 5
**Confidence:** 4

**Summary:**

The paper extends the white-box models from supervised learning to unsupervised (or self-supervised) representation learning. And in particular, it trains a Masked Autoencoder (MAE) to learn transferrable representations to downstream classification tasks. Besides, it shows interesting visualizations to demonstrate learned representations are of emerging semantic properties.

**Strengths:**

+ Learning unsupervised representations for white-box models is a natural topic to study after white-box on supervised models, and could be of interest to members of the community.
+ The paper did a good job in presentation, making both the approach and the experiments easy to follow.
+ I like the visualizations in the end, which is a more intuitive addition to the numerical comparisons in the table.

**Weaknesses:**

- I see a lot of similarities to the main white-box paper (White-Box Transformers via Sparse Rate Reduction) which is already out there for supervised learning. I want more justifications for the meaningfulness of the current work apart from doing unsupervised learning.
- I am not convinced what's the advantage of models being white-box here, especially whether it has synergies with unsupervised learning. The paper explains a lot about what's done, but I don't see a strong motivation of why it is done (especially since the introduction is less of a story but more of a break-down of context and contributions).
- While the explorations are interesting, I don't think the claims are backed up well by experiments. This is my biggest concern and would like to raise them by asking questions. So please see below.

**Questions:**

* The paper lacks a fair comparison with MAE, especially on downstream tasks. Table 1 lists that MAE-base has more parameters and it could partially explain why Crate-MAE has a higher reconstruction loss. So what would a fair comparison (in terms of model parameters) look like? I think it is very easy to train MAE with a smaller encoder/decoder pair given the open-sourced code.
* How are the evaluations done in Table 2? Are they using the encoder only? Are they with fully-visible inputs?
* Table 4 again lacks a comparison of a similarly-sized MAE (how it performs on the same dataset as the mask ratio changes). It is not clear a conclusion from ImageNet classification can be transferred to CIFAR.
* Figure 4: how does it compare with a supervised encoder/decoder trained on ImageNet? I want to know the compression and sparsity behavior is a result of unsupervised learning, or a result of the architecture design. The same applies to Figure 6 and 7.

---

> ### Author Response · Authors · 2023-11-19
> **Author Response, Part 1**
>
> Thank you for your valuable comments. We especially appreciate your remarks regarding the work’s “good job in presentation” and the quality of the various visualizations. Below, we respond to the raised questions and comments.
>
> > I see a lot of similarities to the main white-box paper (White-Box Transformers via Sparse Rate Reduction) which is already out there for supervised learning. I want more justifications for the meaningfulness of the current work apart from doing unsupervised learning.
>
> While the main point of the work is to extend the white-box framework in [1] to unsupervised learning, we believe that this extension is in itself meaningful. To begin with, one cannot use the unrolled optimization framework in [1] to understand or construct white-box decoder models; this is because they map from structured feature spaces to the original data distribution, so they cannot optimize any representation learning objective at all. This means that one needs to develop a new conceptual toolkit to extend the white-box approach in [1] to unsupervised learning. In this work, we propose the novel “structured denoising-diffusion” framework towards this end. We recover the unrolled optimization framework from [1] as a special case, but this time the framework is able to handle inversion and conditional generation (e.g., able to construct a white-box decoder network) via time-reversal. Note that this structured denoising-diffusion framework could be of independent interest in building practical and principled models for various tasks (such as unsupervised generation and self-supervised learning) beyond what we explored in this work. What’s more, it relies on certain quantitative connections, derived in the paper (e.g. Theorem 1), between statistical notions of denoising and information-theoretic notions of lossy compression, which would be of independent technical and conceptual interest.
>
> While remarks to the above effect are interspersed through the paper, we will consolidate and expand on them in the camera-ready version.
>
> > I am not convinced what's the advantage of models being white-box here, especially whether it has synergies with unsupervised learning. The paper explains a lot about what's done, but I don't see a strong motivation of why it is done (especially since the introduction is less of a story but more of a break-down of context and contributions).
>
> Thank you for the comment on desiring more motivation; we will certainly include more exposition towards this end in the camera-ready version. At a high level, white-box or theoretically principled models serve to make models more understandable or interpretable at a high granularity. This line of work thus attempts to bridge the divide between theory and practice; such a divide exists in both supervised and unsupervised deep learning. For theorists, simpler and principled models are easier to analyze and build meaningful theory off of. For practitioners, white-box networks retain empirical benefits over black-box models. To wit, we demonstrate in the paper that we obtain comparable autoencoding results to the MAE-ViT with only 30% of the parameters. (Below, we will clarify several of your remarks about to what degree the results are “comparable”.) We also observe that attention maps in the encoder are interpretable.
>
> > The paper lacks a fair comparison with MAE, especially on downstream tasks. Table 1 lists that MAE-base has more parameters and it could partially explain why Crate-MAE has a higher reconstruction loss. So what would a fair comparison (in terms of model parameters) look like? I think it is very easy to train MAE with a smaller encoder/decoder pair given the open-sourced code.
>
> Thank you for pointing it out. We ran a comparison against MAE-Small [2], which uses a similar number of parameters (47.5M) as CRATE-MAE-Base (44.6M). Since you ask for reconstruction loss, we evaluate the reconstruction loss on the training and testing datasets of ImageNet-1K. It shows that CRATE-MAE slightly underperforms its black-box counterpart, but the performance remains reasonable.
>
> ||Train Reconstruction Loss|Test Reconstruction Loss|
> |-|-|-|
> |CRATE-MAE-Base|0.266|0.303|
> |MAE-Small|0.250|0.283|
>
>
> > How are the evaluations done in Table 2? Are they using the encoder only? Are they with fully-visible inputs?
>
> The precise fine-tuning methodology is discussed in Appendix D due to space constraints. In short: the fine-tuning networks use the encoder only with an attached classification head, and use fully-visible inputs (i.e., no masking).
>
> **(Remaining responses and references posted in a followup.)**

---

> ### Author Response · Authors · 2023-11-19
> **Author Response, Part 2**
>
> > Table 4 again lacks a comparison of a similarly-sized MAE (how it performs on the same dataset as the mask ratio changes). It is not clear a conclusion from ImageNet classification can be transferred to CIFAR.
>
> Thank you for pointing it out. We compared CRATE-MAE-Base against MAE-Small when pre-trained on different mask ratios, and conducted fine-tuning and linear probing on the training sets of CIFAR10. Below, we show the resulting models’ classification accuracies on the test sets of CIFAR10.
>
> |Model (mask ratio)|CRATE-MAE-Base (25%)|CRATE-MAE-Base (50%)|CRATE-MAE-Base (75%)|CRATE-MAE-Base (90%)|*MAE-Small (25%)*|*MAE-Small (50%)*|*MAE-Small (75%)*|*MAE-Small (90%)*|
> |-|-|-|-|-|-|-|-|-|
> Finetuning|89.87|92.83|96.4|96.58|95.79|96.8|97.61|97.34|
> Linear Probing|58.41|62.72|71.12|64.15|54.22|68.99|79.43|75.13|
>
>
> We are unfortunately not sure what you mean by “It is not clear a conclusion from ImageNet classification can be transferred to CIFAR.” The pre-training is on ImageNet, and the fine-tuning in the referenced table is on CIFAR10, so we are not extrapolating classification performance anywhere.
>
> > Figure 4: how does it compare with a supervised encoder/decoder trained on ImageNet? I want to know the compression and sparsity behavior is a result of unsupervised learning, or a result of the architecture design. The same applies to Figure 6 and 7.
>
> Thank you for the question. Actually, we observe from [1] that such a compression and sparsification effect holds for supervised classification training of the encoder network. Figures 6 and 7 are replicated in the supervised classification context in [3, Figures 1 and 3], which also demonstrates that black-box networks generally do not have this interpretability unless specifically trained for it [3, Figure 14]. To answer your question directly: [1] and [3] demonstrate that the main effect of the interpretability properties we observe is due to the _architecture design_, and _not_ due to training procedure. We will briefly clarify this in the camera-ready version.
>
> Again, thank you for your helpful comments. Please let us know if you have any more concerns or questions. We look forward to a fruitful discussion period.
>
> [1]: Yu, Y., Buchanan, S., Pai, D., Chu, T., Wu, Z., Tong, S., ... & Ma, Y. (2023). White-Box Transformers via Sparse Rate Reduction
>
> [2]: Gandelsman, Y., Sun, Y., Chen, X., & Efros, A. (2022). Test-time training with masked autoencoders. Advances in Neural Information Processing Systems, 35, 29374-29385.
>
> [3]: Yu, Y., Chu, T., Tong, S., Wu, Z., Pai, D., Buchanan, S., & Ma, Y. (2023). Emergence of segmentation with minimalistic white-box transformers.

---

> > ### Comment · Reviewer_vMuz · 2023-11-23
> >
> > Thanks, I acknowledge I have read the responses (and I should have said "It is not clear a conclusion *of* ImageNet classification can be transferred *from* CIFAR". I think they are detailed and helpful. However, since it is allowed to update the draft as well, I did expect more changes on the draft to reflect it would be "ready" before publication (sorry I checked late and there is not much time left for detailed updates now). Nonetheless, I think the rebuttal has largely addressed my concerns.

---

> > > ### Author Response · Authors · 2023-11-23
> > > **Author Response**
> > >
> > > Thank you for the clarification and your comments on the rebuttal. Regarding updating the PDF, note that we have provided updated results in the author comments; this is to communicate the results easier by putting all updates in one centralized location. In our camera-ready version, we will include all experimental results presented to the reviewers.

---

### Official Review · Reviewer_pN5Y · 2023-11-13

**Soundness:** 3 good
**Presentation:** 3 good
**Contribution:** 4 excellent
**Rating:** 8
**Confidence:** 3

**Summary:**

This paper generalizes the white-box design of transformer, i.e., CRATE, to the unsupervised representation learning context. The author finds out that the gradient on the rate distortions term $R^c(Z | U_[K])$ plays a similar role as the gradient for the score function with the noised input $\tilde Z$, which points towards the closest point to $\tilde Z$ on the data distribution support. Thus they construct a masked auto-encoder using CRATE backbone and achieves fine results on the representation learning tasks.

===============
The author address most of my concerns and I will increase my score

**Strengths:**

1. This work generalizes the white-box transformer model to the unsupervised representation learning task, which is a novel attempt to both the theory and empirical community.
2. The visualization results are quite impressive and show that CRATE-MAE can reconstruct the original data well, and Fig. 4 roughly shows that the compression measure and sparsity measure match the theory setting.

**Weaknesses:**

1. I think the comparisons between CRATE-MAE-Base amd MAE-Base are not fair. I understand that the empirical evaluations are not optimally engineered and actually the visualization results are every good. However, I still think that the author should compare CRATE-MAE and MAE with (almost) the same amount of parameters and different performance, or alternatively, (almost) the same performance and different amount of parameters, and then to compare these two models.
2. What's the choice of LASSO coefficient hyperparameter $\lambda$, and the step size of discretization $\kappa, \eta$? Are they chose carefully and is the model sensitive to them?

**Questions:**

1. Which dataset does Fig.4 belongs to? Does the model have the similar patterns as Fig.4 on other datasets evaluated in this paper?
2. Empirically, people think that the attention map $Q/K$ plays a different role as the mapping matrix $V$ and they'd better not be set to be the same. However, in this papers' theoretical framework, they can be set to the same parameter $U$. If change the mapping matrix from $U$ to $V\neq U$, will the performance of CRATE-MAE change a lot? If no, what's the main reason why CRATE-MAE has such property?
3. Will the layer normalization influence the rate deduction process, or it's just for making the training process more stable or other reasons?

---

> ### Author Response · Authors · 2023-11-18
> **Author Response, Part 1**
>
> Thank you for your valuable comments. We especially appreciate your remarks that our work is a “novel attempt to both the theory and empirical community” and that “[t]he visualization results are quite impressive.” Below, we respond to the raised questions and comments.
>
> > I think the comparisons between CRATE-MAE-Base amd MAE-Base are not fair. I understand that the empirical evaluations are not optimally engineered and actually the visualization results are every good. However, I still think that the author should compare CRATE-MAE and MAE with (almost) the same amount of parameters and different performance, or alternatively, (almost) the same performance and different amount of parameters, and then to compare these two models.
>
> Thank you for the comment. The CRATE-MAE-Base has similar parameters to MAE-Small [1], which uses a similar number of parameters (47.5M) as CRATE-MAE-Base (44.6M). In the following table, we demonstrate a comparison of linear probing (LP) and finetuning (FT) accuracy on the test data on CIFAR10 between these two models (pretrained on ImageNet-1K), showing that CRATE-MAE-Base is slightly outperformed by its black-box counterpart, but the performance remains reasonable.
>
>
> ||CIFAR10 FT|CIFAR10 LP|CIFAR100 FT|CIFAR100 LP|Oxford Flowers FT|Oxford Flowers LP|Oxford Pets FT|Oxford Pets LP|
> |-|-|-|-|-|-|-|-|-|
> |CRATE-MAE-Base|96.4|71.12|80.24|48.64|70.29|59.23|76.18|38.38|
> |MAE-Small|97.61|79.43|82.96|54.36|84.22|68.45|81.68|51.21|
>
>
> > What's the choice of LASSO coefficient hyperparameter $\lambda$, and the step size of discretization $\kappa$, $\eta$? Are they chose carefully and is the model sensitive to them?
>
> We follow the hyperparameter choices in [2], that is, $\lambda = 0.1$, $\eta = 0.1$, and $\kappa$ and $\epsilon^{2}$ tuned so that $\kappa p/(N\epsilon^{2}) = 1$. We do _not_ choose hyperparameter values carefully (e.g., through ablation or other robust tuning). Indeed, as you already noted the experiments in this work are intended as a proof of concept to illustrate the promise of our white-box approach and framework, though they already demonstrate some empirical benefits such as parameter saving and interpretability. If you would like a more detailed quantitative analysis, let us know and we can upload it to the camera-ready version, but performing a detailed robustness analysis on ImageNet would be computationally infeasible during the review period.
>
> > Which dataset does Fig.4 belongs to? Does the model have the similar patterns as Fig.4 on other datasets evaluated in this paper?
>
> Thank you for pointing out that the paper is missing this information; we will include it in the camera-ready version. Figure 4 demonstrates the compression and sparsity of tokens within an image from ImageNet-1K, averaged over 10,000 random samples with error bars displayed. All models in this work were pre-trained on ImageNet-1K, and this task is evaluated before fine-tuning on any datasets. Thus we are not sure if you suggest that we look at models that are pre-trained from scratch on the other datasets using masked autoencoding, or look at models that are pre-trained on ImageNet-1K and fine-tuned on other datasets. We have tried both on the CIFAR10 dataset; we generally observe the same trends as in the paper Figure 4 for both settings.
> - Here are the results for the model pre-trained on CIFAR10: https://imgur.com/a/KdTzZDU
> - Here are the results for the model pre-trained on ImageNet-1K and fine-tuned on CIFAR10: https://imgur.com/a/GZ2iYbK
>
> **(Remaining responses and references posted in a followup.)**

---

> ### Author Response · Authors · 2023-11-18
> **Author Response, Part 2**
>
> > Empirically, people think that the attention map $Q$/$K$ plays a different role as the mapping matrix $V$ and they'd better not be set to be the same. However, in this papers' theoretical framework, they can be set to the same parameter $U$. If change the mapping matrix from $U$ to $V \neq U$, will the performance of CRATE-MAE change a lot? If no, what's the main reason why CRATE-MAE has such property?
>
> Thank you for pointing this out! We were not aware that people suggest that the $Q$ and $K$ matrices have distinct conceptual roles from the $V$ matrices, and would be interested in any references(s) which explain(s) this, so that we could incorporate them into our paper. We believe that in fact the $Q$, $K$, and $V$ matrices could be considered conceptually in the same role. Our justification in this paper, as well as other works on white-box transformers [2, 3], yield this interpretation from both empirical and theoretical perspectives, as well as a proposal of Hinton: [4]. White-box transformers are simpler [2] and have different, more interpretable [3] operational characteristics than the ViT. Thus, the suggestion that $V \neq U$, obtained through some subtle post-hoc analysis in the ViT context, may not be straightforwardly carried over to CRATE.
>
>
> > Will the layer normalization influence the rate deduction process, or it's just for making the training process more stable or other reasons?
>
> The approximations made to derive the MSSA and ISTA blocks require the features to have constant-order magnitude, so some normalization is required in order to make the implementation match the underlying theoretical principles. Using layer-normalization as opposed to “true” normalization (i.e., manually normalizing each token feature to the unit sphere) is motivated by improved empirical performance and easier training. Our framework does not attempt to give a principled reason why we should use one over the other. We will incorporate this discussion into the camera-ready version.
>
> Once again, we thank you for your insightful comments. Please let us know if you have any other questions or concerns. We look forward to a fruitful discussion period.
>
> [1]: Gandelsman, Y., Sun, Y., Chen, X., & Efros, A. (2022). Test-time training with masked autoencoders. Advances in Neural Information Processing Systems, 35, 29374-29385.
>
> [2]: Yu, Y., Buchanan, S., Pai, D., Chu, T., Wu, Z., Tong, S., ... & Ma, Y. (2023). White-Box Transformers via Sparse Rate Reduction
>
> [3]: Yu, Y., Chu, T., Tong, S., Wu, Z., Pai, D., Buchanan, S., & Ma, Y. (2023). Emergence of segmentation with minimalistic white-box transformers.
>
> [4]: Hinton, G. (2023). How to represent part-whole hierarchies in a neural network. Neural Computation, 35(3), 413-452.

---

> ### Comment · Reviewer_pN5Y · 2023-11-21
>
> Thank you for your detailed response. Your additional experiments and illustration address most of my concerns and I will increase my score.
>
>  As for the sensitive analysis, the ablation study for parameters like $ \lambda $ may be helpful but not necessary since as you mentioned, the main focus of this study is to construct the theoretical framework of a white-box transformer.
>
> For the roles of $Q/K$ and $V$, I’m very sorry that my claim may be confusing. Some previous works like reformer[5] may claim that shared-QK parameters will not significantly influence the performance of model, but people may heuristically think that the attention matrix $W_Q$/$W_K$ and the value matrix $W_V$ will project the input to different spaces. Actually, I can’t find other reference papers for this common choice, and the materials you provide address my concern, I think it may be beneficial to add the discussion to the paper. Maybe the MSSA block and the update process (2.4~2.6) which is different from the heuristically designed transformer result in the effectiveness of using the single matrix $U$. And I think this is really an interesting point of this approach
>
> [5]  Nikita Kitaev, Łukasz Kaiser, and Anselm Levskaya. Reformer: The efficient transformer. ICLR, 2020.

---

> ### Author Response · Authors · 2023-11-21
> **Author Response**
>
> Thank you for the reply, and for raising the score!
>
> We totally agree with your point – we believe that the white-box MSSA block is an efficient and useful operator (in part) because of its derivation from first principles as compressing the representations towards the desired statistical model. In particular, such derivations from first principles could potentially inform useful architecture modifications, even if not all architectures currently have such interpretations.
>
> We will add this discussion on parameter-sharing (including some discussion of related work, such as [5], which considers sharing $Q$/$K$ parameters), and an ablation on the sparsity parameter $\lambda$ and learning rate $\eta$, to the camera-ready version.
>
> Thank you again! Please let us know if you have other questions or comments during the discussion period.

---

### Author Response · Authors · 2023-11-19
**Message to All Reviewers**

We thank all authors for their valuable comments. We are especially gratified by the reviewers’ recognition of several strengths of our work:
- The work is novel and of interest to the community:
    - “the work is …. a novel attempt to both the theory and empirical community” (pN5Y),
    - “is a natural topic [which] … could be of interest” (vMuz),
    - “the first attempt… and it has achieved good results… its methodology and conclusions can be used as good reference for subsequent research on white-box neural networks” (86oH),
    - “the paper carves a significant niche in the contemporary machine learning domain. Indeed, it paves the way for new avenues of exploration for the broader ML community.” (2DR1).
- The presentation and visualizations are good:
    - “The visualization results are quite impressive” (pN5Y),
    - “The paper did a good job in presentation, making both the approach and the experiments easy to follow…I like the visualizations in the end” (vMuz).

In the reviews, several reviewers broadly wanted to see more experimental evaluations and fair comparisons with the traditional MAE. We have conducted these experiments, which required us to train several (ViT-)MAE models, and are still running more. We placed the results in the individual responses, and will be updating them further over the next few days as the final results come in.

In the end, our work aims to partially bridge theory and practice with white-box deep networks, so it is significant that our theoretically principled white-box CRATE-MAE architecture achieves reasonable performance compared to the black-box ViT-MAE architecture.

We again thank the reviewers for their comments, which we will incorporate into the camera-ready version, and look forward to the discussion period.

---

### Meta-Review · Area_Chair_DiZ9 · 2023-12-05

**Metareview:**

This paper uncovers a quantitative connection between denoising and compression and uses it to design a white-box transformer for mask image modeling. I did not find technical flaws in the theory.

Empirically, the paper compares with the "black" counterpart (i.e. the original MAE with a small size) in terms of classification and transfer learning and obtains consistently worse yet comparable results. However, the proposed method enjoys a "white" property: the features are explainable and it is easy to obtain meaningful visualization results.

The rebuttal process was fruitful and several reviewers increased their scores. Reviewer vMuz gave a final score of 5 but said "*the rebuttal has largely addressed my concerns*." Other reviewers gave positive scores.

Overall, I think this paper should be accepted as a poster.

**Justification For Why Not Higher Score:**

Empirically, the paper compares with the "black" counterpart (i.e. the original MAE with a small size) in terms of classification and transfer learning and obtains consistently worse results.

It is still not that clear whether such "white" models can achieve the same performance and be scaled up at the same level as the black ones.

**Justification For Why Not Lower Score:**

This is an early exploration in an interesting direction. It offers explainable properties, which are missing in its black-box counterparts.

---

### Decision · Program_Chairs · 2024-01-16

Accept (poster)